# Fatty acid β-oxidation is required for the differentiation of larval hematopoietic progenitors in *Drosophila*

Satish Kumar Tiwari[1], Ashish Ganeshlalji Toshniwal[2†], Sudip Mandal[2], Lolitika Mandal[1]*

[1]Developmental Genetics Laboratory, Department of Biological Sciences, Indian Institute of Science Education and Research (IISER) Mohali, Mohali, India; [2]Molecular Cell and Developmental Biology Laboratory, Department of Biological Sciences, Indian Institute of Science Education and Research (IISER) Mohali, Mohali, India

*For correspondence:
lolitika@iisermohali.ac.in

Present address: †University of Utah, Department of Biochemistry, Salt Lake City, United States

Competing interests: The authors declare that no competing interests exist.

**Abstract** Cell-intrinsic and extrinsic signals regulate the state and fate of stem and progenitor cells. Recent advances in metabolomics illustrate that various metabolic pathways are also important in regulating stem cell fate. However, our understanding of the metabolic control of the state and fate of progenitor cells is in its infancy. Using *Drosophila* hematopoietic organ: lymph gland, we demonstrate that Fatty Acid Oxidation (FAO) is essential for the differentiation of blood cell progenitors. In the absence of FAO, the progenitors are unable to differentiate and exhibit altered histone acetylation. Interestingly, acetate supplementation rescues both histone acetylation and the differentiation defects. We further show that the CPT1/whd (*withered*), the rate-limiting enzyme of FAO, is transcriptionally regulated by Jun-Kinase (JNK), which has been previously implicated in progenitor differentiation. Our study thus reveals how the cellular signaling machinery integrates with the metabolic cue to facilitate the differentiation program.

## Introduction

Recent studies have highlighted how metabolism regulates the state and fate of stem cells (*Ito and Suda, 2014*; *Shyh-Chang et al., 2013*; *Shyh-Chang and Ng, 2017*). Besides catering to the bioenergetic demands of a cell, metabolic intermediates can also alter the fate of stem cells via epigenetic mechanisms like histone modifications (*Atlasi and Stunnenberg, 2017*; *Saraiva et al., 2010*). Studies on diverse stem cell scenarios, primarily in Hematopoietic Stem Cells (HSCs), have established that at various developmental stages, stem cells have different metabolic requirements (*Kohli and Passegué, 2014*; *Suda et al., 2011*). Nevertheless, the metabolic demand of progenitors, the immediate descendants of stem cells, is yet to be fully elucidated.

Studies to date have evidenced that glucose metabolism impacts the onset and magnitude of HSC induction (*Harris et al., 2013*; *Shyh-Chang and Ng, 2017*), as well as HSC specification (*Oburoglu et al., 2014*). Another major metabolic state that is active in stem and progenitor cells is fatty acid oxidation (FAO) (*Ito et al., 2012*; *Knobloch et al., 2017*; *Lin et al., 2018*; *Wong et al., 2017*; *Ito et al., 2012*). Enzymatic activities of the members of FAO lead to the shortening of fatty acids and the production of acetyl-CoA in mitochondria. The acetyl-CoA thus produced can not only generate NADH and FADH$_2$ through TCA cycle but also can be utilized in acetylation of various proteins including histones (*Fan et al., 2015*; *Houten and Wanders, 2010*; *McDonnell et al., 2016*; *Wong et al., 2017*).

The primary goal of this study was to ascertain whether FAO regulates any aspect of the hemocyte progenitors in the *Drosophila* larval hematopoietic organ, lymph gland. The lymph gland is a multilobed structure consisting of a well-characterized anterior lobe (primary lobe) and

**eLife digest** Stem cells are special precursor cells, found in all animals from flies to humans, that can give rise to all the mature cell types in the body. Their job is to generate supplies of new cells wherever these are needed. This is important because it allows damaged or worn-out tissues to be repaired and replaced by fresh, healthy cells.

As part of this renewal process, stem cells generate pools of more specialized cells, called progenitor cells. These can be thought of as half-way to maturation and can only develop in a more restricted number of ways. For example, so-called myeloid progenitor cells from humans can only develop into a specific group of blood cell types, collectively termed the myeloid lineage.

Fruit flies, like many other animals, also have several different types of blood cells. The fly's repertoire of blood cells is very similar to the human myeloid lineage, and these cells also develop from the fly equivalent of myeloid progenitor cells. These progenitors are found in a specialized organ in fruit fly larvae called the lymph gland, where the blood forms. These similarities between fruit flies and humans mean that flies are a good model to study how myeloid progenitor cells mature.

A lot is already known about the molecules that signal to progenitor cells how and when to mature. However, the role of metabolism – the chemical reactions that process nutrients and provide energy inside cells – is still poorly understood. Tiwari et al. set out to identify which metabolic reactions myeloid progenitor cells require and how these reactions might shape the progenitors' development into mature blood cells.

The experiments in this study used fruit fly larvae that had been genetically altered so that they could no longer perform key chemical reactions needed for the breakdown of fats. In these mutant larvae, the progenitors within the lymph gland could not give rise to mature blood cells. This showed that myeloid progenitor cells need to be able to break down fats in order to develop properly.

These results highlight a previously unappreciated role for metabolism in controlling the development of progenitor cells. If this effect also occurs in humans, this knowledge could one day help medical researchers engineer replacement tissues in the lab, or even increase our own bodies' ability to regenerate blood, and potentially other organs.

uncharacterized posterior lobes (*Figure 1A*, *Banerjee et al., 2019*). The core of the primary lobe houses the progenitor populations and is referred to as the medullary zone (MZ), while the differentiated cells define the outer cortical zone (CZ, *Figure 1A'*). In between these two zones, lies a rim of differentiating progenitors or intermediate progenitors (IPs). The blood progenitors of late larval lymph gland are arrested in G2-M phase of cell cycle (*Sharma et al., 2019*), have high levels of ROS (*Owusu-Ansah and Banerjee, 2009*), lack differentiation markers, are multipotent (*Jung et al., 2005*) and are maintained by the hematopoietic niche/posterior signaling center, PSC (*Krzemień et al., 2007*; *Lebestky et al., 2003*; *Mandal et al., 2007*). The primary lobe has been extensively used to understand intercellular communication relevant to progenitor maintenance (*Gao et al., 2013*; *Giordani et al., 2016*; *Gold and Brückner, 2014*; *Hao and Jin, 2017*; *Krzemień et al., 2007*; *Krzemien et al., 2010*; *Lebestky et al., 2003*; *Mandal et al., 2007*; *Mondal et al., 2011*; *Morin-Poulard et al., 2016*; *Sinenko et al., 2009*; *Small et al., 2014*; *Yu et al., 2018*). Although these studies have contributed significantly toward our understanding of cellular signaling relevant for progenitor homeostasis, the role of cellular metabolism in regulating the state and fate of blood progenitors remains to be addressed.

Here, we show that the G2-M arrested hemocyte progenitors of the *Drosophila* larval lymph gland rely on FAO for their differentiation. While the loss of FAO prevents their differentiation, upregulation of FAO in hemocyte progenitors by either genetic or pharmacological means leads to precocious differentiation. More importantly, acetate supplementation restores the histone acetylation and differentiation defects of the progenitor cells observed upon loss of FAO. Our genetic and molecular analyses reveal that FAO acts downstream to the Reactive Oxygen Species (ROS) and c-Jun N-terminal Kinase (JNK) axis, which is essential for triggering the differentiation of these progenitors (*Owusu-Ansah and Banerjee, 2009*). In this study, we, therefore, provide the unknown link

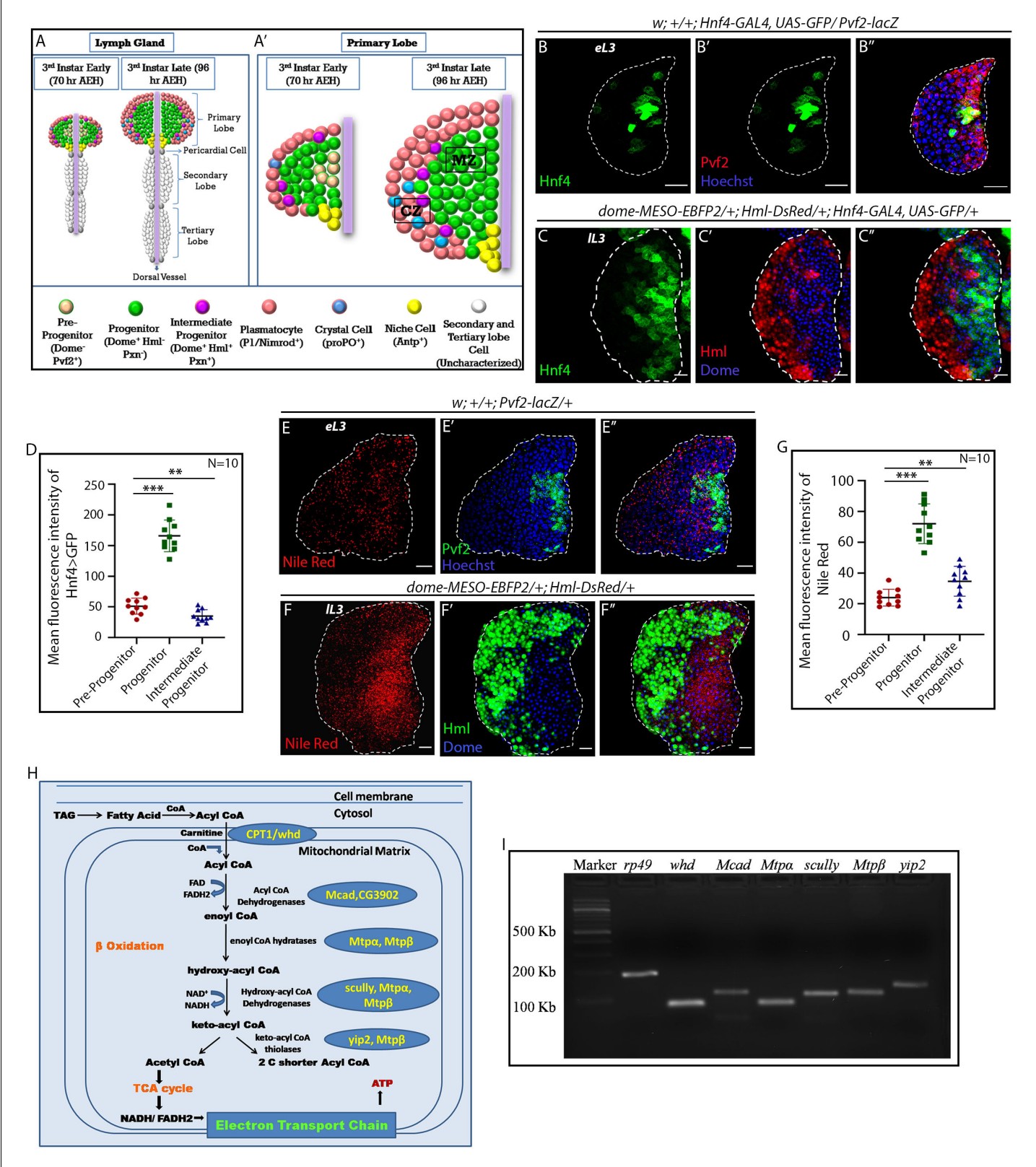

**Figure 1.** FAO genes are expressed in hemocyte progenitors of lymph gland. Age and genotype of the larvae are mentioned in respective panels. (A–A') Model of lymph gland of third early and third late instar stages depicting anterior primary lobes and posterior lobes. (A'). Primary lobe showing different subpopulations: Pvf2$^+$ Dome$^-$ pre-progenitor, Dome$^+$ progenitors and Dome$^+$ Pxn$^+$ Hml$^+$ Intermediate progenitors (IPs) in early third and late

*Figure 1 continued on next page*

*Figure 1 continued*

third instar larval stages. Progenitors are present in the core of the primary lobe called the medullary zone (MZ), and differentiated cells (Plasmatocytes and crystal cells) are present in the outer zone called cortical zone (CZ). (B–B'') Expression of *Hnf4-GAL4 > UAS-GFP* in Pvf2$^+$ pre-progenitors of the early third instar lymph gland. (C–C'') Expression of *Hnf4-GAL4 > UAS-GFP* in Dome$^+$ progenitors and Dome$^+$ Hml$^+$ Intermediate progenitors (IPs) shown in *dome-MESO-EBFP2/+; Hml-DsRed/+* genotype. (D). Quantitative analysis of B–C''- reveals that the Dome$^+$ progenitors have higher levels of Hnf4 expression. p-Value for *Hnf4-GAL4 > UAS-GFP* expression in Dome$^+$ progenitors is $9.55 \times 10^{-9}$ compared to control Pvf2$^+$ pre-progenitors. p-Value for *Hnf4-GAL4 > UAS-GFP* expression for Dome$^+$ Hml$^+$ IPs is $7.34 \times 10^{-3}$ compared to control Pvf2$^+$ pre-progenitors. (E–E'') Nile red staining in Pvf2$^+$ pre-progenitors of early third instar stage lymph gland. (F–F'') Expression of Nile red in Dome$^+$ progenitors and Dome$^+$ Hml$^+$ Intermediate progenitors (IPs) shown in *dome-MESO-EBFP2/+; Hml-DsRed/+* genotype (Dome$^+$: blue, Hml$^+$: green). (G). Quantitative analysis of E–F'' shows higher levels of neutral lipids in the Dome$^+$ progenitors. Compared to control Pvf2$^+$ pre-progenitors, p-Values for nile red expression in Dome$^+$ progenitors is $1.39 \times 10^{-7}$ and Dome$^+$ Hml$^+$ IPs pre-progenitors is $9.11 \times 10^{-3}$. Five optical sections of 1 µm thickness from the middle of the Z-stack were merged into a single section. (H) Schematic representation of FAO and the constituent enzymes. (I) Transcripts of β-oxidation enzymes, *whd, Mcad, Mtpα, scully, Mtpβ,* and *yip2* (Refer to H) can be detected in the third late instar lymph gland. eL3 and lL3 refer to the early and late instar lymph glands. Individual dots represent biological replicates. Values are mean ± SD, asterisks mark statistically significant differences (*p<0.05; **p<0.01; ***p<0.001, Student's *t*-test). Scale bar: 20 µm.

The online version of this article includes the following source data and figure supplement(s) for figure 1:

**Source data 1.** Contains numerical data plotted in *Figure 1D and G*.
**Figure supplement 1.** Temporal analysis of pre-progenitors in the lymph gland and mitochondrial analysis in Dome$^+$ progenitors.
**Figure supplement 2.** FAO components are expressed in hemocyte progenitor subpopulations in the lymph gland.
**Figure supplement 2—source data 1.** Contains numerical data plotted in *Figure 1—figure supplement 2C and F*.

that connects cellular signaling and metabolic circuitry essential for differentiation of the blood progenitors.

## Results

### Genes involved in FAO pathway are expressed in the hemocyte progenitors of late third instar lymph gland

*Drosophila* hemocyte progenitors in the lymph gland proliferate in the early larval stages (*Jung et al., 2005*; *Mondal et al., 2011*). Eventually, they undergo a G2-M arrest in late third instar (*Sharma et al., 2019*). Studies have identified that lymph gland hemocyte progenitors of the primary lobe can be grouped into three subpopulations: Dome⁻ pre-progenitors, Dome$^+$ progenitors, and Dome$^+$ Pxn$^+$ Hml$^+$ Intermediate progenitors (*Banerjee et al., 2019*). These progenitor subpopulations will be henceforth referred to as pre-progenitors, progenitors, and IPs, respectively. The pre-progenitors can also be visualized by Pvf2 expression in first, second, and early third instar larval lymph gland (*Ferguson and Martinez-Agosto, 2017*, *Figure 1—figure supplement 1A–B*). Since in the late third instar lymph gland, only progenitors and IPs are present (*Figure 1A'*), we analyzed the early third instar lymph gland to characterize the pre-progenitors.

To ascertain the involvement of FAO, if any, in hemocyte progenitors of larval lymph gland, we monitored the expression of *Hepatocyte Nuclear Factor 4 (Hnf4)* (*Palanker et al., 2009*), an essential gene of larval FAO and lipid mobilization. As evident from *Figure 1B–D*, G2–M arrested progenitors (visualized by *dome-MESO-EBFP2$^+$*) express high levels of *Hnf4 > GFP* compared to pre-progenitors (visualized by Pvf2 expression) and IPs (Dome$^+$ Hml$^+$). Interestingly, neutral lipids (visualized by Nile red staining: *Figure 1E–G*) are conspicuous in late third instar blood progenitors (*dome-MESO-EBFP2$^+$*), compared to the pre-progenitors (*Figure 1E–E'' and G*) and IPs (*Figure 1F–F'' and G*). The lipid enrichment in the late progenitors is further evident upon LipidTOX (validated marker for neutral lipids) labeling (*Figure 1—figure supplement 2A–C*). Based on the presence of relatively high levels of lipid droplets in a non-lipid storage tissue and the expression of *Hnf4* in the progenitor cells, we speculated a developmental role of FAO in these cells. This prompted us to check for the expression of other genes involved in FAO (*Palanker et al., 2009*, *Figure 1H*). *Figure 1I* shows the expression of the rate-limiting enzyme of FAO, *withered* (*whd*, *Drosophila* homolog of CPT1: Carnitine palmitoyltransferase 1) along with *Mcad* (medium-chain acyl-CoA dehydrogenase), *Mtpα* (mitochondrial trifunctional protein α subunit: Long-chain-3-hydroxyacyl-CoA dehydrogenase), *scully* (3-hydroxyacyl-CoA dehydrogenase), *Mtpβ* (mitochondrial trifunctional protein β subunit: Long-chain-3-hydroxyacyl-CoA dehydrogenase) and *yip2* (yippee interacting protein 2: acetyl-CoA

acyltransferase) in the late lymph gland. Elevated levels of expression of acyl-Coenzyme A dehydrogenase (CG3902) is also seen in the progenitors as compared to the pre-progenitors and IPs (*Figure 1—figure supplement 2D–F*).

Major aspects of FAO takes place in the mitochondria where fat moiety is broken down to generate acetyl-CoA, NADH, and FADH$_2$ (*Bartlett and Eaton, 2004*), we, therefore, looked at the status of mitochondria in the progenitors as well as the differentiated hemocytes. The presence of an abundant reticular network of mitochondria is evident in the progenitors (*dome-GAL4 >UAS-mito-HA-GFP*, *Figure 1—figure supplement 1C–C'*, and *Video 1*). However, for reasons unknown to us Hml$^\Delta$-GAL4 is unable to drive *UAS-mito-HA-GFP*, therefore, we used streptavidin labeling to visualize the mitochondrial status in the differentiated hemocytes. Interestingly, differentiated hemocytes (*Hml$^\Delta$-GAL4 > UAS-GFP*) show less reticular mitochondria (labeled by Strepatvidin-Cy3, *Chowdhary et al., 2017*; *Hollinshead et al., 1997*, *Figure 1—figure supplement 1D–D'*) in comparison to the progenitors (*Hml>GFP* negative) thereby indicating a preference for FAO in the hemocyte progenitors.

Put together; the above results implicate FAO as the metabolic state of the Dome$^+$ cells of the primary lobe of the lymph gland. These observations, in turn, encouraged us to investigate the importance of FAO in maintaining the state and fate of these progenitors during development.

## Loss of FAO affects hemocyte progenitor differentiation

*Drosophila* ortholog of *CPT1*, *withered* (*whd*, *Figure 2A*), is a rate-limiting enzyme for FAO (*Strub et al., 2008*). The loss of function of *CPT1/whd*, therefore, blocks mitochondrial FAO (*Schreurs et al., 2010*). To investigate the role of FAO in late hemocyte progenitors, we employed a null allele of *withered, whd$^1$* (*Strub et al., 2008*). The primary lobes of *whd$^1$* homozygous lymph gland have abundant Dome$^+$ progenitors but drastically reduced number of differentiated hemocytes (P1: plasmatocytes *Figure 2B–C and D*; proPO: Crystal cells, *Figure 2E–F and G*), and Intermediate progenitors (Dome$^+$ Pxn$^+$, *Figure 2H–I''* and *Figure 2—figure supplement 1A*) compared to control.

Detailed temporal analysis of the dynamics of progenitor subpopulation during normal development, as well as upon loss of *whd$^1$* (*Figure 2J–S*), was next carried out. In sync with an earlier report (*Ferguson and Martinez-Agosto, 2017*), our analysis reveals that Dome$^-$ pre-progenitors (*Figure 2K*) are present in the developing lymph gland until the early third instar (*Figure 2L* and *Figure 2—figure supplement 1B*). Beyond this timeline, the subsets that populates the lymph gland are Dome$^+$ progenitors (*Figure 2M–N* and *Figure 2—figure supplement 1B*), and Dome$^+$ Pxn$^+$ IP cells (*Figure 2M–N* and *Figure 2—figure supplement 1B*). Interestingly, in *whd$^1$* mutant lymph glands, while the pre-progenitors are present till the early third instar stage (*Figure 2P* and *Figure 2—figure supplement 1C*), there is an abundance of progenitors (*Figure 2Q–S*) with a small number of IP cells (*Figure 2R–S*). Quantification of the above results reflects that in *whd$^1$* mutant the progenitors are rather stalled instead of undergoing a natural transition to IP cells with time (*Figure 2—figure supplement 1C*). During normal development, the first sign of differentiation (as evidenced by Pxn expression) occurs around 36 hr AEH, and by 96 hr AEH, there is a prominent cortical zone defined by the differentiating cells.

In contrast, the cortical zone is drastically reduced due to lack of differentiation in the *whd$^1$* mutant lymph gland. The timed analysis also revealed that the defect seen in differentiation in this mutant has an early onset (36 hr AEH, Compare *Figure 2O* with *Figure 2J*). These observations implicate that the lack of FAO

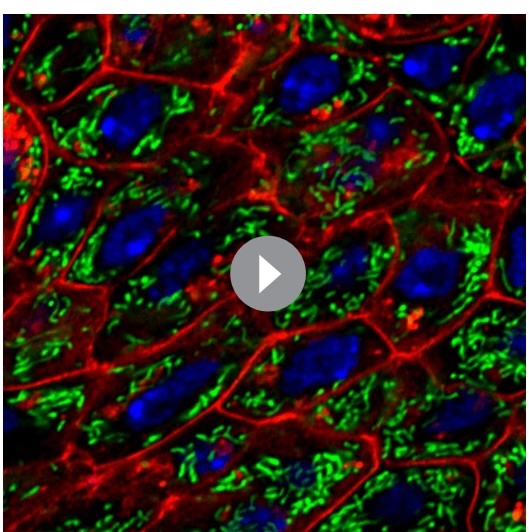

**Video 1.** Mitochondrial distribution in the progenitors (red, Dome$^+$) visualized by *UAS-mito-HA-GFP*.
https://elifesciences.org/articles/53247#video1

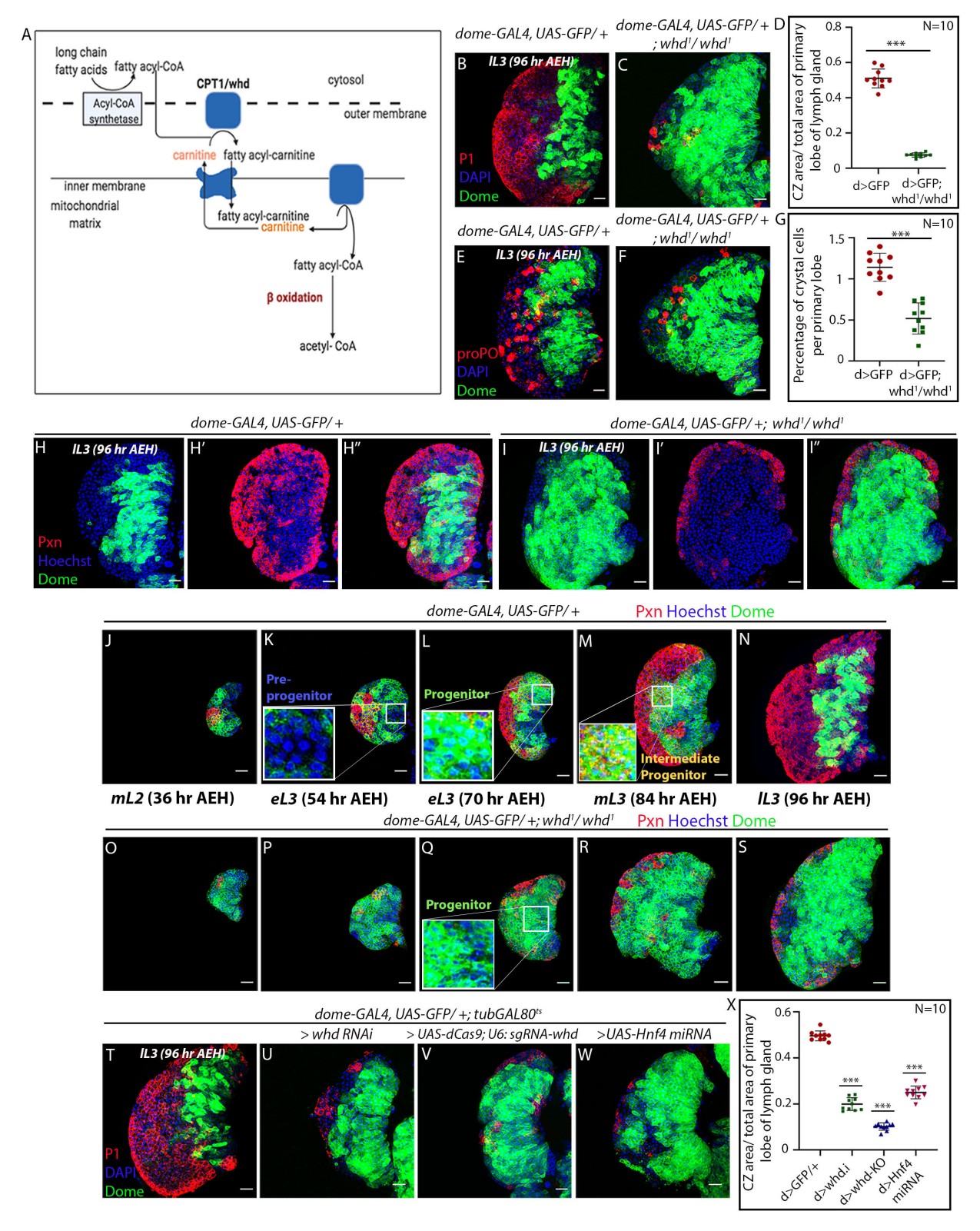

**Figure 2.** Loss of fatty acid β-oxidation affected differentiation of hemocyte progenitors of the lymph gland. (**A**) Schematic representation of fatty acid β-oxidation within the mitochondria of a cell. (**B–D**) Compared to control (**B**) decrease in differentiation (red, reported by P1 immunostaining) and increment in progenitor number (*dome > GFP*) is observed in the lymph gland of a homozygous null allele of *whd* (**C**). (**D**) Quantitative analysis of **B–C** reveals a significant increment in the number of Dome⁺ progenitors. p-Value for *dome-GAL4, UAS-GFP; whd¹/whd¹=2.67×10⁻¹⁰* compared to control.

*Figure 2 continued on next page*

*Figure 2 continued*

(**E–G**) Compared to control (**E**) decrease in crystal cell number (red, reported by proPO immunostaining) and increment in the progenitor cell population (*dome > GFP*) is observed in the lymph gland of the homozygous null allele of *whd* (**F**). (**G**). Quantitative analysis of results from **E–F** shows a significant drop in the number of crystal cells. p-Value for *dome-GAL4, UAS-GFP; whd¹/whd¹*=4.38×10⁻⁷ compared to control. (**H–I''**) The hemocyte progenitor subpopulation dynamics (red, reported by Pxn immunostaining and green marking *dome > GFP*) of Dome⁺ progenitors and Dome⁺ Pxn⁺ (IPs) in the late third instar lymph gland of control (**H–H''**) and homozygous null allele of *whd* (**I–I''**). (**J–S**) Spatio-temporal analysis of hemocyte progenitor subpopulations of Dome⁻ pre-progenitors, Dome⁺ progenitors, and Dome⁺ Pxn⁺ (IPs) (red, reported by Pxn immunostaining and green marking *dome > GFP*) observed in the lymph gland of control (**J–N**) and homozygous null allele of *whd* (**O–S**). Insets in **K, L,** and **M** show pre-progenitors, progenitors and intermediate progenitors respectively in control and inset in **Q** shows abundant progenitors in the homozygous null allele of *whd*. (**T–X**) Compared to control (**T**) decrease in differentiation (red, reported by P1 immunostaining) and increase in progenitor number (*dome > GFP*) is observed in lymph gland upon progenitor specific RNAi based down-regulation of *whd* (**U**) CRISPR-Cas9 based knock-out of *whd* (**V**) and miRNA based knockdown of *Hnf4* (**W**). (**X**) Quantitative analysis of the results from **T–W**, illustrating the significant increase in Dome⁺ progenitors upon targeted loss of FAO. p-Value for *dome-GAL4, UAS-GFP; tubGAL80ᵗˢ²⁰ > UAS-whd RNAi* = 2.84×10⁻¹⁵ compared to control. p-Value for *dome-GAL4, UAS-GFP; tubGAL80ᵗˢ²⁰ > UAS-dCas9; U-6: sgRNA-whd* = 3.84×10⁻¹⁹. p-Value for *dome-GAL4, UAS-GFP; tubGAL80ᵗˢ²⁰ > UAS-Hnf4. miRNA* =6.04×10⁻¹⁴. Individual dots represent biological replicates. Values are mean ± SD, asterisks mark statistically significant differences (*p<0.05; **p<0.01; ***p<0.001, Student's *t*-test). Scale bar: 20 μm.

The online version of this article includes the following source data and figure supplement(s) for figure 2:

**Source data 1.** Contains numerical data plotted in *Figure 2D,G and X*.

**Figure supplement 1.** Fatty acid β-oxidation is essential for lymph gland progenitor differentiation.

**Figure supplement 1—source data 1.** Contains numerical data plotted in *Figure 2—figure supplement 1A,B,C,G,M and Q*.

**Figure supplement 2.** Model depicting the posible role of Hnf4 and FAO in hemocyte progenitor differentiation.

---

dampens the differentiation process of Dome⁺ progenitors.

Since the differentiation of progenitors is affected, we next performed an RNAi-mediated down-regulation of *whd* by the TARGET system (*McGuire et al., 2004*; *Figure 2—figure supplement 1D*). Progenitor-specific downregulation (*dome-GAL4, UAS-GFP; tubGAL80ᵗˢ²⁰; UAS-whd RNAi*) results in a halt in differentiation, as evidenced by an increase in the area of *dome > GFP* and a concomitant decline in adjoining CZ (visualized by differentiated plasmatocyte (Nimrod: P1, *Figure 2U*) compared to control (*Figure 2T*). Upon activation of *whd RNAi* from a different source (VDRC) by another independent progenitor specific driver *TepIV-GAL4* (*Figure 2—figure supplement 1E–G*), a similar result is obtained. Additionally, progenitor-specific knockout of *whd* by CRISPR/Cas9 system (*Hsu et al., 2014*) supports the above results (*Figure 2V and X*).

Likewise, knockdown of the key player in lipid metabolism *Hnf4* results in a decline in the differentiation of the progenitors, endorsing the role of FAO in progenitor differentiation (*Figure 2W and X*, *Figure 2—figure supplement 2*). Lymph glands from a hetero-allelic combination of *dHNF4* (*dHNF4^Δ17^/dHNF4^Δ33^: null allele of Hnf4*) (*Palanker et al., 2009*), exhibits an abundant progenitor pool (Shg: DE-Cadherin) coupled with the reduction in differentiated cells (Pxn, compare *Figure 2—figure supplement 1I with 1H*, and *Figure 2—figure supplement 1M*), further denoting that FAO disruption indeed leads to compromised differentiation of progenitors.

To further verify our observations, we next analyzed the homozygous mutant of two essential enzymes of β−oxidation: *Mtpα* (mitochondrial trifunctional protein α subunit: Long-chain-3-hydroxyacyl-CoA dehydrogenase), and *Mtpβ* (mitochondrial trifunctional protein β subunit: Long-chain-3-hydroxyacyl-CoA dehydrogenase) (*Kishita et al., 2012*). Primary lobes from homozygous *Mtpα^[KO]^* and *Mtpβ^[KO]^* loss of function has a large progenitor pool (Shg: DE-Cadherin) at the expense of differentiated cells (Pxn) (compare *Figure 2—figure supplement 1J–K with H*), a phenotype similar to the whd¹ (*Figure 2—figure supplement 1L and M*). DE-cadherin enrichment in the FAO loss of function progenitors is also indicative of high-maintenance signals (*Gao et al., 2014*).

In addition to genetic knockdown and classical loss-of-function analyses, we performed pharmacological inhibition of FAO in *Drosophila* larvae (*Figure 2—figure supplement 1N–Q*). Larvae grown in food supplemented with FAO inhibitors, Etomoxir (*Lopaschuk et al., 1988*), and Mildronate also demonstrate a more than two-fold reduction in the differentiation of the progenitors of the primary lobe.

Collectively, our genetic and pharmacological studies illustrate the cell-autonomous role of FAO in the differentiation of blood progenitors of the lymph gland.

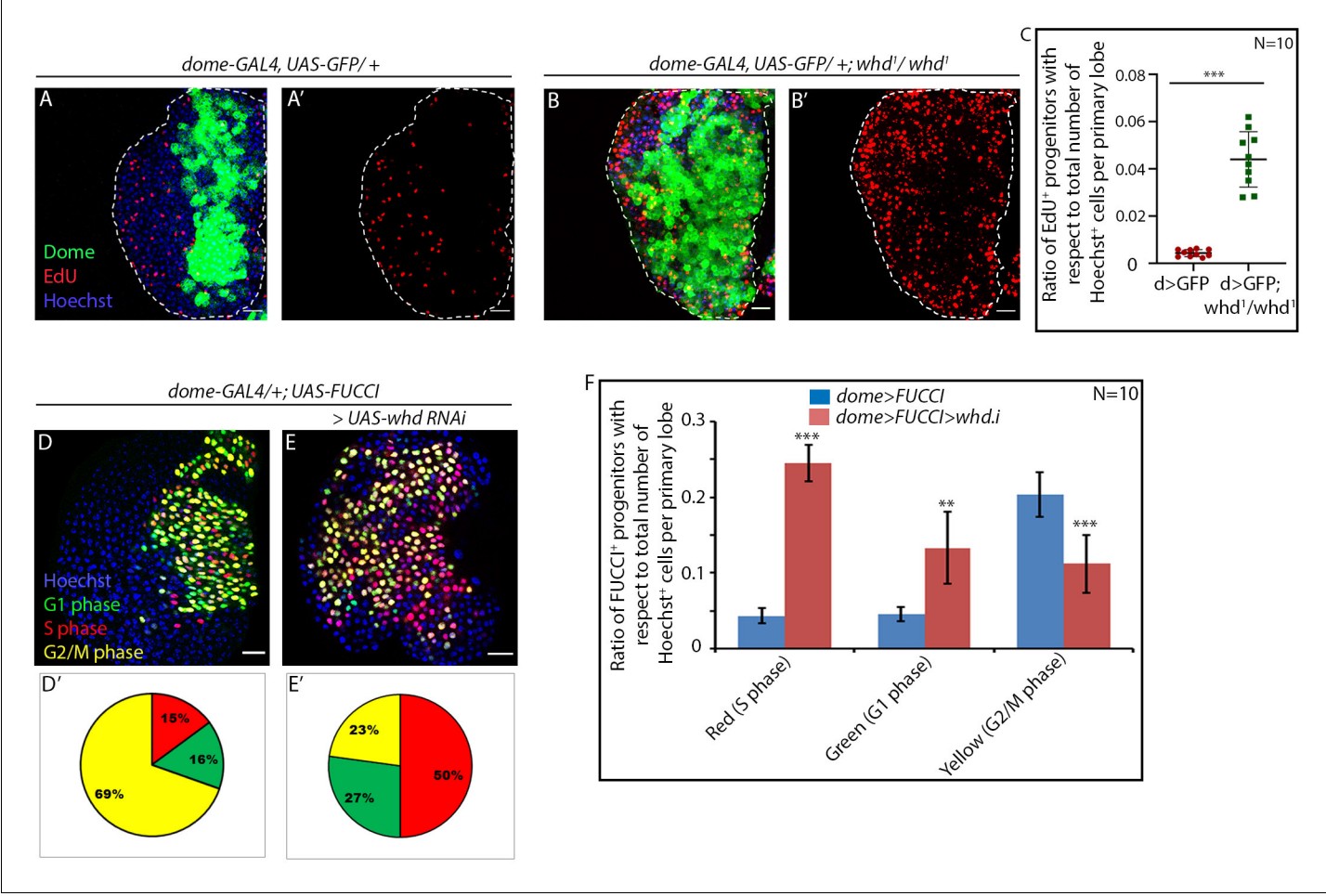

**Figure 3.** Loss of fatty acid β-oxidation causes an increase in proliferation of hemocyte progenitors of the lymph gland primary lobe. Genotype of the larvae are mentioned in respective panels. (A–C) The difference in proliferation status (reported by EdU incorporation) in the lymph gland of third late instar control larvae (A–A') compared to *whd* null mutant (B–B'). (C). Quantitative analysis of the results from A–B' illustrates a significant increase in proliferation in *whd1* Dome+ progenitors. p-Value for *dome-GAL4, UAS-GFP; whd1/whd1*=1.71×10⁻⁶ compared to control. (D–F) Difference in cell cycle status (reported by Fly-FUCCI) in the lymph gland of third late instar control larvae (D) compared to the progenitor-specific RNAi-based down-regulation of *whd* (E). (D'–E'): Pie chart depicting the fraction of G1 (green), S(red), and G2/M (yellow) progenitors in J–K. (F) Quantitative analysis of the results from D–E. p-Value for red cells in *dome-GAL4, UAS-Fly-FUCCI; UAS-whd RNAi = 1.41×10⁻¹¹*, p-Value for green cells in *dome-GAL4, UAS-Fly-FUCCI; UAS-whd RNAi = 2×10⁻⁴*, p-Value for yellow cells in *dome-GAL4, UAS-Fly-FUCCI; UAS-whd RNAi = 1.5×10⁻⁵* compared to control. ns.=not significant, Individual dots represent biological replicates. Values are mean ± SD, asterisks mark statistically significant differences (*p<0.05; **p<0.01; ***p<0.001, Student's t-test). Scale bar: 20 μm.

The online version of this article includes the following source data and figure supplement(s) for figure 3:

**Source data 1.** Contains numerical data plotted in *Figure 3C,D', E' and F*.

**Figure supplement 1.** Loss of fatty acid β-oxidation caused higher redox levels and an increase in maintenance factor of hemocyte progenitors of the lymph gland.

**Figure supplement 1—source data 1.** Contains numerical data plotted in *Figure 3—figure supplement 1C and F*.

## Loss of FAO causes an increase in progenitor proliferation

Previous study from our laboratory demonstrated that G2-M arrest is a hallmark of the otherwise proliferating progenitors prior to differentiation (*Sharma et al., 2019*). In contrast, we found that *whd1* homozygous progenitors exhibit a stark increase in EdU incorporation implicating their highly proliferating status upon loss of FAO when compared to age-isogenised controls (*Figure 3A–C*). To have a further insight into the cell cycle status, we expressed Fly-FUCCI-fluorescent ubiquitination-based cell cycle indicator, specifically in the progenitors. This indicator employs two fluorescent probes: the first probe is an E2F moiety fused to GFP, which is degraded by Cdt2 during the

S-phase. This construct allows the visualization of G1, G2, and M-phase cells by GFP expression. The second probe is a fusion of Red Fluorescent Protein (mRFP.nls) and CycB moiety. It is degraded by the anaphase-promoting complex/cyclosome (APC/C) during midmitosis, thereby reporting the cell in S, G2, and M-phase. Together this system allows the visualization of cells in G1, S, and G2/early mitosis by green, red, and yellow signals, respectively (*Zielke et al., 2014*).

We found that, instead of being in a G2-M arrest, the hemocyte progenitors in *whd* mutant are actively proliferating, as evidenced by more cells in S-phase (red) compared to the control (*Figure 3D–D' with E-E'*). Quantitative analyses reveal more than three-fold increase in the number of cells in the S phase with a concomitant drop in G2-M arrested progenitors (compare *Figure 3D' with E' and F*).

Together these results assert that FAO disruption in proliferating progenitors doesn't allow them to halt at G2-M and subsequently differentiate.

## Failure in differentiation of hemocyte progenitors upon loss of FAO is not due to decline in ROS levels

The perturbation of differentiation prompted us to look at the status of both differentiation and maintenance factors, per se in these hemocyte progenitors. Although the progenitor pool in the larval lymph gland is heterogenous (*Baldeosingh et al., 2018*; *Banerjee et al., 2019*; *Sharma et al., 2019*), our timed analysis indicates that the majority of the progenitors populating the late third instar lymph gland expresses dome. Hedgehog signaling has been implicated in the maintenance of the dome expressing progenitors (*Baldeosingh et al., 2018*; *Mandal et al., 2007*; *Sharma et al., 2019*; *Tokusumi et al., 2010*). Hematopoietic niche/PSC releases Hh, which leads to the expression of the Hh signal transducer Cubitus interruptus (Ci[155][*Alexandre et al., 1996*]) in the progenitors. The homozygous $whd^1$ lymph gland progenitors express a higher level of Ci[155] compared to control (compare *Figure 3—figure supplement 1B-B'' with A-A''* and quantitated in *Figure 3—figure supplement 1C*) correlating with higher proliferation and less differentiation (*Figure 2B–D*). This observation, along with enrichment of DE-cadherin in FAO mutants (*Figure 2—figure supplement 1H–M*), endorses high-maintenance signal in the progenitors.

Reactive oxygen species (ROS) is the major signal attributed to the differentiation of the hemocyte progenitors of the lymph gland (*Owusu-Ansah and Banerjee, 2009*). High levels of developmentally generated ROS trigger Jun Kinase (JNK) signaling, which sets these progenitors toward the differentiation program (*Owusu-Ansah and Banerjee, 2009*). Since homozygous $whd^1$ progenitors fail to differentiate, we rationalized that this might be due to the drop in the differentiation signal ROS.

To probe this possibility, we analyzed the levels of ROS in $whd^1$ lymph glands by dihydroxy ethidium (DHE) staining. Quite strikingly, $whd^1$ homozygous progenitors exhibited elevated levels of ROS (Compare *Figure 3—figure supplement 1D–D'' with 3E-E''* and quantified in *Figure 3—figure supplement 1F*). A similar observation of increased ROS in *CPT1* knockdown endothelial cells has been reported in another study (*Kalucka et al., 2018*). Therefore, we can infer that the halt in differentiation observed in the Ci[155] enriched progenitors of homozygous $whd^1$ is not an outcome of compromised ROS level.

Collectively, these results indicate that despite achieving a high ROS level than the control, the progenitors are unable to move into differentiation, indicating that FAO might act downstream to ROS.

## Upregulation of FAO causes precocious G2-M arrest and differentiation of blood progenitors

Due to the central role of carnitine in fat metabolism, it is often used as a supplement for enhancing fat oxidation (*Pekala et al., 2011*; *Wall et al., 2011*). Several studies have concluded that FAO can be upregulated in the cells by L-carnitine supplementation (*Pekala et al., 2011*; *Sahlin, 2011*; *Wall et al., 2011*).

Our loss-of-function genetic analyses illustrated that FAO is essential for the differentiation of hemocyte progenitors. Whether it is sufficient for the progenitor differentiation was addressed by feeding L-carnitine supplemented food (100 mM concentration for 48 hr) to wandering third instar larvae. Compared to control (*Figure 4A*), the lymph gland from L-carnitine fed larvae (96 hr AEH)

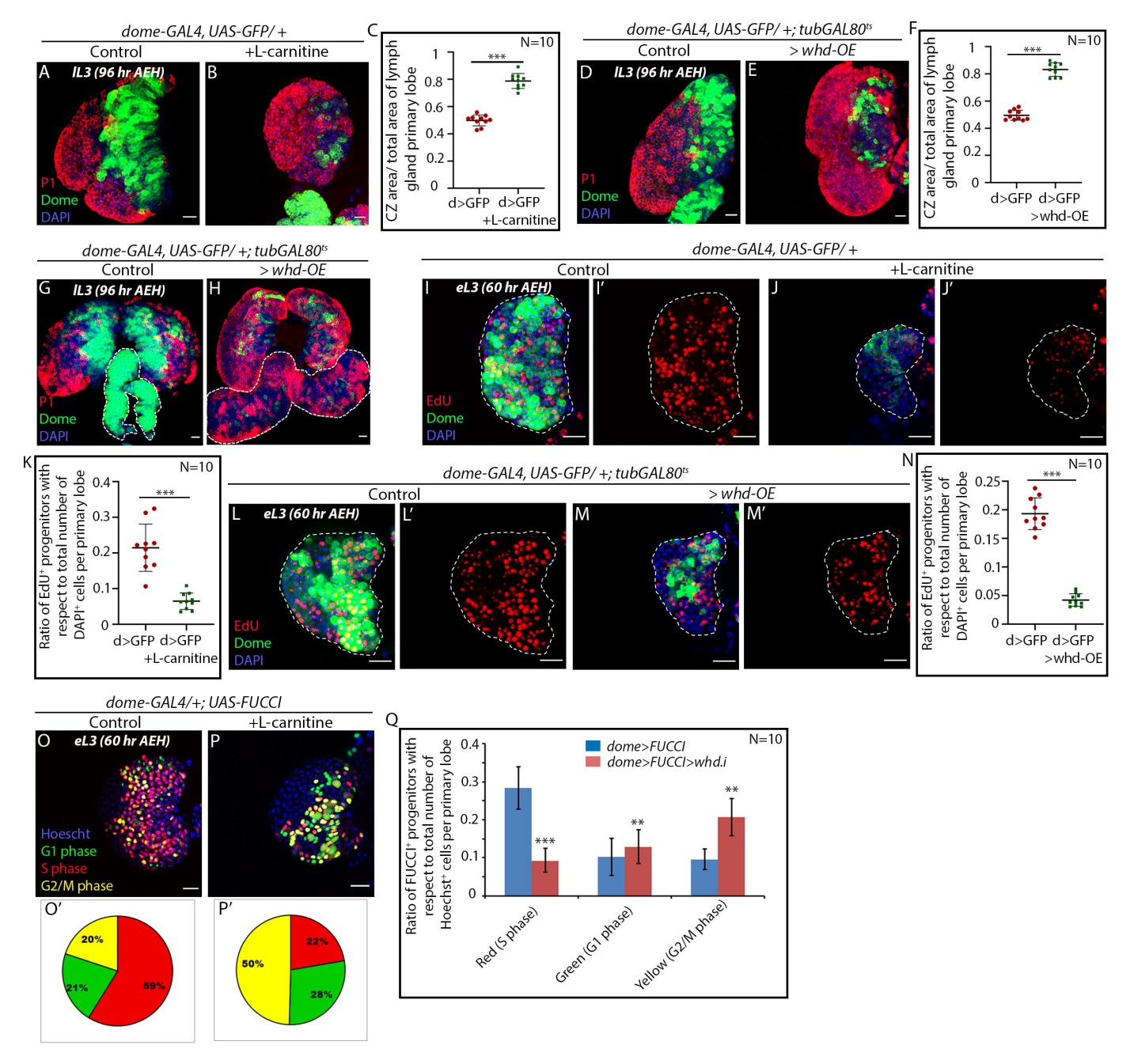

**Figure 4.** FAO upregulation results in precocious differentiation and G2 arrest in hemocyte progenitors. Age and genotype of the larvae are mentioned in respective panels. (**A–C**) Comparison of differentiation (marked by P1) levels in *dome > GFP* lymph gland of control (**A**) and L-carnitine supplemented (**B**) larvae. (**C**) Quantitative analysis of results from **A–B** showing increased differentiation upon L-carnitine supplementation. p-Value for *dome-GAL4, UAS-GFP* = $2.37 \times 10^{-10}$ supplemented with L-carnitine compared to control. (**D–F**) Comparison of differentiation (marked by P1) levels in *dome > GFP* lymph gland of control (**D**) and CRISPR-Cas9 mediated *whd* overexpression (**E**) in larval hemocyte progenitors. (**F**) Quantitative analysis of result from **D–E** depicting a significant increase in differentiation upon overexpression of *whd*. p-Value for *dome-GAL4, UAS-GFP; tubGAL80^{ts20} > UAS-whd-OE* = $5.82 \times 10^{-12}$ compared to control. (**G–H**) Comparison of differentiation (marked by P1) levels in *dome > GFP* lymph gland secondary lobes (marked by the white dotted boundary) of control (**G**) and CRISPR-Cas mediated *whd* overexpression (**H**) in larval hemocyte progenitors. (**I–K**) Proliferation status (marked by EdU) in third early instar hemocyte progenitors (*dome > GFP*) of control (**I–I′**) and L-carnitine supplemented (**J–J′**) larvae. (**K**) Quantitative analysis of results from **I–J′** reveals a decline in the number of proliferating Dome[+] progenitors upon FAO overexpression. p-Value for *dome-GAL4, UAS-GFP* fed with L-carnitine = $2.87 \times 10^{-5}$ compared to control. (**L–N**) The decline in proliferation status (marked by EdU) in third early instar hemocyte progenitors of CRISPR-Cas9 mediated *whd* overexpression (**M–M′**) compared to (*dome > GFP*) of control (**L–L′**). (**N**) Quantitative analysis of result from **L–M′**. p-Value for *dome-GAL4, UAS-GFP; tubGAL80^{ts20} > whd-OE* = $2.28 \times 10^{-9}$ compared to control. (**O–P**) Alteration in cell cycle

*Figure 4 continued on next page*

*Figure 4 continued*

status (reported by Fly-FUCCI) in the lymph gland of third late instar larvae grown in L-carnitine supplemented food (**P**) compared to control (*dome > UAS-FUCCI*) (**O**). (**O'–P'**): Pie chart depicting the fraction of G1 (green), S (red), and G2/M (yellow) progenitors in (**O-P**). (**Q**) Quantitative analysis of the results from **O–P**, illustrating the increase in G2-M upon FAO overexpression. p-Value for red cells in L-carnitine supplemented *dome-GAL4, UAS-FUCCI = 1.78×10⁻⁷*, p-Value for green cells in L-carnitine supplemented *dome-GAL4, UAS-FUCCI; UAS-whd RNAi = 2.16×10⁻¹*. p-Value for yellow cells in L-carnitine supplemented *dome-GAL4, UAS- FUCCI; UAS-whd RNAi = 1.71×10⁻⁵* compared to control. Individual dots represent biological replicates. Values are mean ± SD, asterisks mark statistically significant differences (*p<0.05; **p<0.01; ***p<0.001, Student's *t*-test). Scale bar: 20 μm.

The online version of this article includes the following source data and figure supplement(s) for figure 4:

**Source data 1.** Contains numerical data plotted in *Figure 4C,F,K,N,O', P', and Q*.
**Figure supplement 1.** L-carnitine supplementation rescues differentiation defect.
**Figure supplement 1—source data 1.** Contains numerical data plotted in *Figure 4—figure supplement 1E*.

exhibits a drastic reduction in progenitor zone (visualized by *dome > GFP*) with a concomitant increase in differentiation (visualized by P1: Nimrod; *Figure 4B–C*). The increase in differentiation by L-carnitine supplementation is also apparent in *whd*[1] heterozygous mutants. Feeding L-carnitine could rescue the differentiaion defects associated with *whd*[1] heterozygous mutants (Compare *Figure 4—figure supplement 1A–E*).

As a genetic correlate, *whd* was overexpressed by a Cas9-based transcriptional activator (BDSC68139) (*Ewen-Campen et al., 2017*) in the hemocyte progenitors following the scheme in *Figure 2—figure supplement 1D*. Overexpression of *whd* indeed results in an increase in the differentiation of hemocyte progenitors (compare *Figure 4D* with *Figure 4E–F*). It is interesting to note that the otherwise undifferentiated reserve progenitors of the secondary lobes also differentiate upon *whd* overexpression (marked by a dotted white line in *Figure 4G–H*).

We next employed dual fly-FUCCI construct, and EdU labeling to assay the cell cycle cell status of the FAO upregulated progenitors. The early third instar lymph glands from L-carnitine fed larvae exhibit a radical decline in EdU incorporation compared to the proliferating progenitors of the control larvae of similar age (compare *Figure 4I–I'* with *Figure 4J-J'* and quantified in *Figure 4K*). As a genetic correlate, we overexpressed *whd* in the progenitors, which also led to a decline in EdU incorporation (*Figure 4L–N*). Our FUCCI analysis reveals an abundance of G2-M progenitors in the early third instar larvae reared in L-carnitine supplemented food compared to control samples. Thus, less EdU incorporation due to upregulated FAO resulted in an early onset of G2-M arrest in the progenitors (compare *Figure 4O–O'* with *Figure 4P–P'* and quantified in *Figure 4Q*).

Put together; our results reveal that upon FAO upregulation, the progenitor experiences a precocious G2-M halt in their cell cycle. During normal development, the late progenitors also undergo a G2-M halt before they differentiate. We, thus, infer that FAO is imperative for the differentiation of lymph gland progenitors.

## FAO loss in hemocyte progenitors leads to sustained glycolysis

Next, we investigated whether a compromise in the intracellular energy source (ATP) is the reason behind the differentiation defect seen in the FAO mutants. Quite intriguingly, ATP levels of homozygous *whd*[1] larvae are comparable to similarly aged control (*Figure 5A*). This unaltered ATP level is in sync with our observation of higher proliferation observed in homozygous *whd*[1] hemocyte progenitors (*Figure 3B–B'*). Since higher proliferation is driven by elevated glycolysis in different scenarios (*Lunt and Vander Heiden, 2011*), we hypothesized that loss of FAO might push the progenitors towards a higher glycolytic index. We performed an in vivo glucose uptake assay employing fluorescent derivative of glucose, 2-NBDG (*Zou et al., 2005*) in the late third instar lymph gland of both control and FAO mutant. In control, late third instar Dome⁺ progenitors exhibit low glucose uptake (*Figure 5B–B''* and *Figure 5D*) compared to the higher uptake detectable in the peripheral hemocyte population marked by Hml (*Figure 5—figure supplement 1A–A'''* and quantified in *Figure 5—figure supplement 1B*). In sharp contrast, higher glucose uptake is evident in the FAO mutant progenitors (*Figure 5C–C''* and quantified in *Figure 5D*). In concordance with the above result, in vivo lactate dehydrogenase assay (*Abu-Shumays and Fristrom, 1997*) also revealed a high glycolytic index prevalent in the lymph glands of homozygous *whd*[1] (*Figure 5E–F*).

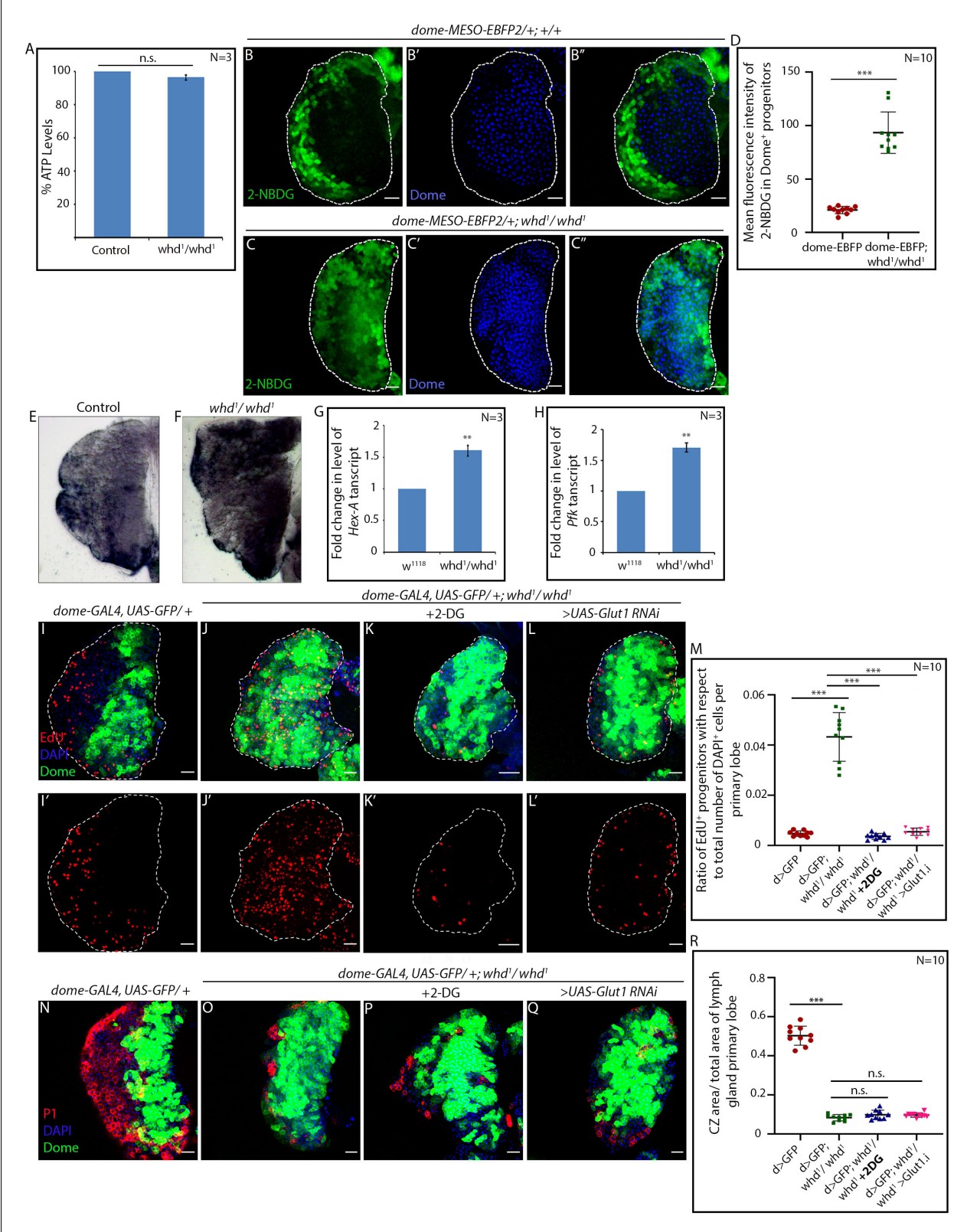

**Figure 5.** FAO loss in hemocyte progenitors led to sustained glycolysis. (**A**) ATP levels in control and $whd^1/whd^1$ whole larvae. p-Value of $whd^1/whd^1$ compared to control = $5.327 \times 10^{-2}$. (**B–D**) Glucose incorporation (marked by 2-NBDG uptake) levels in control *dome-MESO-EBFP2/+* (**B–B''**) and *dome-MESO-EBFP2/+; $whd^1/whd^1$* (**C–C''**) lymph glands. (**D**). Quantitative analysis of results from **B–C** demonstrating a significant increase in glucose uptake in the $whd^1/whd^1$ progenitors. p-Value for *dome-MESO-EBFP2/+; $whd^1/whd^1$* = $6.09 \times 10^{-7}$ compared to control. (**E–F**) Increased lactate

*Figure 5 continued on next page*

*Figure 5 continued*

dehydrogenase in-vivo enzymatic staining assay of $whd^1/whd^1$ lymph gland (**F**) compared to control (**E**). (**G**) Fold change in the level of *Hex-A* mRNA expression in control $w^{1118}$ and $whd^1/whd^1$ lymph glands. p-Value of $whd^1/whd^1 = 6.379 \times 10^{-3}$ compared to control. (**H**) Fold change in the level of *Pfk* mRNA expression in control $w^{1118}$ and $whd^1/whd^1$ lymph glands. p-Value of $whd^1/whd^1 = 3.739 \times 10^{-3}$ compared to control. (**I–M**) Proliferation status (marked by EdU) in control *dome > GFP* (**I–I'**), *dome > GFP; $whd^1/whd^1$* (**J–J'**), 2-DG fed *dome > GFP; $whd^1/whd^1$* (**K–K'**) and *dome > GFP; $whd^1/whd^1$; UAS-Glut1 RNAi* (**L–L'**) lymph glands. (**M**). Quantitative analysis of results from **I–L'**. p-Value for *dome > GFP; $whd^1/whd^1$* = $4.37 \times 10^{-7}$ compared to control and p-value for *dome > GFP; $whd^1/whd^1$* = $3.25 \times 10^{-7}$ fed with 2-DG compared to non-fed *dome > GFP; $whd^1/whd^1$*. p-Value for *dome > GFP; $whd^1/whd^1$; UAS-Glut1 RNAi* = $4.53 \times 10^{-7}$ compared to non-fed *dome > GFP; $whd^1/whd^1$*. (**M–P**) Comparison of differentiation (marked by P1) levels in control *dome > GFP* (**M**), *dome > GFP; $whd^1/whd^1$* (**N**) and 2-DG fed *dome > GFP; $whd^1/whd^1$* (**O**) lymph glands. (**P**). Quantitative analysis of results from **M–O** show decline in proliferation upon 2-DG feeding. p-Value for *dome > GFP; $whd^1/whd^1$* = $4.43 \times 10^{-11}$ compared to control and p-Value for *dome > GFP; $whd^1/whd^1$* = $8.6 \times 10^{-2}$ fed with 2-DG compared to non-fed *dome > GFP; $whd^1/whd^1$*. p-Value for *dome > GFP; $whd^1/whd^1$; UAS-Glut1 RNAi* = $5.9 \times 10^{-2}$ compared to non-fed *dome > GFP; $whd^1/whd^1$*. n.s. = not significant. Individual dots represent biological replicates. Values are mean ± SD, asterisks mark statistically significant differences (*p<0.05; **p<0.01; ***p<0.001, Student's *t*-test). Scale bar: 20 μm.

The online version of this article includes the following source data and figure supplement(s) for figure 5:

**Source data 1.** Contains numerical data plotted in *Figure 5A,D,G,H,M and R*.
**Figure supplement 1.** 2-NBDG assay in the lymph gland primary lobe.
**Figure supplement 1—source data 1.** Contains numerical data plotted in *Figure 5—figure supplement 1B*.

Moreover, in the $whd^1$ lymph gland, the transcript levels of HexA (Hexokinase A) and Pfk (Phos-phofructokinase), the enzymes involved in the two irreversible steps of glycolysis: exhibit a 1.6 fold and 1.7 fold increase in their expression, respectively (*Figure 5G–H*). Together these results reveal that upon FAO disruption, lymph gland progenitors adopt high-glucose utilization/metabolism.

Based on the above observations, we inferred that higher proliferation and differentiation defects observed in hemocyte progenitors with compromised FAO might be due to the surge in glucose uptake/metabolism. Upon rearing $whd^1$ homozygous larvae in food supplemented with glycolysis inhibitor 2-Deoxy-D-glucose (2-DG), the otherwise hyper-proliferating hemocyte progenitors (EdU, *Figure 5J*) demonstrate a significant drop in EdU incorporation to a level that is comparable to control (compare *Figure 5K* with *Figure 5I*, and quantified in *Figure 5M*). Although the glycolytic block by feeding 2-DG rescues the cell cycle status, the abrogated differentiation observed in homozygous $whd^1$ hemocyte progenitors is not restored (compare *Figure 5P* with *Figure 5O and N* and quantified in *Figure 5R*). Inhibition of glucose uptake by genetic perturbation of glucose transporter Glut1 in progenitor specific manner in the FAO mutant also endorses the above result (*Figure 5L–L' and M*, and *Figure 5Q–R*).

From these observations, it is evident that the surge in glucose metabolism encountered by the hemocyte progenitors upon FAO loss is responsible for their altered cell cycle. However, the glyco-lytic surge upon FAO disruption is unable to initiate progenitor differentiation. Collectively, the above results indicate that FAO plays a critical role in regulating the differentiation of hemocyte progenitors.

## FAO loss in progenitors causes an altered histone acetylation

Acetyl-CoA generated from FAO, apart from serving as a substrate for the Krebs cycle, is essential for the acetylation of various proteins, including histones. We, therefore, wondered that disruption of FAO in $whd^1$ hemocyte progenitors might also result in altered histone acetylations, which may, in turn, result in cell cycle and differentiation defects.

Histone acetylation mediated by Histone Acetyl Transferases (HATs) directly controls the expression of differentiation factor, thereby regulating germline stem cell differentiation (*McCarthy et al., 2018*; *Xin et al., 2013*). To ascertain whether HATs play a similar role in hemocyte progenitor differentiation, progenitor-specific RNAi-mediated knockdown of HATs function was done following the timeline, as shown in *Figure 2—figure supplement 1D*. Quite strikingly, loss of Histone Acetyl Transferase (HAT) genes, *Gcn5* (*Carré et al., 2005*) and *chm* (*chameau*) (*Grienenberger et al., 2002*; *Miotto et al., 2006*) phenocopy the differentiation defect seen in the hemocyte progenitors of FAO loss of function (*Figure 6A–C*). Additionally, downregulation of *Acetyl Coenzyme A syn-thase/AcCoAS* (the *Drosophila* orthologue of ACSS2) (*Mews et al., 2017*) results in a phenotype identical to HAT or FAO loss (*Figure 6D*) confirming the essential role of acetylation in hemocyte progenitor differentiation.

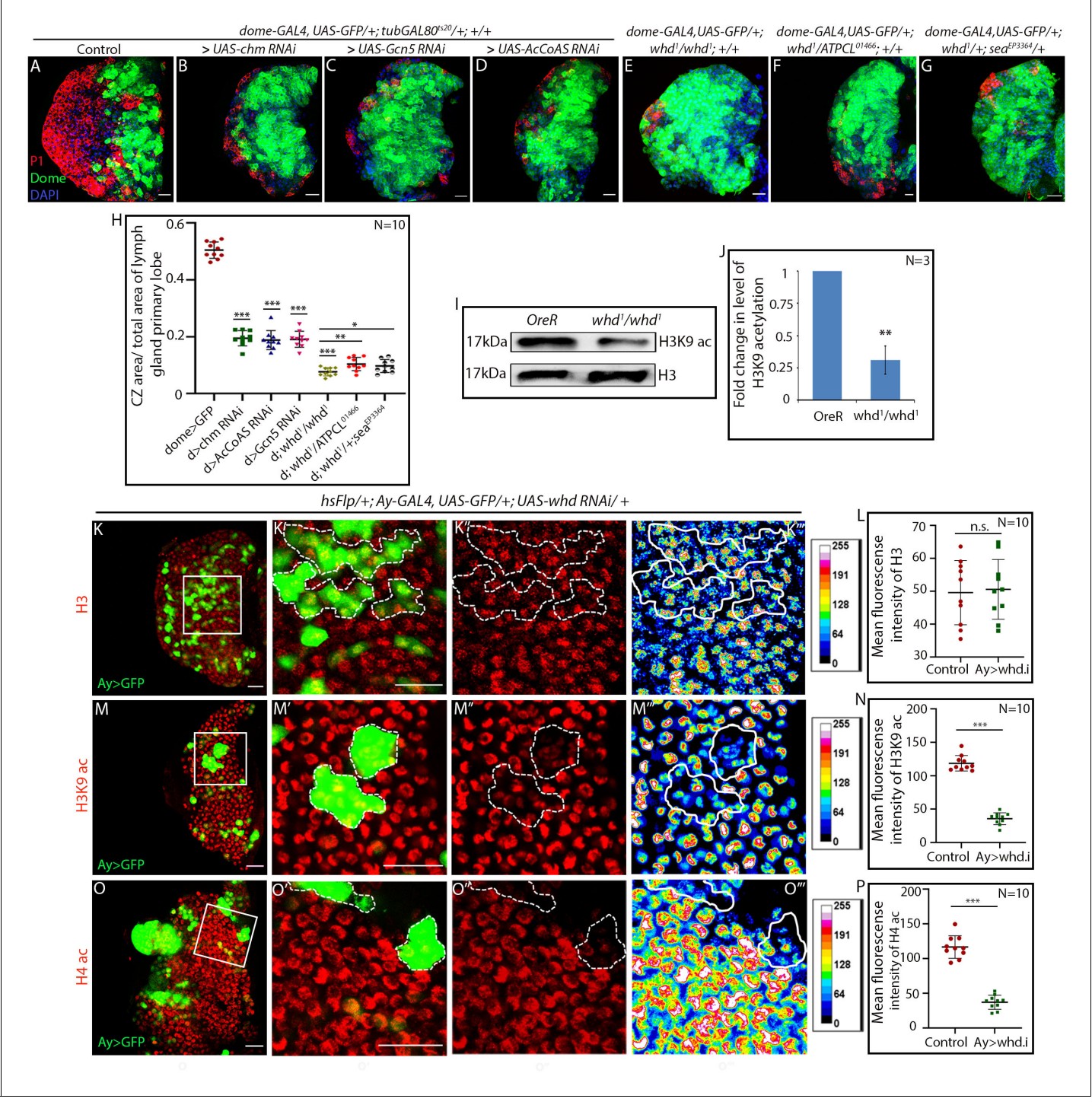

**Figure 6.** Hemocyte progenitors of HAT and FAO loss of function exhibits altered histone acetylation. (A–H) Comparison of differentiation (marked by P1) levels in *dome > GFP* lymph gland of control (A) with progenitor-specific downregulation of (B) *chm*, (C) *Gcn5*, (D) *AcCoAS* and (E) *whd¹/whd¹*, (F) transheterozygote of *whd* and *ATPCL* (*whd¹/ATPCL⁰¹⁴⁶⁶*) and transheterozygote of *whd* and *sea* (*whd¹/seaEP³³⁶⁴*) (G). (H) Quantitative analyses of the results from A–G. p-Value for *dome-GAL4, UAS-GFP; tubGAL80ᵗˢ²⁰ > UAS-chm RNAi = 2.267×10⁻¹⁵* compared to control. p-Value for *dome-GAL4, UAS-GFP; tubGAL80ᵗˢ²⁰ > UAS-Gcn5 RNAi = 1.990×10⁻¹⁴* compared to control. p-Value for *dome-GAL4, UAS-GFP; tubGAL80ᵗˢ²⁰ > UAS-AcCoAS RNAi = 2.601×10⁻¹⁵* compared to control. p-Value for *dome-GAL4, UAS-GFP; whd¹/whd¹ = 1.400×10⁻¹⁵* compared to control. p-Value for *dome-GAL4, UAS-GFP; whd¹/ATPCL⁰¹⁴⁶⁶ = 6.835×10⁻³* compared to control *dome-GAL4, UAS-GFP; whd¹/whd¹*. p-Value for *dome-GAL4, UAS-GFP; whd¹/ seaEP³³⁶⁴ = 2.974×10⁻²* compared to control *dome-GAL4, UAS-GFP; whd¹/whd¹*. (I–J) Western blot analysis of H3K9 acetylation level in control *OreR* and *whd¹/whd¹* larvae with H3 as a loading control (I). (J). Quantitative analysis of H3K9 acetylation level in I. p-Value for *whd¹/whd¹ = 8.056×10⁻³* compared to control *OreR*. (K–P) Clonal analysis of histone acetylation in the GFP-positive *hs-Flp/Ay-GAL4* based clonal patches (GFP indicates cells

*Figure 6 continued on next page*

*Figure 6 continued*

where the *whd* function is knocked down). Immunostaining with H3 (**K–L**), H3K9 acetylation (**M–N**), and H4 pan acetylation (**O–P**) antibodies. (**L**). Quantitative analyses of H3 acetylation level in **K–K'''**. p-Value for *hs-Flp/Ay-GAL4. UAS-GFP; UAS-whd RNAi* = $8.188 \times 10^{-1}$ compared to control. (**N**). Quantitative analysis of H3K9 acetylation level in (**M–M'''**). p-Value for *hsFlp/+; Ay-GAL4. UAS-GFP; UAS-whd RNAi* = $2.238 \times 10^{-12}$ compared to control. (**P**). Quantitative analysis of H4 acetylation level in **O–O'''**. p-Value for *hsFlp/Ay-GAL4. UAS-GFP, UAS-whd RNAi* = $1.083 \times 10^{-9}$ compared to control. Individual dots represent biological replicates. Values are mean ± SD, asterisks mark statistically significant differences (*p<0.05; **p<0.01; ***p<0.001, Student's *t*-test). Scale bar: 20 μm n.s. = not significant.

The online version of this article includes the following source data and figure supplement(s) for figure 6:

**Source data 1.** Contains numerical data plotted in *Figure 6H,J,L,N and P*.
**Figure supplement 1.** Hemocyte progenitors of *whd¹* loss of function exhibit altered histone acetylation.
**Figure supplement 1—source data 1.** Contains numerical data plotted in *Figure 6—figure supplement 1B,D,G,J and M*.

Next, we performed an epistatic interaction of *whd¹* allele with other major acetyl-CoA related genes. Citrate transporter SLC25A1 or *scheggia/sea* in *Drosophila* (*Carrisi et al., 2008*) exports Krebs cycle metabolite citrate from the mitochondria to the cytoplasm to generate acetyl-CoA. In the cytosol, citrate gets converted to acetyl-CoA by *ATP citrate lyase* (ACLY encoded by *Drosophila* orthologue, *ATPCL*). Trans-heterozygous loss-of-function allelic combinations of *whd¹* with either *ATPCL⁰¹⁴⁶⁶* (*Figure 6F*) or *seaᴱᴾ³³⁶⁴* (*Figure 6G*) phenocopy differentiation defects seen in *whd¹* homozygous lymph gland (*Figure 6E*). Above set of genetic correlations reveal that alteration in histone acetylation affects hemocyte progenitor differentiation (quantified in *Figure 6H*).

Knockdown of the rate-limiting enzyme of FAO:CPT1, leads to a reduced level of H3K9 acetylation in lymphatic endothelial cells (*Wong et al., 2017*). We wondered whether similar acetylation defect occurs in *whd¹* homozygous mutant larvae. Immunoblot analysis of extracted histones with antibody against acetylated anti-H3K9 reveals that *whd¹* homozygous larvae have low levels of H3K9 acetylation (*Figure 6I–J*) when compared to control. However, the level of histone H3 is comparable to that of control.

Whether the tissue of our interest also reflects this decline in acetylation level, H3K9 acetylation labeling was performed in mosaic clones using hsFlp/Ay-GAL4 mediating RNAi knockdown of *Drosophila* CPT1 orthologue *whd*. The clonal patches positively marked with GFP (where *whd* has been downregulated) show a significant drop in H3K9 acetylation levels (*Figure 6M–N*) compared to surrounding hemocyte progenitors. However, histone H3 labeling in both mutant and control clonal patches are comparable (*Figure 6K–L*) and serve as a control. This observation, along with the western blot analyses, reveals the occurrence of H3K9 acetylation defects in FAO loss of function. Likewise, histone H4 acetylation visualized by pan anti-H4 acetylation antibody reveals a drastic drop in *whd* knockdown clonal patches (*Figure 6O–P*). Both the expression of H3K9 and pan H4 acetylation remains unaltered in mock/wild type clones (*Figure 6—figure supplement 1A–D*). Further, upon progenitor specific downregulation of *whd* function, a decline in the level of both H3K9 acetylation (compare *Figure 6—figure supplement 1H–H'* with *Figure 6—figure supplement 1I–I' and J*) and pan H4 acetylation (*Figure 6—figure supplement 1K–K'* with *Figure 6—figure supplement 1L–L' and M*) is evident. In all these scenarios, histone H3 labeling remains unaffected (*Figure 6—figure supplement 1E–G*).

Above molecular and genetic analyses demonstrate that the downregulation of FAO in the hemocyte progenitors leads to a decline in histone acetylation. The next step was to correlate whether the differentiation defects of hemocyte progenitors in FAO loss of function is a consequence of altered histone acetylation levels.

## Acetate supplementation rescues differentiation defects of FAO mutant hemocyte progenitors

Histone acetylation in eukaryotes relies on acetyl-Coenzyme A (acetyl-CoA). It has been established earlier that compromised histone acetylation levels can be restored by supplementing acetate (*Carrisi et al., 2008*; *Wellen et al., 2009*; *Wong et al., 2017*). The supplemented acetate is converted to acetyl-CoA, which restores the endogenous histone acetylation in a cell. We wondered whether replenishing the H3K9 acetylation levels in *whd¹* by acetate supplementation (50 mM, supplemented fly food post first instar) can rescue the differentiation defect seen in those lymph glands.

Intriguingly, acetate feeding does not affect progenitor differentiation in control, whereas it rescues the differentiation defects seen in homozygous *whd¹* hemocyte progenitors (*Figure 7A–E*). At the molecular level, we observed that acetate supplementation to *whd¹* mutant larvae leads to a restoration of the H3K9 acetylation level (*Figure 7F–G*), which might lead to the rescue of the differentiation defect (*Figure 7A–E*). In order to probe this possibility, the lymph gland from acetate fed larvae were dissected and assayed for the status of H3K9 acetylation level. *Figure 7H–L* reveals that acetate supplementation indeed restores the compromised acetylation status in the *whd¹* lymph gland (compare *Figure 7J–J'* with *Figure 7K–K'*).

Conversely, upregulation of FAO by L-carnitine feeding leads to elevated H3K9 acetylation in the lymph gland (compare *Figure 7M–M'* with *Figure 7N–N'* and quantified in *Figure 7O*). Likewise, the level of H3K9 acetylation in L-carnitine fed larvae demonstrates a significant upregulation when compared to age-isogenised non-fed control larvae (*Figure 7P–Q*).

These results establish that the hemocyte progenitors require FAO mediated histone acetylation for their differentiation.

## JNK signaling regulates FAO in the hemocyte progenitors

Next, we attempted to understand how the FAO-mediated metabolic circuitry collaborates with the known differentiation signals of hemocyte progenitors. Jun-Kinase and dFOXO (Forkhead box O) mediated signal has been previously implicated for hemocyte progenitor differentiation (*Owusu-Ansah and Banerjee, 2009*). Analogous to *whd¹* lymph glands, expression of a dominant-negative allele of *basket* (*bsk*, *Drosophila* orthologue of Jun-Kinase) in the progenitors results in stalled differentiation (*Figure 8A–C*). On the other hand, overexpression of FOXO, pushes the progenitor fate towards precocious differentiation (*Owusu-Ansah and Banerjee, 2009*; *Figure 8D–E* and *Figure 8H*). However, genetic removal of one copy of *whd* is sufficient enough to prevent the precocious differentiation as observed in progenitor-specific overexpression of FOXO (*Figure 8F–H*). These results illustrate an unappreciated link between the differentiation signals and FAO in the lymph gland progenitors.

Next, we addressed whether JNK signaling regulates the expression of genes of FAO. The transcription of *CPT1/whd, Mcad, Mtpα, scully, Mtpβ,* and *yip2* was assayed upon down-regulation of *bsk* from hemocyte progenitors. The transcript level of *CPT1/whd* indicates a ~ 41% drop (*Figure 8I*) while the expression of rest of the enzymes either exhibited a mild drop (~18% in *yip2*) or no significant alteration upon loss of *bsk* from the progenitors. This observation established that JNK controls the transcription of *CPT1/whd,* thereby regulating FAO.

Interestingly, clonal analyses of *bsk/JNK* knockdown in hemocyte progenitors reveals a drop in the levels of H3K9 acetylation (*Figure 8L–M* and *Figure 8—figure supplement 1A–A'''*) and H4 pan acetylation (*Figure 8N–O* and *Figure 8—figure supplement 1B–B'''*) similar to *whd* downregulated clonal patches (*Figure 6M–P*). Additionally, expression of *bsk^DN* in progenitor-specific manner brings about a conspicuous downregulation of both H3K9 acetylation (*Figure 8—figure supplement 1F–H*) and H4 pan acetylation (*Figure 8—figure supplement 1I–K*). However, in the above experiments, histone H3 labeling remains unaffected (*Figure 8J–K* and *Figure 8—figure supplement 1C–E*).

Since compromised acetylation leads to differentiation defect in *whd¹*, which can be rescued upon acetate feeding, we wondered whether the block in differentiation upon JNK loss could be rescued similarly. Indeed, the hemocyte progenitors that lacked JNK (*dome >bsk^DN*) when reared in acetate supplemented food, demonstrate differentiation levels comparable to the similarly-aged control (*Figure 8P–T*).

Therefore, lack of histone acetylation encountered in JNK loss leads to the block in hemocyte progenitor differentiation. Moreover, *whd* transcription is under the regulation of JNK, which further endorses that JNK mediated regulation of FAO is crucial for differentiation.

If this is true, then the upregulation of FAO in JNK loss either by L-carnitine supplementation or by overexpression of *whd* should facilitate differentiation. Our results demonstrate that the upregulation of FAO in hemocyte progenitors that lacked JNK indeed elicits differentiation (*Figure 8U–Y* and *Figure 8—figure supplement 1L–P*).

Collectively, these results are in agreement with the fact that JNK regulates the differentiation of hemocyte progenitors by FAO-mediated histone acetylation.

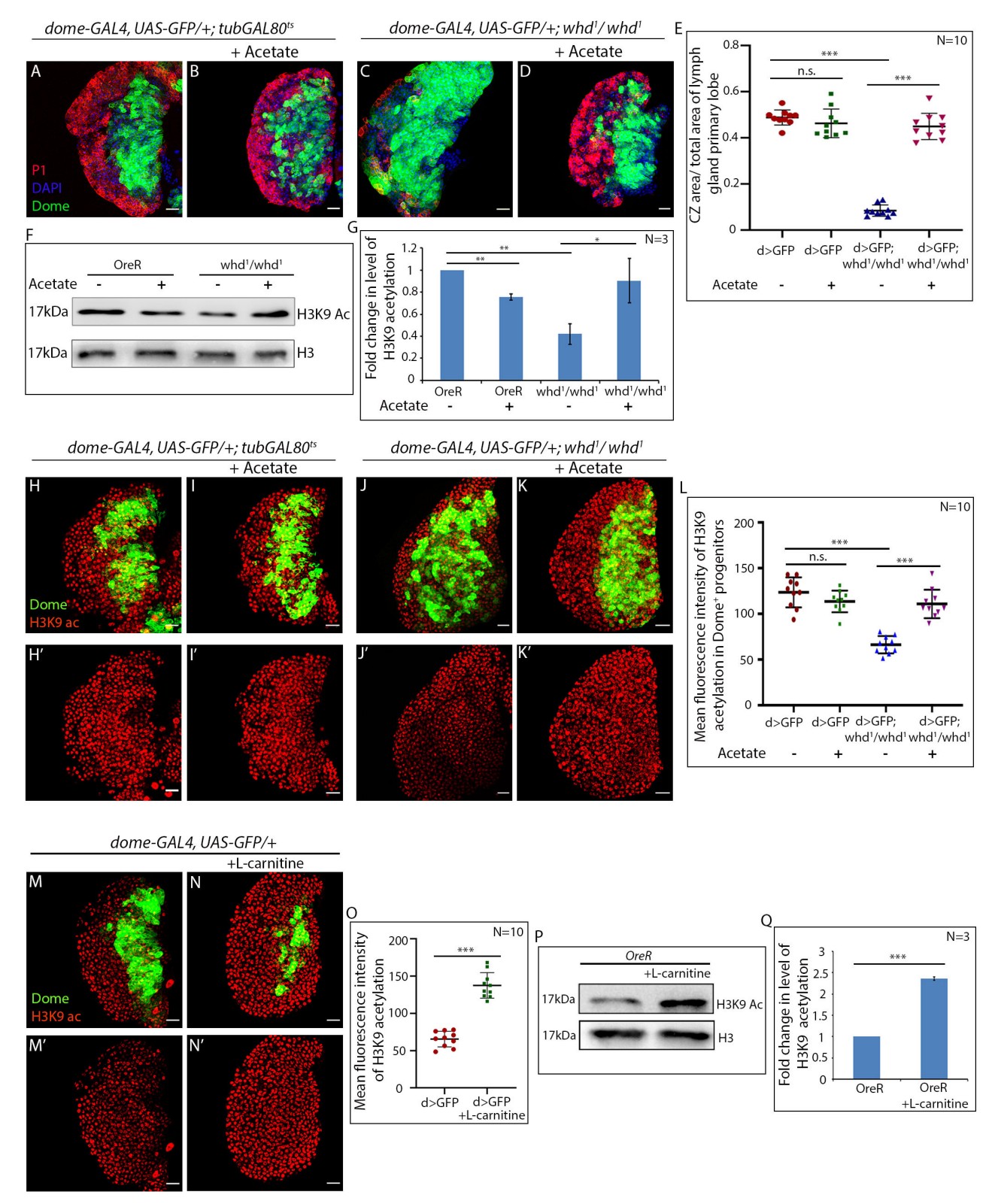

**Figure 7.** Acetate supplementation rescues differentiation defects of FAO mutant hemocyte progenitors. (A–E) Comparison of differentiation (marked by P1) levels in *dome > GFP* lymph gland of control (A) *dome > GFP* supplemented with acetate (B) *dome > GFP; whd¹/whd¹* (C) and *dome > GFP; whd¹/whd¹* supplemented with acetate (D). (E). Quantitative analysis of results from A–D. p-Value for *dome-GAL4, UAS-GFP; tubGAL80ᵗˢ²⁰ = 2.718×10⁻¹* fed with acetate compared to control *dome-GAL4, UAS-GFP; tubGAL80ᵗˢ²⁰*. p-Value for *dome-GAL4, UAS-GFP; whd¹/whd¹ = 3.18×10⁻¹⁶* compared to
*Figure 7 continued on next page*

*Figure 7 continued*

control *dome-GAL4, UAS-GFP; tubgal80^{ts20}*. p-Value for *dome-GAL4, UAS-GFP; whd^1/whd^1* = $2.576×10^{-10}$ fed with acetate compared to control *dome-GAL4, UAS-GFP; tubGAL80^{ts20}*. (F–G) Western blot analysis of H3K9 acetylation levels in control *OreR* and *whd^1/whd^1* larvae supplemented with acetate and non-fed controls with H3 as a loading control. (G) Quantitative analysis of H3K9 acetylation levels in F. p-Value for *OreR* = $4.589×10^{-3}$ supplemented with acetate compared to non-fed control *OreR*. p-Value for non-fed *whd^1/whd^1* = $8.001×10^{-3}$ compared to non-fed control *OreR*. p-Value for acetate supplemented *whd^1/whd^1* = $3.582×10^{-2}$ compared to non-fed control *whd^1/whd^1*. (H–L) Acetate supplementation restores H3K9 acetylation status in the *whd^1/whd^1* lymph gland (H-I'). (L) Quantitative analysis of acetylation level in control, *whd* mutant, and *whd* mutant fed on acetate. p-Value for acetate supplemented *dome-GAL4, UAS-GFP* = $1.38×10^{-1}$ compared to non-fed control. p-Value for *dome-GAL4, UAS-GFP; whd^1/whd^1* = $1.276×10^{-7}$ compared to *dome-GAL4, UAS-GFP*. p-Value for acetate supplemented *dome-GAL4, UAS-GFP; whd^1/whd^1* = $1.31×10^{-6}$ compared to non-fed control *dome-GAL4; UAS-GFP; whd^1/whd^1*. (M–O) Comparison of H3K9 acetylation level in Dome+ progenitors of L-carnitine fed larvae (N–N') with non-fed control (M–M'). (O) Quantitative analysis of H3K9 acetylation levels in M–N'. p-Value for *dome-GAL4, UAS-GFP* = $1.079×10^{-8}$ supplemented with L-carnitine compared to non-fed control *dome-GAL4, UAS-GFP*. (P–Q) Western blot analysis of H3K9 acetylation levels in *OreR* larvae supplemented with L-carnitine and non-fed controls with H3 as a loading control. Quantitative analysis of H3K9 acetylation levels in N. p-Value for *OreR* = $3.17×10^{-4}$ supplemented with L-carnitine compared to non-fed control *OreR*. Individual dots represent biological replicates. Values are mean ± SD, asterisks mark statistically significant differences (*p<0.05; **p<0.01; ***p<0.001, Student's *t*-test). Scale bar: 20 µm n.s. = not significant.

The online version of this article includes the following source data for figure 7:

**Source data 1.** Contains numerical data plotted in *Figure 7E,G,L,O and Q*.

## Discussion

Along with cellular signaling network, stem/progenitor cell fate and state are directly governed by their metabolism in normal development and during pathophysiological conditions (*Ito and Ito, 2016*; *Ito and Suda, 2014*; *Oginuma et al., 2017*; *Shyh-Chang et al., 2013*; *Shyh-Chang and Ng, 2017*). How the metabolic circuitry works in sync with the cell signaling machinery to achieve cellular homeostasis is yet to be fully understood. Here, we show the developmental requirement of FAO in regulating the differentiation of hemocyte progenitors in *Drosophila*.

Our molecular genetic analyses reveal a signaling cascade that links ROS-JNK-FAO and histone modification essential for the differentiation of hemocyte progenitors (*Figure 9*). High ROS levels in the progenitors evoke differentiation program by triggering JNK and FOXO mediated signals (*Owusu-Ansah and Banerjee, 2009*). We show that activated JNK, in turn, leads to transcriptional induction of *whd* to facilitate the import of fat moiety into mitochondria for β-oxidation. Optimal level of acetyl-CoA, the end product of FAO, is critical for the acetylation of several proteins, including histones. Altering this pathway either at the level of JNK or FAO affects histone acetylation in a HAT dependent manner. On the other hand, it is quite possible that precocious differentiation of blood progenitors in the lymph gland of starved larvae (*Shim et al., 2012*) might be an outcome of starvation induced fat mobilization and increased FAO.

JNK signaling has been associated with histone acetylation in different biological processes (*Miotto et al., 2006*; *Wu et al., 2008*). In *Drosophila*, Fos, a transcriptional activator of JNK, interacts with Chm (HAT) and causes modification of histones. Our investigation reveals that indeed upon downregulation of *JNK* (*Figure 8A–C*) and *Chm* (*Figure 6A–B*), differentiation of hemocyte progenitors is halted. Further, we show that halt in differentiation upon JNK loss is associated with alteration of the acetylation profile of H3K9 and H4. Given the fact that JNK signaling regulates FAO, which in turn provides the acetyl moiety for histone acetylation, our study provides a new dimension in JNK's role for histone acetylation in an FAO dependent manner. As a result, despite having high ROS levels, the hemocytes fail to differentiate if FAO is attenuated. Thus, the current work provides a metabolic link between JNK and epigenetic regulation of gene expression.

Our results show that upon disruption of FAO, hematopoietic progenitors adopt glycolysis to overcome the G2-M arrest but fail to initiate differentiation. Pharmacological and genetic inhibition of glycolysis in the FAO mutant restores their cell cycle defect but fails to facilitate their differentiation. The glycolytic surge in FAO mutants is not capable to take them through the differentiation process. This indicates that for the process of differentiation, the acetyl moiety derived from FAO plays a key role to facilitate hemocyte progenitor differentiation.

A recent study has demonstrated that alteration in acetyl-CoA levels can affect proteome and cellular metabolism by modulating intracellular crosstalk (*Dieterich et al., 2019*). It is intriguing to see that restoring acetylation level by the acetate supplementation is capable of rescuing hematopoietic

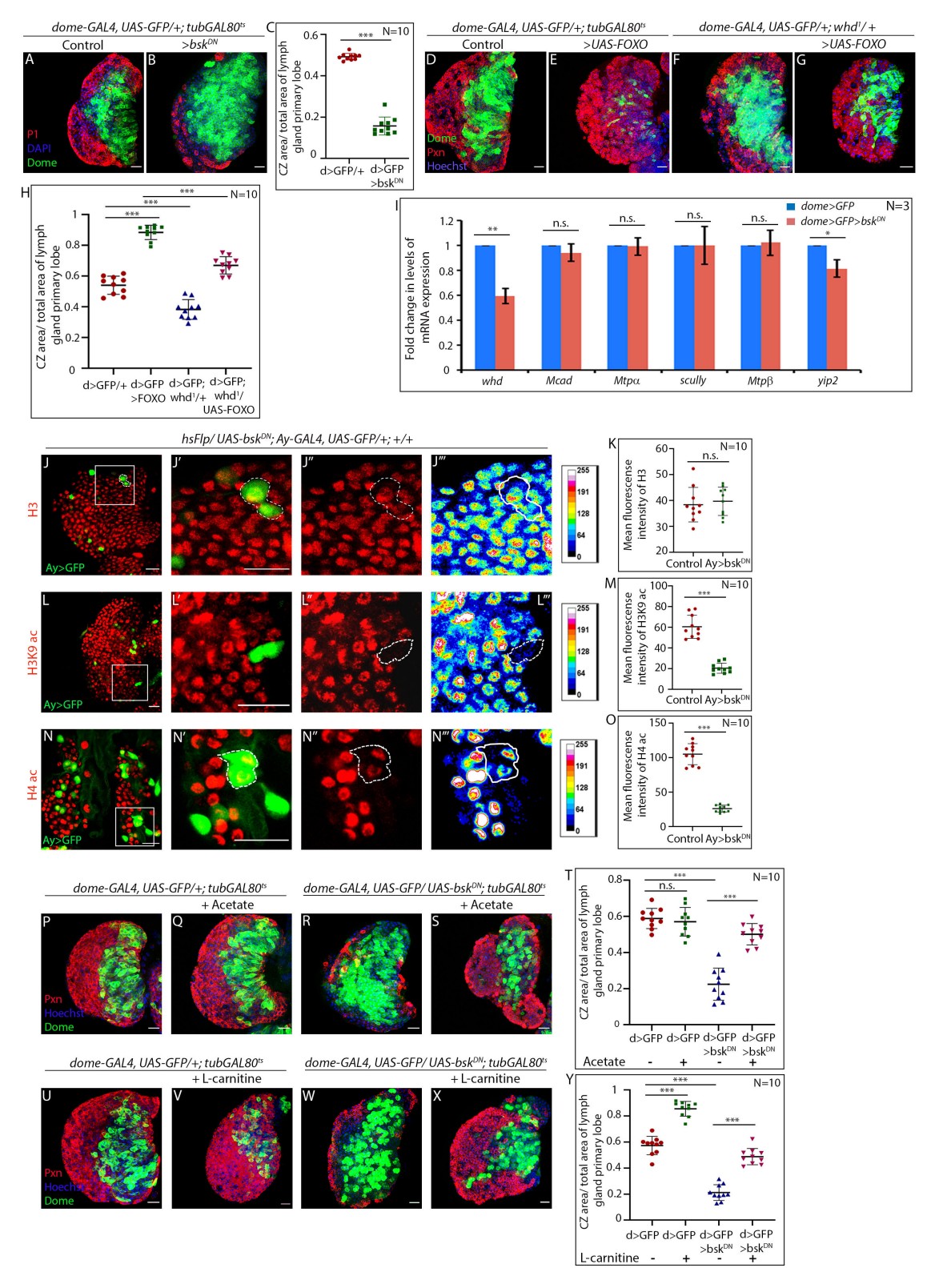

**Figure 8.** JNK regulates FAO in hemocyte progenitors of larval lymph gland. (**A–C**) Comparison of differentiation (marked by P1) levels in *dome > GFP* lymph gland of control (**A**), and *bsk/JNK* knockdown in hemocyte progenitors by *dome-GAL4, UAS-GFP; tubGAL80^{ts20} > UAS-bsk^{DN}* (**B**). (**C**). Quantitative analysis of the differentiation level from **A–B** reveals a significant increase in the Dome+ progenitor zone and a decrease in differentiation. p-Value for *dome-GAL4, UAS-GFP; tubGAL80^{ts20} > UAS-bsk^{DN}=5.84×10^{−11}* compared to control. (**D–H**) Differentiation levels (red, marked by Pxn) in

*Figure 8 continued on next page*

Figure 8 continued

overexpression of FOXO by *dome-GAL4, UAS-GFP; tubGAL80$^{ts20}$ > UAS-FOXO* (E) is significantly increased compared to control (D). The increased differentiation in FOXO overexpression background is significantly rescued by one copy of the null allele of *whd* (G). (F). The differentiation level in one copy null allele of *whd*. (H). Quantitative analysis of the differentiation level from D–G reveals a significant increment in Pxn+ differentiated cell area in FOXO overexpression from Dome+ progenitors, which is significantly rescued by one copy null allele of *whd*. p-Value for *dome-GAL4, UAS-GFP; tubGAL80$^{ts20}$ > UAS-FOXO* =5.77×10$^{-11}$ compared to control. p-Value for *dome-GAL4, UAS-GFP; whd$^1$/+* = 2.11×10$^{-5}$ compared to control. p-Value for *dome-GAL4, UAS-GFP; tubGAL80$^{ts20}$ > UAS-FOXO/whd$^1$* = 3.84×10$^{-9}$ compared to *dome-GAL4, UAS-GFP; tubGAL80$^{ts20}$ > UAS-FOXO*. (I) Real-time expression analysis of fatty acid oxidation enzymes, *whd, Mcad, Mtpα, scully, Mtpβ,* and *yip2* from *dome > GFP* and *dome > GFP > UAS-bsk$^{DN}$* lymph glands. The expression of *whd* shows a significant drop ~41% in *dome > GFP > UAS-bsk$^{DN}$* compared to control *dome > GFP*. p-Value for *whd* expression in *dome-GAL4, UAS-GFP; tubGAL80$^{ts20}$ > UAS-bsk$^{DN}$*=7.06×10$^{-3}$ compared to control. p-Value for *Mcad* expression in *dome-GAL4, UAS-GFP; tubGAL80$^{ts20}$ > UAS-bsk$^{DN}$*=6.71×10$^{-1}$ compared to control. p-Value for *Mtpα* expression in *dome-GAL4, UAS-GFP; tubGAL80$^{ts20}$ > UAS-bsk$^{DN}$*=8.95×10$^{-1}$ compared to control. p-Value for *scully* expression in *dome-GAL4, UAS-GFP; tubGAL80$^{ts20}$ > UAS-bsk$^{DN}$*=9.73×10$^{-1}$ compared to control. p-Value for *Mtpβ* expression in *dome-GAL4, UAS-GFP; tubGAL80$^{ts20}$ > UAS-bsk$^{DN}$*=7.7×10$^{-1}$ compared to control. p-Value for *yip2* expression in *dome-GAL4, UAS-GFP; tubGAL80$^{ts20}$ > UAS-bsk$^{DN}$*=2.42×10$^{-2}$ compared to control. (J–O) Clonal analysis of histone acetylation in GFP-positive *hsFlp/Ay-GAL4* based clonal patches expressing a dominant-negative form of *bsk* and immunostaining with H3 (J–J′′′), H3K9 acetylation (L–L′′′) and H4 pan acetylation (N–N′′′) antibodies. (K). Quantitative analysis of H3 acetylation level in J–J′′′. p-Value for *hsFlp/Ay-GAL4. UAS-GFP, UAS-bsk$^{DN}$* = 6.32×10$^{-1}$ compared to control. (M). Quantitative analysis of H3K9 acetylation level in L–L′′′. p-Value for *hsFlp/Ay-GAL4. UAS-GFP; UAS-bsk$^{DN}$* = 1.911×10$^{-7}$ compared to control. (O). Quantitative analysis of H4 acetylation level in N–N′′′. p-Value for *hs-Flp/Ay-GAL4. UAS-GFP, UAS-bsk$^{DN}$* = 8.22×10$^{-9}$ compared to control. (P–T) Stalled differentiation levels (red, marked by Pxn) in *dome-GAL4, UAS-GFP; tubGAL80$^{ts20}$ > UAS-bsk$^{DN}$* (R) is significantly rescued in larvae reared in fly food supplemented with acetate (S). The differentiation level in control (P) *dome-GAL4, UAS-GFP; tubGAL80$^{ts20}$* remain unaltered upon acetate feeding (Q). (T). Quantitative analysis of the differentiation level from P–S reveals a significant rescue of differentiated cells upon acetate supplementation in *dome-GAL4, UAS-GFP; tubGAL80$^{ts20}$ > UAS-bsk$^{DN}$* lymph glands. p-Value for acetate supplemented *dome-GAL4, UAS-GFP; tubGAL80$^{ts20}$* = 5.655×10$^{-1}$ compared to non-fed control. p-Value for *dome-GAL4, UAS-GFP; tubGAL80$^{ts20}$ > UAS-bsk$^{DN}$*=1.32×10$^{-8}$ compared to control *dome-GAL4, UAS-GFP; tubGAL80$^{ts20}$*. p-Value for acetate fed *dome-GAL4, UAS-GFP; tubGAL80$^{ts20}$ > UAS-bsk$^{DN}$*=4.73×10$^{-7}$ compared to non-fed *dome-GAL4, UAS-GFP; tubGAL80$^{ts20}$ > UAS bsk$^{DN}$*. (U–Y) The differentiation level (red, marked by Pxn) in control (U) *dome-GAL4, UAS-GFP; tubGAL80$^{ts20}$* increases upon L-carnitine feeding (V). Defect in differentiation levels (red, marked by Pxn) in *dome-GAL4, UAS-GFP; tubGAL80$^{ts20}$ > UAS bsk$^{DN}$* (W) is significantly rescued in larvae reared in fly food supplemented with L-carnitine (X). (Y). Quantitative analysis of the differentiation level from U–X, reveals a significant rescue of differentiated cells upon L-carnitine supplementation in *dome-GAL4, UAS-GFP; tubGAL80$^{ts20}$ > UAS-bsk$^{DN}$* lymph glands. p-Value for L-carnitine supplemented *dome-GAL4, UAS-GFP; tubGAL80$^{ts20}$* = 1.69×10$^{-8}$ compared to non-fed control. p-Value for *dome-GAL4, UAS-GFP; tubGAL80$^{ts20}$ > UAS-bsk$^{DN}$* =4.5×10$^{-10}$ compared to control *dome-GAL4, UAS-GFP; tubGAL80$^{ts20}$*. p-Value for L-carnitine fed *dome-GAL4, UAS-GFP; tubGAL80$^{ts20}$ > UAS-bsk$^{DN}$* =8.307×10$^{-9}$ compared to non-fed *dome-GAL4, UAS-GFP; tubGAL80$^{ts20}$ > UAS -bsk$^{DN}$*. Individual dots represent biological replicates. Values are mean ± SD, asterisks mark statistically significant differences (*p<0.05; **p<0.01; ***p<0.001, Student's *t*-test). Scale bar: 20 µm.

The online version of this article includes the following source data and figure supplement(s) for figure 8:

**Source data 1.** Contains numerical data plotted in *Figure 8C,H,I,K,M,O,T and Y*.

**Figure supplement 1.** JNK regulates FAO in hemocyte progenitors of the larval lymph gland.

**Figure supplement 1—source data 1.** Contains numerical data plotted in *Figure 8—figure supplement 1E,H,K and P*.

defects in the lymph gland progenitors of FAO mutants. The acetate supplemented is converted into the end product of FAO: acetyl-CoA, the metabolite that is essential for histone acetylation. The involvement of acetyl-CoA in facilitating differentiation is further evidenced when on genetically downregulating AcCoAs (the major enzyme in acetyl-CoA generation) from the progenitor leads to a halt in their differentiation. Our study thus establishes that for hemocyte progenitor differentiation, the metabolic process FAO involves its metabolite acetyl-CoA for epigenetic modification. Earlier studies in diverse model systems have demonstrated that compromised in vivo histone acetylation defects can be rescued by acetate supplementation (*Gao et al., 2016*; *Soliman et al., 2012*). A similar finding in *Drosophila* hematopoiesis signifies the relevance of acetate supplementation across taxa. In light of this study, it would be interesting to see whether metabolite supplementation of FAO can modulate pathophysiological scenarios like certain forms of cancer which rely on fat oxidation.

FAO has been implicated in HSCs maintenance downstream of the PML-Peroxisome proliferator-activated receptor delta (PPARδ) pathway (*Ito et al., 2012*). Mechanistically, the PML-PPARδ-FAO pathway regulates HSC maintenance by controling asymmetric division. In FAO inhibition, HCSs undergo symmetric divisions, which lead to exhaustion and depletion of the stem cell pool resulting in their differentiation (*Ito et al., 2012*). Another study in mice shows that upon short term starvation, there is a decline in the number of HSC (*Takakuwa et al., 2019*). Since the HSC maintenance is FAO dependent (*Ito et al., 2012*), a loss in number might be attributed to heightened fat oxidation

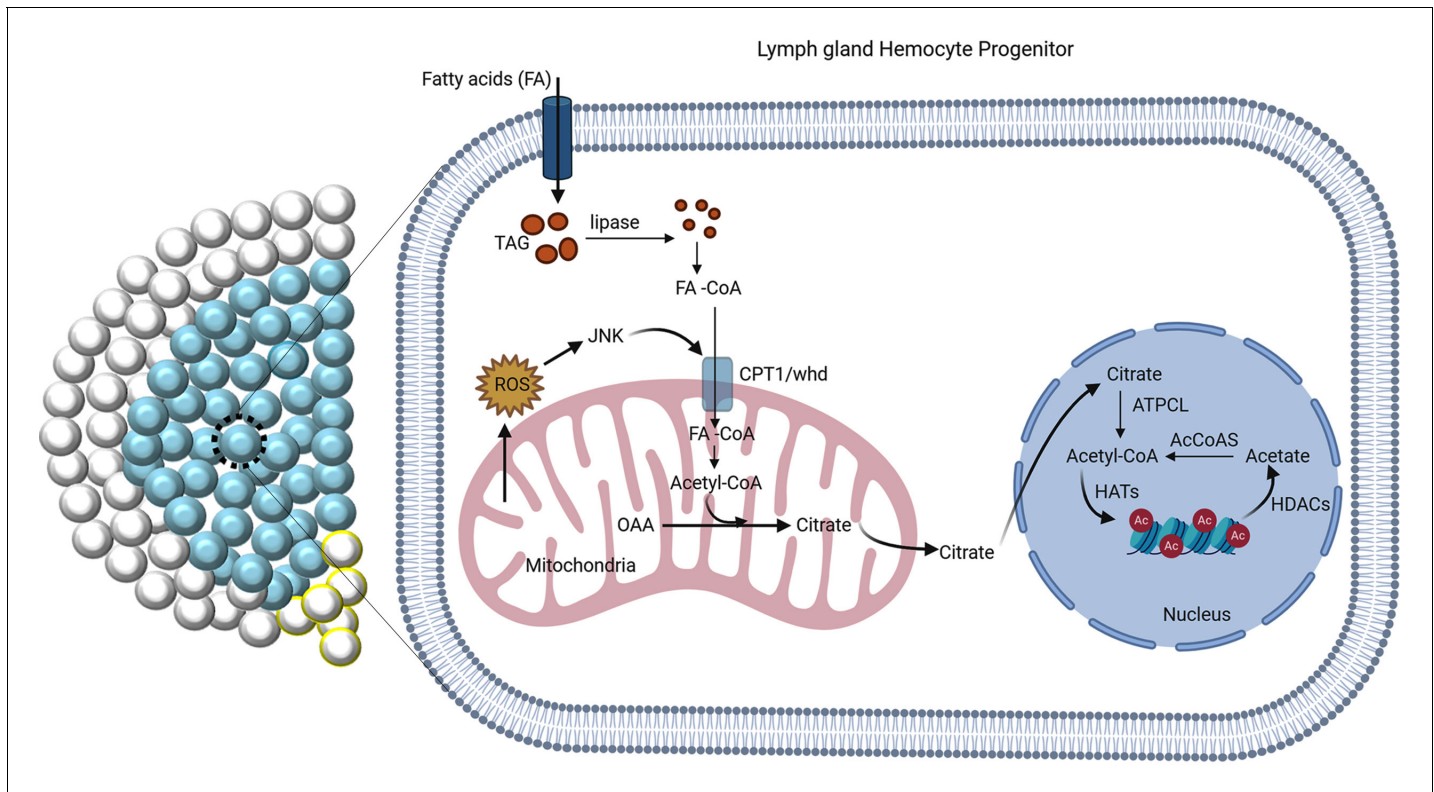

**Figure 9.** The regulation of FAO by JNK is critical for differentiation. ROS-JNK link has been previously shown to be essential for differentiation (*Owusu-Ansah and Banerjee, 2009*). The G2-M arrested hemocyte progenitors employ β-oxidation for their differentiation. ROS–JNK circuit impinges on FAO to facilitate progenitor differentiation. JNK signaling transcriptionally regulates *whd*, the rate-limiting enzyme of FAO leading to the production of acetyl-CoA. Acetyl-CoA leads to acetylation of histones in the hemocyte progenitors, which is critical for their differentiation.

during starvation. Interestingly, metabolic dependence on FAO has been reported in mammalian neural stem cell (*Knobloch et al., 2017*), muscle stem cells (*Ryall et al., 2015*) and intestinal stem cells (*Chen et al., 2020*). Although endothelial precursors (*Wong et al., 2017*) is also known to be dependent on FAO, it remains to be seen whether FAO is a preferred metabolic requirement for progenitor differentiation.

The entire blood cell repertoire in *Drosophila* is engaged in innate immunity, maintenance of tissue integrity, wound healing, and heterogeneous stress responses, and is therefore functionally considered to be similar to myeloid cells in mammals (*Banerjee et al., 2019*; *Gold and Brückner, 2014*). Interestingly, several molecular mechanisms that regulate *Drosophila* lymph gland hematopoiesis are essential players in progenitor-based hematopoiesis in vertebrates (*Banerjee et al., 2019*; *Gold and Brückner, 2014*; *Krzemien et al., 2010*).

Based on the above conservations, it is reasonable to propose the requirement of FAO in progenitor differentiation described here will help us in understanding mammalian myeloid progenitor differentiation.

## Materials and methods

**Key resources table**

| Reagent type (species) or resource | Designation | Source or reference | Identifiers | Additional information |
|---|---|---|---|---|
| Gene (*Drosophila melanogaster*) | dome | Flybase:FB2020_01 | FLYB:FBgn 0043903 | |

*Continued on next page*

*Continued*

| Reagent type (species) or resource | Designation | Source or reference | Identifiers | Additional information |
|---|---|---|---|---|
| Gene (*Drosophila melanogaster*) | Hml | Flybase:FB2020_01 | FLYB:FBgn 0029167 | |
| Gene (*Drosophila melanogaster*) | Tep4 | Flybase:FB2020_01 | FLYB:FBgn 0031888 | |
| Gene (*Drosophila melanogaster*) | CG3902 | Flybase:FB2020_01 | FLYB:FBgn 0036824 | |
| Gene (*Drosophila melanogaster*) | Mtpα | Flybase:FB2020_01 | FLYB:FBgn 0041180 | |
| Gene (*Drosophila melanogaster*) | Mtpβ | Flybase:FB2020_01 | FLYB:FBgn 0025352 | |
| Gene (*Drosophila melanogaster*) | whd | Flybase:FB2020_01 | FLYB:FBgn 0261862 | |
| Gene (*Drosophila melanogaster*) | Hnf4 | Flybase:FB2020_01 | FLYB:FBgn 0041180 | |
| Gene (*Drosophila melanogaster*) | chm | Flybase:FB2020_01 | FLYB:FBgn 0028387 | |
| Gene (*Drosophila melanogaster*) | Gcn5 | Flybase:FB2020_01 | FLYB:FBgn 0020388 | |
| Gene (*Drosophila melanogaster*) | AcCoAS | Flybase:FB2020_01 | FLYB:FBgn 0012034 | |
| Gene (*Drosophila melanogaster*) | Glut1 | Flybase:FB2020_01 | FLYB:FBgn 0264574 | |
| Gene (*Drosophila melanogaster*) | ATPCL | Flybase:FB2020_01 | FLYB:FBgn 0020236 | |
| Gene (*Drosophila melanogaster*) | sea | Flybase:FB2020_01 | FLYB:FBgn 0037912 | |
| Gene (*Drosophila melanogaster*) | bsk | Flybase:FB2020_01 | FLYB:FBgn 0000229 | |
| Genetic reagent (*Drosophila melanogaster*) | dome-GAL4 | Bloomington *Drosophila* Stock Center | BDSC:81010; FLYB:FBti0022298; RRID:BDSC_81010 | FlyBase symbol: P{GawB}dome[PG14] |
| Genetic reagent (*Drosophila melanogaster*) | Hml-dsRed.Δ | *Makhijani et al., 2011* | FLYB:FBgn 0041180 | FlyBase symbol: P{Hml-dsRed.Δ} |
| Genetic reagent (*Drosophila melanogaster*) | HmlΔ-GAL4 | *Sinenko and Mathey-Prevot, 2004* | FLYB: FBgn 0040877 | FlyBase symbol: P{Hml-GAL4.Δ} |
| Genetic reagent (*Drosophila melanogaster*) | Pvf2-lacZ | *Choi et al., 2008* | FLYB:FBtp0052107 | FlyBase symbol: P{Pvf2-lacZ.C} |
| Genetic reagent (*Drosophila melanogaster*) | TepIV-GAL4 | Kyoto Stock Center | DGGR:105442; FLYB:FBti0037434; RRID:DGGR_105442 | FlyBase symbol: P{GawB}NP7379 |

*Continued on next page*

*Continued*

| Reagent type (species) or resource | Designation | Source or reference | Identifiers | Additional information |
|---|---|---|---|---|
| Genetic reagent (*Drosophila melanogaster*) | CG3902-YFP | Kyoto Stock Center | DGGR:115356; FLYB:FBti0143519; RRID:DGGR_115356 | FlyBase symbol: PBac{566 .P.SVS-1} CG3902$^{CPTI100004}$ |
| Genetic reagent (*Drosophila melanogaster*) | Mtpα[KO] | Kyoto Stock Center | DGGR:116261; FLYB:FBal0267653; RRID:DGGR_116261 | FlyBase symbol: Mtpα$^{KO}$ |
| Genetic reagent (*Drosophila melanogaster*) | Mtpβ[KO] | Kyoto Stock Center | DGGR:116262; FLYB:FBal0267654; RRID:DGGR_116262 | FlyBase symbol: Mtpβ$^{KO}$ |
| Genetic reagent (*Drosophila melanogaster*) | UAS-whd RNAi [KK] | Vienna *Drosophila* RNAi Center | VDRC:v105400; FLYB:FBti0116709; RRID:FlyBase_FBst0477227 | FlyBase symbol: P{KK100935}VIE-260B |
| Genetic reagent (*Drosophila melanogaster*) | OreR | Bloomington *Drosophila* Stock Center | BDSC:5; FLYB:FBsn0000277; RRID:BDSC_5 | FlyBase symbol: Oregon-R-C |
| Genetic reagent (*Drosophila melanogaster*) | w[1118] | Bloomington *Drosophila* Stock Center | BDSC:3605; FLYB:FBal0018186; RRID:BDSC_3605 | FlyBase symbol: w$^{1118}$ |
| Genetic reagent (*Drosophila melanogaster*) | UAS-Hnf4.miRNA | Bloomington *Drosophila* Stock Center | BDSC:44398; FLYB:FBti0152533; RRID:BDSC_44398 | FlyBase symbol: P{UAS-Hnf4.miRNA}attP16 |
| Genetic reagent (*Drosophila melanogaster*) | UAS-whd RNAi | Bloomington *Drosophila* Stock Center | BDSC:34066; FLYB:FBal0263076; RRID:BDSC_34066 | FlyBase symbol: whd$^{HMS00040}$ |
| Genetic reagent (*Drosophila melanogaster*) | UAS-FOXO.P | Bloomington *Drosophila* Stock Center | BDSC:9575; FLYB:FBtp0017636; RRID:BDSC_9575 | FlyBase symbol: P{UAS-foxo.P} |
| Genetic reagent (*Drosophila melanogaster*) | Hnf4-GAL4 | Bloomington *Drosophila* Stock Center | BDSC:47618; FLYB:FBti0136396; RRID:BDSC_47618 | FlyBase symbol: P{GMR50A12-GAL4}attP2 |
| Genetic reagent (*Drosophila melanogaster*) | UAS-FUCCI | Bloomington *Drosophila* Stock Center | BDSC:55121; RRID:BDSC_55121 | FlyBase symbol: P{UAS-GFP.E2f1.1–230}32; P{UAS-mRFP1.NLS.CycB.1–266}19 |
| Genetic reagent (*Drosophila melanogaster*) | UAS-mito-HA-GFP | Bloomington *Drosophila* Stock Center | BDSC:8442; FLYB:FBti0040803; RRID:BDSC_8442 | FlyBase symbol: P{UAS-mito-HA-GFP.AP}2 |
| Genetic reagent (*Drosophila melanogaster*) | UAS-chm RNAi | Bloomington *Drosophila* Stock Center | BDSC:27027; FLYB:FBal0220716; RRID:BDSC_27027 | FlyBase symbol: chm$^{JF02348}$ |
| Genetic reagent (*Drosophila melanogaster*) | UAS-Gcn5 RNAi | Bloomington *Drosophila* Stock Center | BDSC:33981; FLYB:FBal0257611; RRID:BDSC_33981 | FlyBase symbol: Gcn5$^{HMS00941}$ |
| Genetic reagent (*Drosophila melanogaster*) | UAS-AcCoAS RNAi | Bloomington *Drosophila* Stock Center | BDSC:41917; FLYB:FBal0279313; RRID:BDSC_41917 | FlyBase symbol: AcCoAS$^{HMS02314}$ |
| Genetic reagent (*Drosophila melanogaster*) | UAS-Glut1RNAi | Bloomington *Drosophila* Stock Center | BDSC:28645; FLYB:FBal0239561; RRID:BDSC_28645 | FlyBase symbol: Glut1$^{JF03060}$ |
| Genetic reagent (*Drosophila melanogaster*) | ATPCL[01466] | Bloomington *Drosophila* Stock Center | BDSC:11055; FLYB:FBal0007976; RRID:BDSC_11055 | FlyBase symbol: ATPCL$^{01466}$ |
| Genetic reagent (*Drosophila melanogaster*) | sea[EP3364] | Bloomington *Drosophila* Stock Center | BDSC:17118; FLYB:FBal0131420; RRID:BDSC_17118 | FlyBase symbol: sea$^{EP3364}$ |

*Continued on next page*

*Continued*

| Reagent type (species) or resource | Designation | Source or reference | Identifiers | Additional information |
|---|---|---|---|---|
| Genetic reagent (*Drosophila melanogaster*) | UAS-bsk[DN] | Bloomington *Drosophila* Stock Center | BDSC:6409; FLYB:FBti0021048; RRID:BDSC_6409 | FlyBase symbol: P{UAS-bsk.DN}2 |
| Genetic reagent (*Drosophila melanogaster*) | UAS-mCD8::GFP | Bloomington *Drosophila* Stock Center | BDSC:5137; FLYB:FBti0180511; RRID:BDSC_5137 | FlyBase symbol: P{UAS-mCD8::GFP.L}2 |
| Genetic reagent (*Drosophila melanogaster*) | UAS-mCD8::RFP | Bloomington *Drosophila* Stock Center | BDSC:27400; FLYB:FBti0115747; RRID:BDSC_27400 | FlyBase symbol: P{UAS-mCD8.mRFP.LG}28a |
| Genetic reagent (*Drosophila melanogaster*) | U-6;sgRNA-whd-KO | Bloomington *Drosophila* Stock Center | BDSC:77066; FLYB:FBal0335953; RRID:BDSC_77066 | FlyBase symbol: whd[TKO.GS00854] |
| Genetic reagent (*Drosophila melanogaster*) | U-6;sgRNA-whd-OE | Bloomington *Drosophila* Stock Center | BDSC:68139; FLYB:FBal0337690; RRID:BDSC_68139 | FlyBase symbol: whd[TOE.GS00536] |
| Genetic reagent (*Drosophila melanogaster*) | whd[1] | Bloomington *Drosophila* Stock Center | BDSC:441; FLYB:FBal0018515; RRID:BDSC_441 | FlyBase symbol: whd[1] |
| Genetic reagent (*Drosophila melanogaster*) | Hnf4[Δ33] | Bloomington *Drosophila* Stock Center | BDSC:43634; FLYB:FBal0240651; RRID:BDSC_43634 | FlyBase symbol: Hnf4$^{Δ33}$ |
| Genetic reagent (*Drosophila melanogaster*) | Hnf4[Δ17] | Bloomington *Drosophila* Stock Center | BDSC:44218; FLYB:FBal0240650; RRID:BDSC_44218 | FlyBase symbol: Hnf4$^{Δ17}$ |
| Genetic reagent (*Drosophila melanogaster*) | tubGAL80[ts20] | Bloomington *Drosophila* Stock Center | BDSC:7109; FLYB:FBti0027796; RRID:BDSC_7109 | FlyBase symbol: P{tubP-GAL80[ts]}20 |
| Genetic reagent (*Drosophila melanogaster*) | hsFlp | Bloomington *Drosophila* Stock Center | BDSC:1929; FLYB:FBti0000784; RRID:BDSC_1929 | FlyBase symbol: P{hsFLP}12 |
| Genetic reagent (*Drosophila melanogaster*) | Ay-GAL4, UAS-GFP | Bloomington *Drosophila* Stock Center | BDSC:4411; FLYB:FBti0012290; FBti0003040 RRID:BDSC_4411 | FlyBase symbol: P{AyGAL4}25; P{UAS-GFP.S65T} Myo31DF[T2] |
| Antibody | anti-P1 (Mouse monoclonal) | *Kurucz et al., 2007* | Cat# NimC1, RRID:AB_2568423 | IF(1:50) |
| Antibody | anti-Pxn (Mouse) | *Nelson et al., 1994* | | IF(1:400) |
| Antibody | anti-proPO (Rabbit polyclonal) | *Jiang et al., 1997* | | IF(1:1000) |
| Antibody | anti-DE-cadherin (Rat polyclonal) | Developmental Studies Hybridoma Bank | Cat# DE-cad, RRID:AB_2314298 | IF(1:50) |
| Antibody | anti-Ci[155] (Rat polyclonal) | Developmental Studies Hybridoma Bank | Cat# 2A1, RRID:AB_2109711 | IF(1:2) |
| Antibody | anti-GFP (Rabbit polyclonal) | Invitrogen | Cat# A-11122, RRID:AB_221569 | IF(1:100) |
| Antibody | anti-H3 (Rabbit polyclonal) | Cell Signaling Technologies | Cat# 9927, RRID:AB_330200 | IF(1:400), WB(1:1000) |
| Antibody | anti-H3K9 acetylation (Rabbit polyclonal) | Cell Signaling Technologies | Cat# 9927, RRID:AB_330200 | IF(1:300), WB(1:1000) |
| Antibody | anti-H4 pan acetylation (Rabbit polyclonal) | Cell Signaling Technologies | Cat# 06–598, RRID:AB_2295074 | IF(1:500) |

*Continued on next page*

*Continued*

| Reagent type (species) or resource | Designation | Source or reference | Identifiers | Additional information |
|---|---|---|---|---|
| Chemical compound, drug | Sodium butyrate | EMD Millipore | 19–137 | |
| Chemical compound, drug | Nicotinamide | Sigma-Aldrich | 72345 | |
| Chemical compound, drug | Etomoxir | Cayman Chemicals | Cay11969 | 5 µM |
| Chemical compound, drug | Mildronate | Cayman Chemicals | Cay15997 | 100 µM |
| Chemical compound, drug | L-carnitine hydrochloride | Sigma-Aldrich | C0283 | 100 mM |
| Chemical compound, drug | 2-DG | Sigma-Aldrich | D8375 | 100 mM |
| Chemical compound, drug | Sodium acetate | Sigma-Aldrich | 71196 | 50 mM |
| Chemical compound, drug | 2-NBDG | Invitrogen | N13195 | 0.25 mM |
| Chemical compound, drug | LipidTOX | Molecular Probes | H34477 | 1:1000 |
| Chemical compound, drug | Streptavidin-Cy3 | Molecular Probes | SA1010 | 1:200 |
| Chemical compound, drug | Nile red | Molecular Probes | N1142 | 0.5 ug/mL |
| Chemical compound, drug | DHE (Dihydroxy Ethidium) | Molecular Probes | D11347 | 0.3 µM |
| Sequence-based reagent | Pfk_F | This paper | PCR primers | ATCGTATTTTGGCTTGCCGC |
| Sequence-based reagent | Pfk_R | This paper | PCR primers | CCAGAGAGATGACCACTGGC |
| Sequence-based reagent | Hex_F | This paper | PCR primers | CTGCTTCTAACGGACGAACAG |
| Sequence-based reagent | Hex_R | This paper | PCR primers | GCCTTGGGATGTGTATCCTTGG |
| Sequence-based reagent | whd_F | This paper | PCR primers | GGCCAATGTGATTTCCCTGC |
| Sequence-based reagent | whd_R | This paper | PCR primers | TGCCCTGAACCATGATAGGC |
| Sequence-based reagent | Act5C_F | This paper | PCR primers | ACACATTTTGTAAGATTTGGTGTGT |
| Sequence-based reagent | Act5C_R | This paper | PCR primers | CCGTTTGAGTTGTGCTGT |
| Sequence-based reagent | Mcad_F | This paper | PCR primers | GGCCTGGATCTCGATGTGTT |
| Sequence-based reagent | Mcad_R | This paper | PCR primers | GATCACAGGAGTTTGGCCCAG |
| Sequence-based reagent | Mtpα_F | This paper | PCR primers | ATCACTGTTGGTGACGGACC |
| Sequence-based reagent | Mtpα_R | This paper | PCR primers | CTGCAGCAGTCTGATGGCTT |
| Sequence-based reagent | scully_F | This paper | PCR primers | GATCAAGAACGCCGTTTCCC |
| Sequence-based reagent | scully_R | This paper | PCR primers | CAGATCGGCCAGGATCACG |

*Continued on next page*

*Continued*

| Reagent type (species) or resource | Designation | Source or reference | Identifiers | Additional information |
|---|---|---|---|---|
| Sequence-based reagent | Mtpβ_F | This paper | PCR primers | CAGGCACTCGCTTTTGTCAT |
| Sequence-based reagent | Mtpβ_R | This paper | PCR primers | CCTGGCAATGTTGGAGGTCT |
| Sequence-based reagent | yip2_F | This paper | PCR primers | TCTGCCGCAACCAAAGGTAT |
| Sequence-based reagent | yip2_R | This paper | PCR primers | TTAAGACCGGCAGCATCCAG |
| Software, algorithm | Fiji | Fiji | RRID:SCR_002285 | |
| Software, algorithm | Photoshop CC | Adobe | RRID:SCR_014199 | |
| Software, algorithm | Imaris | Bitplane | RRID:SCR_007370 | |
| Commercial assay or kit | Click-iT EdU plus (DNA replication kit) | Invitrogen | C10639 | |
| Commercial assay or kit | ATP bioluminescence kit HSII | Sigma | 11699709001 | |
| Commercial assay or kit | Histone extraction kit | Abcam | ab113476 | |
| Commercial assay or kit | RNAeasy Mini Kit | Qiagen | 74104 | |

## Fly stocks

The fly stocks used were *dome-GAL4, dome-MESO-EBFP2, Hml-DsRed* (K. Bruckner), *HmlΔ-GAL4* (S. Sinenko) *Pvf2-LacZ* (M. A. Yoo), *TepIV-GAL4, CG3902-YFP, Mtpα[KO], Mtpβ[KO]* (DGRC, Kyoto), *UAS-whd RNAi^{KK}* (VDRC, Vienna), *OreR, w^{1118}, UAS-Hnf4.miRNA* (**Lin et al., 2009**), *UAS-whd RNAi^{HMS00040}* (**Manzo et al., 2018**), *UAS-FOXO.P, Hnf4-GAL4^{GMR50A12}* (**Tokusumi et al., 2017**), *UAS-FUCCI, UAS-mito-HA-GFP, UAS-chm-RNAi^{JF02348}* (**Dietz et al., 2015**), *UAS-Gcn5-RNAi^{HMS00941}* (**Janssens et al., 2017**), *UAS-AcCoAS RNAi^{HMS02314}* (**Eisenberg et al., 2014**), *UAS-Glut1 RNAi^{JF03060}* (**Charlton-Perkins et al., 2017**), *ATPCL^{01466}, sea^{EP3364}, UAS-bsk^{DN}, UAS-mCD8::GFP, U-6;sgRNA-whd^{TKO.GS00854}, U-6;sgRNA-whd^{TOE.GS00536}, whd^{1}, Hnf4^{Δ33}, Hnf4^{Δ17}, tub-GAL80^{ts20}, hsFlp and Ay-GAL4, UAS-GFP* (BDSC, Bloomington *Drosophila* Stock Center).

Following genotypes were recombined for the current study:

1. *w; +/+; Hnf4-GAL4^{GMR50A12}, UAS-mCD8::GFP*
2. *dome-GAL4, UAS::mCD8RFP/FM7; +/+; +/+*
3. *dome-GAL4, UAS-GFP/FM7; whd^{1}/whd^{1}; +/+*
4. *w; U6:sgRNA-whd/U6:sgRNA-whd; UAS-dCas9/UAS-dCas9*
5. *dome-GAL4/FM7; UAS-FUCCI/Cyo; +/+*
6. *w; P{y[+t7.7] v[+t1.8]=TOE.GS00536}attP40/CyO; UAS-dCas9/UAS-dCas9*
7. *dome-MESO-EBFP2/+; whd^{1}/whd^{1}; +/+*
8. *hsFlp/hsFlp; Ay-GAL4, UAS-GFP/Ay-GAL4, UAS-GFP; +/+*
9. *UAS-bsk^{DN}/UAS bsk^{DN}; P{y[+t7.7] v[+t1.8]=TOE.GS00536}attP40/CyO; UAS-dCas9/UAS-dCas9*

All Stocks and crosses were maintained at 25°C, except for those used in RNAi based and *GAL4-UAS* expression experiments. In those cases, crosses were maintained at 29°C. For *GAL80^{ts}* experiments, crosses were initially maintained at 18°C for 5 days (equivalent to 60 hr at 25°C), and then shifted to 29°C till dissection (*Figure 2—figure supplement 1D*).

For synchronization of larvae, flies were allowed to lay eggs for 2 hr and newly hatched larvae within 1 hr interval were collected and transferred onto fresh food plates and aged for specified time periods at 25°C.

## Metabolic supplements and inhibitors

Fatty acid β-oxidation inhibitors: Etomoxir (Cayman Chemicals, Cay11969, inhibitor of CPT1) and Mildronate (Cayman Chemicals, Cay15997, inhibitor of carnitine biosynthesis and transport) were used at a concentration of 5 µM and 100 µM respectively mixed in fly food and fed to larvae from 48 hr AEH and analysis of lymph gland was done in late third instar stages. L-carnitine hydrochloride (Sigma-Aldrich, C0283) at a concentration of 100 mM has been used to augment FAO by allowing the entry of palmitic acid into the mitochondria. L-carnitine was used at 100 mM concentrations in fly food and fed to larvae for 48 hr in third instar analysis and for 24 hr in second instar analysis. Glycolytic inhibitor: 2-DG (2-Deoxy-D-Glucose (2-Deoxyglucose) (Sigma-Aldrich, D8375) used at a concentration of 100 mM mixed in fly food and fed to larvae for 48 hr in third instar analysis and for 24 hr in second instar analysis. Sodium acetate (Sigma-Aldrich, 71196) supplement was used at a concentration of 50 mM and fed to larvae from second instar 36 hr AEH onwards and analysis was done in late third instar stages. Similar aged larvae fed on vehicle controls served as control larvae. For all feeding experiments control larvae had same vehicle control level mixed in fly food. Fly food mixed with permissible food dye was fed to the control and experimental larvae and larvae with abundant food intake were picked for the experimental analysis.

## Clonal analysis using flp-out clone using Ay-GAL4 system

Generation of clones was done by the *Ay-GAL4* system that combines the technique of Flippase (Flp)/FRT system and the GAL4/UAS system (*Ito et al., 1997*). In this system, the Act5C promoter GAL4 fusion gene is interrupted by a FRT cassette containing yellow ($y^+$) gene. Heat shock treatment activates the *Flp* gene which in-turn excises the FRT cassette between the Act5C promoter and GAL4 sequence. This activates the expression of Act5C-GAL4 in cells. To induce *UAS-whd RNAi* and *UAS-bsk^DN* clones, mid second instar larvae of genotypes: *hsFlp; Ay-GAL4, UAS-GFP; UAS-whd RNAi* and *hsFlp/UAS-bsk^DN; Ay-GAL4, UAS-GFP* were subjected to heat shock for 90 min at 37°C, respectively. Post heat shock, larvae were transferred to 25°C to recover for 2 hr, then to express the respective knockdown constructs, larvae were reared at 29°C till dissection.

## Immunohistochemistry and imaging

The primary antibodies used in this study includes mouse anti-P1 (*Kurucz et al., 2007*), rabbit anti-Pxn (J. Fessler), rabbit anti-proPO (M. Kanost), rat anti-DE Cadherin (Cat# DE-cad (DE-cadherin), RRID:AB_2314298, 1:50, DSHB), rat anti-Ci[155] (Cat# 2A1, RRID:AB_2109711, 1:2, DSHB), rabbit anti-GFP (Cat# A-11122, RRID:AB_221569, 1:100, Invitrogen), rabbit anti-H3 (Cat# 9927, RRID:AB_330200, 1:400, Cell Signaling Technologies), rabbit anti-H3K9 acetylation (Cat# 9927, RRID:AB_330200, 1:300, Cell Signaling Technologies), rabbit anti-H4 pan acetylation (Cat# 06–598, RRID:AB_2295074, 1:500, Merck-Millipore). The following secondary antibodies mouse Cy3 (Cat# 115-165-166, RRID:AB_2338692), mouse FITC (Cat# 715-096-151, RRID:AB_2340796), rabbit Cy3 (Cat# 711-165-152, RRID:AB_2307443), rabbit FITC (Cat# 111-095-003, RRID:AB_2337972), rat Cy3 (Cat# 712-165-153, RRID:AB_2340667) from Jackson Immuno-research Laboratories were used at 1:400.

Lymph gland from synchronized larvae of required developmental age was dissected in cold PBS (1X Phosphate Buffer Saline, pH-7.2) and fixed in 4% Paraformaldehyde (PFA) for 45 min (*Mandal et al., 2007*) at room temperature (RT) on a shaker. Tissues were then permeabilized by 0.3% PBT (0.3% triton-X in 1X PBS) for 45 min (3 × 15 min washes) at RT. Blocking was then done in 10% NGS, for 30–45 min at RT. Tissues were next incubated in the respective primary antibody with appropriate dilution in 10% NGS overnight at 4°C. Post incubation in primary antibody, tissues were washed thrice in 0.3% PBT for 15 min each. This was followed by incubation of tissues in secondary antibody overnight at 4°C. The tissues were then subjected to four washes in 0.3% PBT for 15 min each, followed by incubation in DAPI solution (Invitrogen) for 1 hr at RT. Excess DAPI was washed off from the tissues by 1X PBS before mounting in Vectashield (Vector Laboratories).

## Immunohistochemical analysis of histone acetylation in lymph gland

Immunostaining for specific histone acetylations were performed with a slight modification of the above protocol. Lymph gland from synchronized larvae was dissected in ice cold PBS with deacetylate inhibitors (Sodium butyrate (10 mM, EMD Millipore, 19–137) and Nicotinamide (10 mM, Sigma-Aldrich, 72345) and fixed in 4% PFA prepared in ice-cold 1X PBS (pH 7.2) for 5 hr at 4°C. Tissue were

then permeabilized by 0.3% PBT for 45 min. Blocking was done with 5% BSA made in 1X PBS. Primary antibody and secondary antibody incubation solutions were made in 5% BSA in 1X PBS and subsequent washings were done with 0.1% PBT.

### Immunohistochemical analysis of DE-cadherin expression in lymph gland

To detect the DE-cadherin expression, lymph glands were incubated in DE-Cadherin antibody (1:50 in PBS) before fixation (*Langevin et al., 2005*) for 1 hr at 4°C. Tissues were then fixed in 4% PFA prepared in ice cold 1X PBS (pH 7.2) for 5 hr at 4°C. Then, tissues were washed thrice with 0.3% PBT for 30 min. Secondary antibody incubation, washes, and mounting were performed following the standard protocol (*Sharma et al., 2019*).

### Streptavidin-Cy3 labeling of mitochondria

Larvae were dissected in cold PBS followed by fixation in 4% PFA overnight at 4°C. This was followed by permeabilization with 0.1% PBT (0.1% triton-X in 1x PBS) for 45 min at RT and incubation in Streptavidin-Cy3 in 1:200 dilution (Molecular Probes, 434315) in 1XPBS for 1 hr at RT in dark. Post incubation samples were, washed thrice in PBS for 30 min. Lymph glands were then mounted in Vectashield and imaged in Zeiss LSM 780 confocal microscope.

### EdU labeling

Click-iT EdU plus (5- ethynyl-2'- deoxyuridine, a thymidine analog) kit (Invitrogen, C10639) plus was used to perform DNA replication assay. Lymph glands were dissected and incubated in EdU solution (1:1000 in PBS) for 40 min at RT for EdU incorporation. Next fixation was done in 4% PFA prepared in 1X PBS (pH 7.2) for 45 min at RT. Tissue were then permeabilized by 0.3% PBT (0.3% triton-X in 1X PBS) for 45 min at RT. Blocking was then done in 10% NGS, for 30–45 min at RT. To detect the incorporated EdU in cells, azide-based fluorophore were used as described in manufacturer protocol. EdU-labeled cell counting was done using spot detection function in Imaris Software.

### 2-NBDG assay

The protocol was slightly modified after (*Zou et al., 2005*). Larvae were dissected in ice-cold PBS and incubated in PBS with 0.25 mM 2-NBDG (Invitrogen, N13195) for 45 min at RT, washed twice in PBS for 5 min, fixed 45 min in 4% PFA and washed twice for 10 min in PBS. All washes and the fixation were done with ice-cold PBS (4°C). Lymph glands were speedily dissected and mounted in Vectashield and were imaged immediately with a Zeiss LSM 780 confocal microscope.

### Lactate dehydrogenase assay

Lactate dehydrogenase in vivo staining was modified from *Abu-Shumays and Fristrom, 1997*. Lymph glands of wandering third instar larvae were dissected in cold 1X PBS (pH8). Samples were fixed for 25 min in 0.5% glutaraldehyde in 1X PBS at room temperature, followed by four washes in 1X PBS for 15 min each. Staining was performed at 37°C in a solution of 0.1M NaPO$_4$ (pH 7.4), 0.5 mM lithium lactate, 2.75 mM NAD$^+$, 0.5 mg/ml NBT (Nitro blue tetrazolium) with 0.025 mg/ml PMS (Phenazine methosulfate). Reaction was stopped by washing in cold 1X PBS having pH 7.5. The samples were washed in four 1X PBS washes of 5 min each and immediately mounted and imaged.

### LipidTOX staining

Larvae were dissected in cold PBS followed by fixation in 4% PFA for 1 hr at RT, permeabilized by 0.1% PBT (0.1% triton-X in 1X PBS) for 45 min at RT. It was then incubated in 1X LipidTOX (diluted from 1000X stock provided by the manufacturer; Molecular Probes, H34477) in PBS for 1 hr at RT in dark, washed thrice in PBS for 30 min. Lymph glands were then mounted in Vectashield and imaged in Leica SP8 confocal microscope.

### Nile red staining

Larvae were dissected in cold PBS followed by fixation in 4% PFA for 1 hr at RT, permeabilized by 0.1% PBT (0.1% triton-X in 1X PBS) for 45 min at RT and incubated in 0.5 ug/mL Nile red (Molecular Probes, N1142) in PBS for 1 hr at RT in dark, washed thrice in PBS for 30 min. Lymph glands were mounted in Vectashield and imaged in Leica SP8 confocal microscope.

## Detection of ROS

Larvae were dissected in Schneider's medium (Gibco, 21720001) followed by incubation in 0.3 μM DHE (Molecular Probes, D11347) in Schneider's medium for 8 min at room temperature in dark. This was followed by two washes in 1X PBS for 5 min each; a brief fixation was done with 4% PFA for 10 min followed by two quick 1X PBS washes. Tissues were then mounted in Vectashield and imaged in Zeiss LSM 780 confocal microscope.

## Imaging and statistical analyses

Images were captured as confocal Z-stacks in Zeiss LSM 780, Leica SP8 confocal, and Olympus Flouview FV10i microscopes. Same confocal imaging settings were employed for image acquisition of control and experimental samples related to an experiment. Each experiment was repeated with appropriate controls at least three times to ensure reproducibility of the results. Data expressed as mean+/-Standard Deviation (SD) of values from three sets of independent experiments in GraphPad. Each dot in GraphPad represents a data point. Graphs plotted in EXCEL have Error Bars representing the Standard Deviation while graphs plotted in GraphPad employs Error Bars as mean+/-Standard Deviation. At least 10 images were analyzed per genotype, and statistical analyses performed employed two-tailed Student's t-test. Raw data related to statistical analysis are attached in the source file of each figure along with graphs plotted in excel.

p-Value of $<0.05; <0.01$ and $<0.001$, mentioned as *, **, *** respectively are considered as statistically significant while n.s. = not significant.

## Quantitative analysis of differentiation index in lymph gland

To measure the differentiation index of the primary lobe of lymph glands, middle confocal Z-stacks of a lymph gland image covering the Medullary Zone (MZ) were merged into a single section using ImageJ/Fiji (NIH) software as previously described (Shim et al., 2012). The merged section reflects the differentiated cell and hemocyte progenitor area clearly. For images with more than one fluorophore channel, each channel was separately analyzed. To measure differentiation index, P1-positive area was recalibrated into an identical threshold by using the Binary tool (Process–Binary–Make binary, Image J). Wand tool was used to capture the area with identical threshold whereas the size was measured using the Measure tool (Analyse–Measure). To measure the total area of one primary lobe of lymph gland, recalibration of the total area was then done by the Threshold tool until it was overlaid with identical threshold colour. Wand tool was used for selecting the total area for measurement. Differentiation index/fraction was estimated by dividing the size of the P1/Pxn positive area by the total size of the primary lobe. At least 10 lymph glands were analyzed per genotype, and two-tailed Student's t-test was done to evaluate the statistical significance.

## Quantification of the number of EdU⁺, FUCCI⁺ and progenitor subpopulations

Counting the number of EdU⁺ and FUCCI⁺ progenitors in lymph glands was done as described earlier (Sharma et al., 2019), using spot detection and surface tool in Imaris software and normalized by total number of nuclei per primary lobe of lymph gland. Different progenitor sub-populations in lymph glands were counted using the surface and spot detection function in Imaris as illustrated in detail (Sharma et al., 2019) and normalized by total number of cells (nuclei) in primary lobe and represented as percentage of progenitors in each primary lobe. Using surface tool, surface is created over Dome⁺ progenitors and nuclear label channel is masked in Dome⁺ surface. By utilizing the spot detection tool, the number of nuclei is counted in Dome⁺ surface. Similarly, number of nuclei (DAPI/ Hoechst) is counted in another surface created over Pxn⁺ cells. Next, in Dome⁺ surface, Pxn⁺ channel is masked. A surface is created over Dome⁺ Pxn⁺ region and nuclear channel is masked in this surface. Using spot detection tool, number of Dome⁺ Pxn⁺ IP nuclei are counted and its percentage can be calculated from the total number of nuclei in the primary lobe of lymph gland.

## ATP assay

ATP assay was performed with three biological replicates from late third instar larvae. Whole larvae were homogenized in ATP assay Lysis buffer (Costa et al., 2013). The samples were boiled at 95°C for 5 min and diluted 1:100 in dilution buffer provided in ATP bioluminescence kit HSII (Sigma,

11699709001). Further assay was performed as per manufacturer protocol in Glomax 96 microwell Luminometer (Promega). Standard curve was generated and ATP concentrations were calculated. The ATP concentration was normalized with protein concentration and expressed in percentage to plot the graph in EXCEL.

## Histone extraction and detection by western blotting

Histone from late third instar larvae were extracted using Histone extraction kit (Abcam, ab113476) following manufacturer protocol and quantitated by Bradford reagent (Biorad, 5000006). Equal amount of protein of each genotype was run on 4–12% SDS-PAGE and transferred to PVDF membrane (Millipore, IPVH00010). Blots were developed using Luminata Crescendo Western HRP substrate (Millipore, WBLUR0500) in LAS2000 blot imaging instrument. Primary antibodies used rabbit anti-H3 (Cat# 9927, RRID:AB_330200, 1:1000, Cell Signaling Technologies), rabbit anti-H3K9 acetylation (Cat# 9927, RRID:AB_330200, 1:1000, Cell Signaling Technologies). Secondary antibody rabbit anti-IgG-HRP (Cat# A00098-1 mg, RRID:AB_1968815, 1:5000, GenScript) was used. The band intensity was measured in Image J and normalized with histone H3 as loading controls. Analysis was done using three biological replicates.

## Quantitative RT-PCR

Extraction of RNA from lymph gland was performed from late third instar larvae of each genotype using TRIzol (Invitrogen, 15596018) followed by RNAeasy Mini Kit (Qiagen, 74104) according to the manufacturer's instructions. cDNA was prepared using the Verso cDNA Synthesis Kit (Thermo Scientific, AB1453B). To quantitate transcripts, qPCR was done using iTaq Universal SYBR Green Supermix (Biorad, 1725124) on a CFX96 Real-Time system/C1000 thermal Cycler (Biorad). *Drosophila* Actin5C was used as internal control.

Analysis was done using at least three biological replicates.

# Acknowledgements

We thank I Ando, U Banerjee and M Kanost for reagents. We thank all members of the two labs for their valuable inputs. Thanks to Parvathy Ramesh for her help in imaging with Olympus Flouview FV10i. We thank IISER Mohali's Confocal Facility, Bloomington *Drosophila* Stock Center, at Indiana University, DGRC (Kyoto), VDRC (Vienna) and Developmental Studies Hybridoma Bank, University of Iowa for flies and antibodies. Models 'Created with BioRender.com'. DBT/Wellcome-Trust India Alliance Senior Fellowship [IA/S/17/1/503100] to LM and Institutional support to SM and CSIR funding to SKT for this study duly acknowledged.

# Additional information

## Funding

| Funder | Grant reference number | Author |
| --- | --- | --- |
| Wellcome Trust/DBT India Alliance | IA/S/17/1/503100 | Lolitika Mandal |
| CSIR | | Satish Kumar Tiwari |
| Indian Institute of Science Education and Research Mohali | | Lolitika Mandal Sudip Mandal |

The funders had no role in study design, data collection and interpretation, or the decision to submit the work for publication.

## Author contributions

Satish Kumar Tiwari, Conceptualization, Data curation, Formal analysis, Validation, Investigation, Visualization, Methodology, Writing - original draft, Writing - review and editing; Ashish Ganeshlalji Toshniwal, Formal analysis, Validation, Investigation, Visualization; Sudip Mandal, Formal analysis, Methodology, Writing - review and editing; Lolitika Mandal, Conceptualization, Formal analysis,

Supervision, Funding acquisition, Validation, Investigation, Visualization, Methodology, Writing - original draft, Project administration, Writing - review and editing

### Author ORCIDs
Satish Kumar Tiwari [iD] https://orcid.org/0000-0002-1827-4603
Ashish Ganeshlalji Toshniwal [iD] https://orcid.org/0000-0001-8957-7970
Sudip Mandal [iD] https://orcid.org/0000-0002-2211-483X
Lolitika Mandal [iD] https://orcid.org/0000-0002-7711-6090

### Decision letter and Author response
Decision letter https://doi.org/10.7554/eLife.53247.sa1
Author response https://doi.org/10.7554/eLife.53247.sa2

## Additional files

### Supplementary files
- Transparent reporting form

### Data availability
All data generated or analyzed during this study are included in the manuscript and supporting files. Source data files have been provided for all Figures (that includes GraphPad or excel representations of the quantitative analyses).

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
