## [Decision Letter]

**Acceptance summary:**

In this manuscript, Mandal and colleagues explore the function of FAO (fatty acid oxidation) in differentiation of hematopoietic progenitors in *Drosophila* larval hematopoietic organ, lymph gland (LG). They show that supplementation with acetate rescues the phenotype associated with defective FAO, and that FAO is downstream of JNK, a known regulator of differentiation, providing insights into how metabolic pathway may be intertwined with differentiation pathway.

**Decision letter after peer review:**

[Editors’ note: the authors submitted for reconsideration following the decision after peer review. What follows is the decision letter after the first round of review.]

Thank you for submitting your work entitled "Fatty acid β-oxidation regulates hemocyte progenitor homeostasis in *Drosophila* larval lymph gland" for consideration by *eLife*. Your article has been reviewed by three peer reviewers, and the evaluation has been overseen by a Reviewing Editor and a Senior Editor. The following individual involved in review of your submission has agreed to reveal their identity: Yukiko M Yamashita (Reviewer #1).

Our decision has been reached after consultation between the reviewers. Based on these discussions and the individual reviews below, we regret to inform you that your work will not be considered for publication in *eLife*. Should they be able to address all substantive comments and choose to consider *eLife*, a fresh submission can be examined, without any guarantees.

Our consultation made every effort to understand the important assertions made and whether they were borne by the data presented. While the reviewers started with different perspectives, we have converged to a common position. We suggest that the authors go through our preamble, which captures this converged view, and also the specific concerns of each reviewer. While some of our major concerns can be addressed by substantive re-writing, cutting down generalisations and over-interpretation others require experimental attention.

One important issue, which was reiterated in our consultations, is exemplified by the following statement and related data "We show that the self-renewing hemocyte progenitors prefer glycolysis while the quiescent progenitors adopt Fatty acid β-oxidation (FAO) for their differentiation. Perturbation of β-oxidation in progenitors results in loss of quiescence…." Though the authors' primary question is centered on distinguishing metabolic states within progenitors, the authors do not identify these. Generic results are interpreted as applying to specific populations and extended to homeostasis.

Staging larvae and using appropriate established Gal4 drivers is, of course, acceptable, but the authors have not done this properly. Unlike, for example, the wing and eye discs, the larval LG do not have a uniform, reproducible structure or pattern. One cannot delineate zones morphologically or by using a single marker. Progenitor sub-populations (dome negative pre-progenitors, dome^+^ progenitors and dome^+^Pxn^+^hml^+^ intermediate progenitors) (reviewed in Banerjee et al., Flybook2019) and the three types of differentiated cells (plasmatocytes, crystal cells and lamellocytes) are functionally and metabolically different. However, the authors do not distinguish between these populations in their analysis. For example, mitochondrial analysis is shown at low resolution and in an undefined population (Figure 1L, M) and hml>GFP is used only as a marker of differentiation (Figure 4).

The assertion of the manuscript is that they are distinguishing between progenitor types, which they call self-renewing and quiescent. This can be done only upon identifying and characterising subpopulations. Further, since they incorrectly use the presence or absence of proliferation marks as a readout for self-renewal and quiescence, it is not clear how the claim that these are two metabolically distinct precursor states is valid. Rather, based on the analysis done (EdU, Figure 1B,C) they are simply proliferating or non-proliferating cells (which may or may not be progenitors), the latter poised for, or in the process of differentiation. Expectedly, their metabolic states differ- and the authors show that these are glycolysis for the former and FAO for the latter. Figure 1B,C: TepIV and EdU labelling – it looks as if there are co-labeled cells, but these are ignored. Also TepIVGAL4 reporter does not label the entire progenitor pool. The existence of self-renewing or quiescent progenitors has not been formally demonstrated in the LG- if the authors had demonstrated their existence, this would have been very interesting indeed.

The metabolic fluctuations we are concerned about are intra-LG. This is a major challenge in characterizing metabolic states of heterogeneous progenitors. Since all analysis is done only with one progenitor marker at a time (mainly dome) which is also expressed in early stages of differentiation, or with no marker at all, (Example Figure 1 ; Figure 1—figure supplement 1, compare panels A and C) claims regarding metabolic differences are not substantiated. As is also evident from the images in Figure 1, the same tissue from the same genotype and stage varies greatly in terms of size, shape and organisation, hence rigorous quantitation with additional markers is essential.

The authors seem confused about what the focus of their study is. They state in the Abstract that the major outcome of the study is coupling FAO with epigenetic modifications linking to regulation of differentiation by JNK- while this part of the study has merit, it is not very relevant to the primary question they pose about blood cell progenitor homeostasis.

The reviews give a very detailed analysis, including comments for almost every panel of Figure 1. However, some examples are expanded on a couple of specific cases below.

Figure 1D, E states that self-renewing early stage progenitors exhibit elevated levels of glucose uptake compared to late-stage quiescent progenitors. In D there is no marker for progenitors or differentiated cells- clearly there are varying intensities of signal. How can one assume that all 2-NBDG cells are progenitors, early or late? Further, given variation between preps, how does one compare levels between second and third instar without having a control to normalise it or assess relative levels between progenitors and differentiated cells? Assuming second and third instar preps were stained and imaged in the same experiment, how much variation is seen? A graph representing this is essential as the lymph gland morphology can vary greatly depending on the status of the larval cultures. Additionally, a proliferation assay is required along with the uptake or demarcation of the MZ/CZ boundaries to show reduced uptake in the pro-hemocyte pool of the 3rd instar lymph gland.

Figure 1F, G – To say that there is high Glut-1 GFP in proliferating progenitors of the second instar, labelling with a proliferation marker, progenitor marker, and differentiation marker is required along with quantitation to show that High Glut 1-GFP is expressed in proliferating progenitors.

Figure 3E-G, the authors claim a higher number of EdU incorporation is seen in the progenitors, however, the images provided show higher EdU signal at the periphery; also CZ/MZ boundary is not demarcated, and progenitor markers are not used.

If the results about FAO and its downstream effects in the LG are de-linked from progenitor sub-types and homeostasis, this would be more acceptable. Currently, two parts of the manuscript (progenitor types and metabolic status connecting to signaling) are not well connected. The authors may want to consider that they are perhaps best dealt separately and not in the same manuscript.

Reviewer #1:

This study by Mandal and colleagues studied the role of metabolic circuitry in stem cell maintenance/differentiation. General impacts of metabolism on cell fates, particularly those between stem cells and their progenies, have increasingly drawn researchers' attention in recent years. By using *Drosophila* hematopoietic progenitors (in lymph gland), the authors explored the role of metabolic circuitry in this lineage. The main conclusion is that actively proliferating stem cells utilize glycolytic pathway whereas quiescent stem cells favor fatty acid β-oxidation, which promotes differentiation.

Furthermore, they propose that FAO generates acetyl-CoA, which not only generates energy through TCA cycle, but can contribute to protein (e.g. histone) acetylation. Perturbation of FAO leads to defects in quiescence (progenitor starts proliferating and cannot differentiate). Upregulation of FAO leads to stem cell quiescence and precocious differentiation. The authors link this observation to histone acetylation and thus epigenetic regulation.

Figure 1: Does blocking glycolysis have any impact on differentiation? Does it only affect progenitor proliferation without impacting later differentiation?

Figure 2: blocking FAO by means of whd knockdown resulted in increased progenitors and decreased differentiating cells. Further in Figure 3, they show that loss of FAO also increases progenitor proliferation and thus progenitors are not entering quiescence. Collectively these results indicate that FAO is required for progenitors to enter quiescence, and thus their “differentiation defect” is secondary to their inability to enter quiescence. These results are quite clear, but when I was reading the description for Figure 2, I was misled to think that FAO blocks differentiation. Any ways to make this transition (Figure 2 to Figure 3) a bit less misleading?

In Figure 4, they show that upregulation of FAO is sufficient to induce quiescence and subsequent differentiation in progenitors.

Figure 5 demonstrate the inter-relationship between glycolysis and FAO. Defective FAO led to sustained glycolysis.

Figure 6-7 slightly shift in the focus and now addresses FAO's function in histone acetylation. The authors now suggest that it is the key to differentiation.

Figure 8 talks about JNK pathway in progenitor differentiation.

Although the authors' claims are well supported in general, Figures 6-8 make me wonder whether FAO contributes to progenitor cell differentiation through metabolic aspects at all. Although each piece of discoveries in this paper is important, I don't know if the authors can claim that FAO regulates progenitor differentiation through BOTH metabolic regulation and histone acetylation. Especially because Figure 7 shows that acetate supplementation is enough to rescue whd mutant phenotype, one has to wonder whether “metabolic” aspect ever contributes to self-renewal/differentiation choices. And if this is all about histone acetylation, what is the importance of glycolysis in self-renewing progenitors? Of course, it is an intriguing possibility that this is nature's elegant solution, natural shifts in metabolism also triggers epigenetic programming. I am not asking to do experiments to tell apart the possibilities, but the fact that their data do not necessarily show the requirement of FAO regulating metabolism per se in regulating differentiation, and the possibilities have to be carefully discussed.

I understand it might be technically difficult, but can they tell that histones are less acetylated in self-renewing progenitors compared to quiescent progenitors?

I think that Figure 8 is unnecessary. This does not add to the major message of the paper.

Reviewer #2:

Tiwari et al. present data that they claim suggests a potential role for FAO in metabolic regulation of progenitors in *Drosophila* larval lymph gland. They interpret their data to say that self-renewing and quiescent progenitors in the LG differ in their metabolic state. The latter exhibit FAO which results in acetylCoA production and thereby increase in histone acetylation and altered gene expression. Pharmacological increase in acetylation has the same effect whereas reduced histone acetylation maintains progenitor self-renewal. They state that the major outcome of the study is coupling FAO with epigenetic modifications linking to regulation of differentiation by JNK.

The metabolic state of blood cell progenitors has been studied extensively in vertebrates and to a limited extent in *Drosophila*. This study further characterizes progenitor sub-populations for their metabolic status. However, a major shortcoming of the manuscript is the misinterpretation of literature on which their hypothesis and analyses are based. As a result, the fundamental question being addressed about self-renewal and quiescence in LG progenitors is flawed. Further, there are systemic problems with the methods used to analyze, present and quantify data in this manuscript that make it impossible for me to judge the validity of their conclusions. Controls are rarely presented and cell sub-populations are not identified or marked. It is standard practice to include a control for all gene expression especially when reported for the first time and also to identify and distinguish between cell sub-populations when differences in their properties are claimed. Furthermore, the data in this paper are difficult to interpret and I am confused by how the authors came to their conclusions.

There is a general lack of quantification that would be required to make the interpretations presented.

Hence the first part (Figure 1) that emphasizes self-renewal and quiescence distracts from the main findings and the data are unclear and analyses questionable. Since the role of JNK in LG differentiation and its connection to metabolism are already established, the results presented here are only incremental to our understanding of LG progenitor maintenance. Further, a lot of the experiments related to this section will have to be repeated with appropriate markers and controls and quantitated rigorously to be convincing. Hence I think the present manuscript is not suitable for *eLife*.

A detailed review is provided below:

Introduction:

The authors seem to be unaware that of the different hematopoietic populations in vertebrates and how they compare with that of *Drosophila*. A major problem is the idea that there are self-renewing and quiescent progenitors in the larval LG. It is important to keep in mind that the LG is a powerful but limited model of vertebrate hematopoiesis. Hence a one-on-one correlation between cell types, states and metabolic status should not be made. The Abstract should be corrected to avoid the use of terms such as self-renewal and quiescence. The authors start the Introduction with the aim of testing the relation between metabolic intermediates and histone modifications but deviate to self-renewal and quiescence.

Introduction section

Studies on various stem cell scenarios primarily in Hematopoietic Stem Cells(HSCs) have established that various states adopted by stem cells like, quiescence, proliferation, and differentiation are liable to different metabolic requirements…

This suggests that a stem cell can occur in multiple states and is dynamically switching between them. However, this is not the case. Note that the references cited are for multiple sub-populations of vertebrate HSC. There has been no formal demonstration of stem cells in the LG nor of self-renewal. However, the authors interpret all their data in the context of self-renewal and quiescence and hence this is incorrect and misleading. Further, the experimental evidence does not support the conclusions (see below).

Introduction paragraph two

The blood progenitors found in *Drosophila* late larval lymph gland are akin to the vertebrate common myeloid progenitors (CMP)(Owusu-Ansah and Banerjee, 2009). They are quiescent, have high levels of ROS (Owusu-Ansah and Banerjee, 2009), lack differentiation markers, and can give rise to all *Drosophila* blood lineages.

This again is a misinterpretation. Firstly, CMPs are not quiescent, only long-term repopulating HSCs are proven to be quiescent. LG progenitor quiescence has not been conclusively demonstrated either. The authors interpret lack of proliferation as quiescence, these are two very different cellular states and associated with different gene expression profiles as well as cell phenotype. Finally, unlike in vertebrate studies, the ability of a LG progenitor to give rise to "all *Drosophila* blood lineages" has not been demonstrated.

Introduction paragraph three:

This should be edited to indicate that not all transcription factors are conserved between the two systems.

The authors equate proliferation with self-renewal- these are two different processes that share gene networks. EDU labelling indicates proliferation not self-renewal. Self-renewal is a very special property of stem cells that can also differentiate. Similarly, absence of EDU label indicates the cells are not proliferating or are cycling slowly- quiescence requires existence in the G0 phase, which has not been demonstrated in the LG.

Introduction paragraph four:

Dynamic states of self-renewal, quiescence and differentiation.

Note that a single cell does not switch between these phenotypic states. Differentiation is normally a one-way street down the lineage through each state, and necessitates metabolic shifts.

The data do not provide evidence that the progenitors primed to differentiate rely on FAO, this is merely correlative.

General comments on results:

Pharmacological treatments: This is another major point that the rest of the paper relies on, yet the authors again show no controls and inadequate quantification. While the entire paper rests on analysis of a few hundred cells in the LG, the Western blot analysis is done with the whole larva and results are interpreted in the context of the LG.

The authors claim that there are metabolic differences between progenitor sub-populations that correlate with potential to differentiate. However pharmacological treatments such as acetate feeding is done at the level of the whole animal. How much of acetate is sensed by the progenitors in question? how does a general increase in acetylation affect histone in various tissues including the LG? As systemic signals are key to maintaining blood progenitor homeostasis, the effects seen could be due to cell extrinsic factors. The conclusion that this treatment proves the role of FAO and histone acetylation in progenitor quiescence is far-fetched. Further there is no evidence of that glycolytic and FAO metabolizing sub-population are differentially affected.

It is similarly unclear in which progenitor sub-populations whd and JNK signaling are affected. Quantification is required for differentiation checked by P1 and progenitor status should be checked by dome; also differentiation to crystal cells and lamellocytes should be examined. This is the first report of metabolic requirement for progenitor maintenance; a thorough analysis of various differentiation markers is essential.

Figure wise comments

Figure 1 is fraught with misinterpretations. (Results paragraph one and two). Further there is no quantitation whatsoever and a lack of controls. Two different developmental stages are compared and cell identity is arbitrarily assigned. No markers are used for self-renewing or quiescent cells and it is not clear how the authors identify these. Edu positive cells are only a subset of the second instar LG (these could be differentiating) and seen only in the CZ of the third instar.

Figure 1D, E states that self-renewing early stage progenitors exhibit elevated levels of glucose uptake compared to late stage quiescent progenitors. In this case a proliferation assay is required along with markers to demarcate the MZ/CZ boundaries to show that the uptake is less in the prohemocyte pool of the 3rd instar lymph gland. It is likely that the 2NBDG labeled cells could be late progenitors in the second instar that persist in CZ. Without appropriate markers this cannot be distinguished. Further third instar progenitors may be slower proliferating or arrested in the cell cycle- this is not quiescence. A cell cycle analysis needs to be done through L2 and L3 to resolve the two states, so called self-renewal and differentiation.

What the authors call quiescence is likely a non-proliferative state that precedes differentiation, which is well documented in many developmental contexts. Similarly, as mentioned in Results paragraph two, glycolysis is seen in rapidly proliferating cells, such as cancer cells. This is not necessarily self-renewal.

GlutGFP expression is patchy (Figure 1F, G) and not clear whether it is in or around the cells, this looks more like background. A no primary antibody control and a Glut RNAi control are required. MZ should be marked by dome or DEcad and perform EdU labeling in order to show that High Glut 1-GFP is expressed in proliferating progenitors. Similarly, aldolase expression (supplement 1A, B) is not convincing and is seen only in the periphery of the second instar.

Figure 1 H, I imaging plane is unclear. It is not clear what the authors are trying to show. Imaging parameters are not uniform.

Figure 1 K: Which area of the LG is shown? Nuclear staining should be included.

Figure 1 L-O', it is impossible to comment on the status of mitochondrial network with this analysis. Which part of the LG is shown? High resolution images with live tracking and video microscopy are required to analyze mitochondrial morphology, length and dynamics. Without detailed information regarding these parameters in L2 and L3 progenitors and in differentiated CZ cells, one cannot make any comparisons as there is a large variation in mitochondrial size and morphology within any population. This needs co-analysis with progenitor and differentiation markers and thorough quantitation- CZ/ MZ mitochondrial pattern needs to be mapped first.

There is no data to show mitobiogenesis.

Figure 1Q, R: the accumulation/increase of Lsd-2 in the 3rd instar lymph gland would be clear if 2nd instar images are presented for Nile Red.

Figure 1 requires quantification and insets indicating region of interest; double labeling with progenitor marker and or CZ markers or EdU labeling is essential.

Figure 2: Dapi staining or demarcation of LG boundary as well as MZ/CZ marker staining should be shown. Crystal cell and lamellocytes differentiation status should also be analyzed.

2H,L Q Graph is misleading, it suggests there is a decrease in dome^+^ area. Y axis should indicate plamatocyte % or area. What is differentiation index?

Figure 3E-G: the authors claim higher number of EdU incorporation is seen in the progenitors, however the images provided show higher EdU signal at the periphery; also CZ/MZ boundary is not demarcated, progenitor marker should be used and data quantitated.

Figure 4F, G quantitation required for commenting on precocious differentiation (Hml^+^ve cells).

Figure 5F-H quantitation required for EdU labeling to show that feeding with 2-DG rescues the hyperproliferation phenotype.

Figure 6A-C claims loss of Histone Acetyl Transferase (HAT), Gcn5 and chm phenocopies differentiation defect seen in the progenitors of FAO loss of function. What is the effect on proliferation in panels 6A-G. Does it corroborate with active proliferation seen in whd mutants.

For Figure 7F materials method suggests western blots were done from larva of the respective genotypes, this cannot be correlated to the LG effects. It would be more appropriate to show H3K acetylation in situ in the lymph gland.

Figure 7 H-I Lymph gland images required in order to visualize that the addition of H3K acetylation increases in L-carnitine fed v/s unfed larvae.

Figure 8A- C, expression of dominant negative allele of basket (bsk, *Drosophila* ortholog of Jun-Kinase) results in a compromised differentiation analogous to whd knockdown. Can the phenotype be rescued by feeding L-carnitine or acetate?

The effect of Jun (bsk) on whd is shown at the transcriptional level. Does Jun (bsk) affect enzymes of the FAO pathway? Also Jnk is known to activate Foxo, what is the status of Foxo reporter (Thor-LacZ) in whd mutants.

Foxo overexpression advances differentiation (Owusu-Ansah and Banerjee, 2009), does overexpression of Foxo affect/ rescue whd mutant phenotype by inducing differentiation.

Reviewer #3:

Mandal and colleagues use the lymph gland of *Drosophila* to identify the genetic events by which a metabolic switch (from glycolysis to fatty acid oxidation, FAO) drives a developmental transition (from progenitor to differentiated cells). They first combine fluorescent glucose uptake analysis, expression of metabolic enzymes and glucose transporters and exposure to a glucose antagonist to present evidence that early but not late lymph glands make use of glucose to grow. They next combine genetic analysis and pharmacological treatments to present functional evidence that a metabolic switch to FAO is required in late lymph gland cells to differentiate. FAO requirement does correlate with a change in the protein levels of Ci but appears to act downstream of the differentiation signal ROS. Genetic or pharmacological premature induction of FAO in early lymph glands is sufficient to drive progenitor cells into differentiation and the use of glucose is an absolute requirement for progenitor cells to keep proliferating in the absence of FAO. At the end of the manuscript, experimental data indicate that FAO is required in differentiated cells for histone acetylation downstream of ROS-induced JNK signaling.

The paper is well written (even though grammar should be improved), figures self-explanatory and the main message (from a metabolic to a developmental switch) and experimental setting (combination of genetics and pharmacology) is timely and appropriate for a journal like *eLife*.

[Editors’ note: further revisions were suggested prior to acceptance, as described below.]

Thank you for submitting your article "Fatty acid β-oxidation is required for the differentiation of larval hematopoietic progenitors in *Drosophila*" for consideration by *eLife*. Your article has been reviewed by three peer reviewers, and the evaluation has been overseen by K VijayRaghavan as the Senior and Reviewing Editor. The following individual involved in review of your submission has agreed to reveal their identity: Yukiko M Yamashita (Reviewer #1).

The reviewers have discussed the reviews with one another and the Reviewing Editor has drafted this decision to help you prepare a revised submission.

Summary:

Mandal and colleagues beautifully use the lymph gland of *Drosophila* to identify a role of FAO in the differentiation step from progenitor to differentiated blood cells. Authors combine a great variety of tools (drugs, reporters, inhibitors) together with powerful genetics (RNAi, CRISPR, mutants) to draw a line from JNK to Whd expression, from Whd to FAO, and from FAO to histone acetylation through the production of acetlyCoA. The present manuscript is a revised version of a previous one submitted to *eLife* and has improved in many aspects (experimental setting is very elegant, the combination of genetics and drugs/reporters is extraordinary and the epistatic analysis to demonstrate their claims is very robust). Others (writing, figure presentation, cartoons, etc) should be seriously improved in order to attract the attention of the general reader. While we strongly support publication of this manuscript in *eLife*, the following issues should be thoroughly addressed. Otherwise, even though the science is excellent, the quality of the manuscript will not be deserving to be published in *eLife*.

While the data are generally of good quality, there are significant gaps that need to be filled. The results must be seen in the context of what we already know – the link between JNK and FAO is known, as is the role of acetyl CoA in epigenetic modification. Tiwari et al. show that this also occurs in and is essential for blood progenitor differentiation. However, the same has been shown in vertebrate HSC. The authors conflate two very different cell types and approaches. Vertebrate HSCs have been extensively characterized and their quiescence, multipotency and repopulating ability has been demonstrated unambiguously. The same is not true for *Drosophila* hematopoiesis – unfortunately several studies try to force cellular parallels between the two. This is not required and *Drosophila* is an excellent hematopoietic model even though it does not have HSC.

Essential revisions:

Authors should talk to non-*Drosophila* colleagues to help them in thoroughly revising the following aspects of the manuscript:

1) Abstract is not well written and should be improved, specially the last two sentences. It does not summarize the major aspects of the work and will not attract the attention of any reader that has not a special interest in the *Drosophila* LG.

2) Cartoon depicted in Figure 1A and explanation in the Introduction do not have anything to do with each other (CZ only explained in the text and others only in the figure) in the figure and does not clarify anything. It's just confusing and should be completely changed and re-done.

3) Last paragraph of the Introduction should summarize the major findings and they do not so. References to the JNK/ROS axis are lacking. Jnk itself is not introduced. Again, this part of the manuscript is intended to attract the non-expert community

4) First part of the results: the why of using two developmental stages and different markers (pvf2, Dome, Hml, etc) is not clearly explained anywhere. Dome^+^, Dome^+^ Pxn^+^, Dome^-^, etc nomenclature is confusing (2N-O). Any way of improving it? Authors should be able to avoid technicalities.

5) Tens of mistakes (prolifeartes, progenitors experiences, etc) should be corrected.

6) Writing and conclusions in all sections should be improved.

Scientific issues:

7) Lipid markers and CG3902 expression. A Z section is required to demonstrate that the differential expression/incorporation levels between progenitor and differentiated cells is real not a consequence of the focal plane or thickness of the tissue.

8) Authors should use ubiquitous Gal4 drivers (NOT dome-Gal4) to demonstrate differences in the amount of mitochondria and cell cycle profiles (Fucci) between progenitor and differentiated cells.

9) Some negative results (changes in Hh and ROS levels w/o any impact on the process) should be moved to supplementary data.

10) CycB degradation of FlyFucci is not degraded in G1, as authors claim in subsection “Loss of FAO caused an increase in progenitor proliferation, and high redox levels”.

11) Single channels to monitor differences in EdU incorporation levels should be included (3G-I, 5M-P). Figure 5E, F is not described properly in the text.

12) Y axis in panel 3F: should be ROS and not Ci, right?

13) Materials and methods: Codes and/or references of all Fly stocks required. By the way, many of the fly lines used in the work are not included in Materials and methods. Stocks such as AyGal4 should be properly explained.

14) There is a lot of room for improvement in the organization of panels within Figures. Labels of most histograms are too small and error bars are not explained in the legends (SD or SDM?).

15) What is the trigger for differentiation? Is dietary lipid content a regulator of this as has been shown in vertebrates for regulation by FAO?

Mere carnitine supplementation is not sufficient as carnitine is not an essential nutrient, is abundant in the body under regular diet and carnitine deficiency results from mutations in its transport machinery. As the claim is that whd links JNK to FAO, and is transcriptionally regulated by JNK, a genetic rescue experiment with whd is essential, especially as Foxo overexpression increases differentiation. This manuscript often relies on compound treatment strategies alone despite having a genetically tractable model. Yet the role and effect of endogenously generated systemic cues is completely disregarded.

[Editors' note: further revisions were suggested prior to acceptance, as described below.]

Thank you for resubmitting your work entitled "Fatty acid β-oxidation is required for the differentiation of larval hematopoietic progenitors in *Drosophila*" for further consideration by *eLife*. Your revised article has been evaluated by K VijayRaghavan (Senior Editor) and a Reviewing Editor.

The manuscript has been improved but there are some remaining issues that need to be addressed before acceptance, as outlined below:

Summary

In this revised manuscript, Mandal and colleagues explore the function of FAO (fatty acid oxidation) in differentiation of hematopoietic progenitors in *Drosophila* larval hematopoietic organ, lymph gland (LG). They show that supplementation with acetate rescues the phenotype associated with defective FAO, and that FAO is downstream of JNK, a known regulator of differentiation, providing insights into how metabolic pathway may be intertwined with differentiation pathway. They use abundant genetic models (mutants, reporters) to support their claim.

The revised manuscript by Tiwari et al. has addressed all the comments made on the previous version. The authors have repeated experiments, included controls, or done additional experiments where required. Importantly, progenitors have been correctly identified and analyzed. Also, progenitor-specific genetic analysis (perturbation of glucose uptake; downregulation of withered; upregulation of FAO in JNK depletion) has been done. The data are of good quality, presented clearly and gaps in analysis have been filled satisfactorily. The Discussion is much improved. This manuscript is significantly improved over the last submission and is an important contribution to the field.

However, there are still issues with the writing as following, which should be addressed prior to publication. Most of comments can be addressed by modifying writing, but some possible experiments are suggested : these experiments are just suggestions (in case they already have data) and not intended to be “essential” for revision, under the new editorial policy at *eLife* during the covid-19 pandemic.

Essential revisions

1) In the Introduction, the authors contrast hematopoietic stem cells (HSCs) in mammals with *Drosophila* hematopoietic progenitors, assuming that studying *Drosophila* hematopoietic progenitors will provide insights into how mammalian non-HSC progenitors are regulated by FAO. However, given that *Drosophila* does not have HSCs, and whether they represent HSCs or progenitors remain unclear (or whether it's comparison to mammalians HSC vs. progenitor is more relevant). In mammalian HSCs, FAO is required for stem cell maintenance, whereas in *Drosophila* LG, FAO is required for differentiation. I see that this difference may have driven the authors to make the above contrast, but I don't think it's a well-grounded argument and they should modify the Discussion accordingly.

2) Figure 1—figure supplement 1C shows reticular mitochondria as an indication of high FAO. Ideally, they should show less reticular mitochondria in other cell types in comparison.

3) Figure 4: to nail down the pathway, the epistasis analysis would be ideal, such as combination of genetic mutations (inhibit differentiation, such as whd mutant) and L-carnitine (drive differentiation). Especially, they conduct similar experiments in Figure 5, Figure 6, and Figure 8.

4) Figure 5I-L, 2-DG treatment rescues G2/M arrest defects in whd mutant, but does not seem to rescue differentiation defect. Any discussions?

---

## [Author Response]

[Editors’ note: the authors resubmitted a revised version of the paper for consideration. What follows is the authors’ response to the first round of review.]

While the reviewers started with different perspectives, we have converged to a common position. We suggest that the authors go through our preamble, which captures this converged view, and also the specific concerns of each reviewer. While some of our major concerns can be addressed by substantive re-writing, cutting down generalisations and over-interpretation others require experimental attention.

Thanks for your inputs. We have worked extensively on the manuscript addressing all the concerns regarding the generalization and over interpretation. However, we are still open to suggestions that might improve the readability of the current manuscript.

One important issue, which was reiterated in our consultations, is exemplified by the following statement and related data "We show that the self-renewing hemocyte progenitors prefer glycolysis while the quiescent progenitors adopt Fatty acid β-oxidation (FAO) for their differentiation. Perturbation of β-oxidation in progenitors results in loss of quiescence…." Though the authors' primary question is centered on distinguishing metabolic states within progenitors, the authors do not identify these. Generic results are interpreted as applying to specific populations and extended to homeostasis.

In the current manuscript, we have focused on the differentiation of progenitors instead of their homeostasis. The stages that we have worked out in detail include the third instar early and late. We want to draw your attention towards the progenitor subsets populating the LG at these two-time points. Please refer to Figure 2M and 2O. The Dome^-^ pre-progenitors are present in early third along with Dome^+^ and Dome^+^ Pxn^+^ Hml^+^ IP cells. However, in the late third instar, the pre-progenitors (Dome^-^) are absent. These data are in sync with the findings from (Ferguson and Martinez-Agosto, 2017).

Using *Hnf4-Gal4.UAS-GFP* as the reporter for β-oxidation, we show that a Dome^-^ preprogenitors start expressing *Hnf4.GFP* early on, which becomes conspicuous in Dome^+^ progenitors in the third instar. However, it is downregulated in Dome^+^Hml^+^ IP cells (Figure 1 B-C''). A similar trend is noticeable in the quantitative analysis in Figure 1D).

Staging larvae and using appropriate established Gal4 drivers is, of course, acceptable, but the authors have not done this properly. Unlike, for example, the wing and eye discs, the larval LG do not have a uniform, reproducible structure or pattern. One cannot delineate zones morphologically or by using a single marker. Progenitor sub-populations (dome negative pre-progenitors, dome^+^ progenitors and dome^+^Pxn^+^hml^+^ intermediate progenitors) (reviewed in Banerjee et al., Flybook2019) and the three types of differentiated cells (plasmatocytes, crystal cells and lamellocytes) are functionally and metabolically different. However, the authors do not distinguish between these populations in their analysis.

We have now used appropriate double markers for characterization as well as expression studies and taken utmost care for the staging of the larvae.

For example, mitochondrial analysis is shown at low resolution and in an undefined population (Figure 1L, M)

We understand your concern. We have now performed super-resolution microscopy of the mitochondria in the Dome^+^ progenitors of late third instar lymph gland (*DomeGal4-UASRFP >UAS-mito-GFP*). Please refer to Figure 1—figure supplement 1C-C' and Video 1 and discussed in Main text.

and hml>GFP is used only as a marker of differentiation (Figure 4).

Instead of using Hml only, we have now used Dome and P1 to mark the progenitors and differentiated plasmatocytes respectively. Please refer to Figure 4 of the current manuscript.

The assertion of the manuscript is that they are distinguishing between progenitor types, which they call self-renewing and quiescent. This can be done only upon identifying and characterising subpopulations. Further, since they incorrectly use the presence or absence of proliferation marks as a readout for self-renewal and quiescence, it is not clear how the claim that these are two metabolically distinct precursor states is valid. Rather, based on the analysis done (EdU, Figure 1B,C) they are simply proliferating or non-proliferating cells (which may or may not be progenitors), the latter poised for, or in the process of differentiation. Expectedly, their metabolic states differ- and the authors show that these are glycolysis for the former and FAO for the latter. Figure 1B,C: TepIV and EdU labelling – it looks as if there are co-labeled cells, but these are ignored. Also TepIVGAL4 reporter does not label the entire progenitor pool. The existence of self-renewing or quiescent progenitors has not been formally demonstrated in the LG- if the authors had demonstrated their existence, this would have been very interesting indeed.

We are very sorry for the confusion created regarding the cell cycle status of the progenitors. We used the terminologies “self-renewal” and “quiescence” to denote the “proliferative” and “non-proliferative” states of the second instar and late third instar lymph gland progenitors, respectively. We want to bring to your kind attention that a recent work from the lab (Sharma et al., 2019) we found that the Dome^+^ Pxn- progenitors in the late third instar lymph gland are arrested in the G2 phase of the cell cycle. We have done an extensive analysis of the cell cycle in different progenitor sub-populations to establish the G2 arrest. In the revised version of our manuscript, we have used the terminology “proliferating” and “G2 arrested” to address the cell cycle status of the lymph gland progenitors.

The metabolic fluctuations we are concerned about are intra-LG. This is a major challenge in characterizing metabolic states of heterogeneous progenitors. Since all analysis is done only with one progenitor marker at a time (mainly dome) which is also expressed in early stages of differentiation, or with no marker at all, (Example Figure 1 ; Figure 1—figure supplement 1, compare panels A and C) claims regarding metabolic differences are not substantiated.

We understand your concern and have now used double markers in the current manuscript to assay the metabolic status of the heterogeneous population. Please refer above for the details on progenitor sub populations.

As is also evident from the images in Figure 1, the same tissue from the same genotype and stage varies greatly in terms of size, shape and organisation, hence rigorous quantitation with additional markers is essential.

We fully understand your concern and have taken the utmost care of staging, used appropriate markers, and did rigorous quantitation of expression patterns and phenotypes wherever possible. Thanks for your guidance.

The authors seem confused about what the focus of their study is. They state in the Abstract that the major outcome of the study is coupling FAO with epigenetic modifications linking to regulation of differentiation by JNK, while this part of the study has merit, it is not very relevant to the primary question they pose about blood cell progenitor homeostasis.

Based on your guidance, we have chosen to build up the manuscript concentrating on the role of FAO in progenitor differentiation, which the reviewers, along with you found to have merit. Progenitor homeostasis and glycolysis will be dealt in a separate manuscript at a later date.

The reviews give a very detailed analysis, including comments for almost every panel of Figure 1. However, some examples are expanded on a couple of specific cases below.Figure 1D, E states that self-renewing early stage progenitors exhibit elevated levels of glucose uptake compared to late-stage quiescent progenitors. In D there is no marker for progenitors or differentiated cells- clearly there are varying intensities of signal. How can one assume that all 2-NBDG cells are progenitors- early or late? Further, given variation between preps, how does one compare levels between second and third instar without having a control to normalise it or assess relative levels between progenitors and differentiated cells? Assuming second and third instar preps were stained and imaged in the same experiment, how much variation is seen? A graph representing this is essential as the lymph gland morphology can vary greatly depending on the status of the larval cultures.

We understand your concern. As mentioned earlier, the current manuscript deals with differentiation, and therefore, we have analyzed third instar stages. Markers for delineating CZ/MZ (dome-Meso-EBFP2) with 2-NBDG have been used along with quantitative analysis. Please refer to Figure 5 B-C'' and Figure 5D for quantitation. The maximized uptake of glucose is in the peripheral hemocytes (2-NBDG in *HmlDsRed* of control is further endorsed: Figure 5—figure supplement 1)

Additionally, a proliferation assay is required along with the uptake or demarcation of the MZ/CZ boundaries to show reduced uptake in the pro-hemocyte pool of the 3rd instar lymph gland.

We want to draw your attention to the fact that there is a technical constraint in performing any antibody assay or proliferation assay with 2-NBDG. Mild fixation is essential for 2– NBDG assay. Upon prolonged fixation or longer incubation 2-NBDG signals quenches.

In order to map the peripheral signal of 2-NBDG in the LG, we have used Hml-dsRed (Figure 5—figure supplement 1), which makes it evident that the Hml population is indeed high in glucose uptake compared to Dome^+^ progenitors.

Figure 1F, G – To say that there is high Glut-1 GFP in proliferating progenitors of the second instar, labelling with a proliferation marker, progenitor marker, and differentiation marker is required along with quantitation to show that High Glut 1-GFP is expressed in proliferating progenitors.

Since the current manuscript dwells on FAO mediated differentiation, and hence we plan to include this experiment in a separate manuscript that deals with Glycolysis, progenitor proliferation, and their homeostasis.

Figure 3E-G, the authors claim a higher number of EdU incorporation is seen in the progenitors, however, the images provided show higher EdU signal at the periphery; also CZ/MZ boundary is not demarcated, and progenitor markers are not used.

In the current manuscript, we have brought *DomeGFP* in the background of *whd^1^* mutant to address this issue. Please refer to Figure 3 G-H and quantitation in Figure 3I.

If the results about FAO and its downstream effects in the LG are de-linked from progenitor sub-types and homeostasis, this would be more acceptable.

Thanks for your guidance. In the current submission, we exactly did this. We have worked on FAO and its downstream effects on the hemocyte progenitor of third instar stages. Our current study connects ROS-JNK and FAO circuit that regulates progenitor differentiation by epigenetic modification.

Currently, two parts of the manuscript (progenitor types and metabolic status connecting to signaling) are not well connected. The authors may want to consider that they are perhaps best dealt separately and not in the same manuscript.

Please refer to the point above. Moreover, we have now strengthened the connection between JNK and FAO with histone acetylation in the current version.

Reviewer #1:This study by Mandal and colleagues studied the role of metabolic circuitry in stem cell maintenance/differentiation. General impacts of metabolism on cell fates, particularly those between stem cells and their progenies, have increasingly drawn researchers' attention in recent years. By using *Drosophila* hematopoietic progenitors (in lymph gland), the authors explored the role of metabolic circuitry in this lineage. The main conclusion is that actively proliferating stem cells utilize glycolytic pathway whereas quiescent stem cells favor fatty acid β-oxidation, which promotes differentiation.Furthermore, they propose that FAO generates acetyl-CoA, which not only generates energy through TCA cycle, but can contribute to protein (e.g. histone) acetylation. Perturbation of FAO leads to defects in quiescence (progenitor starts proliferating and cannot differentiate). Upregulation of FAO leads to stem cell quiescence and precocious differentiation. The authors link this observation to histone acetylation and thus epigenetic regulation.Figure 1: Does blocking glycolysis have any impact on differentiation? Does it only affect progenitor proliferation without impacting later differentiation?

Although the focus of the current paper is differentiation and β-oxidation, we, however, have done this experiment. Glycolysis inhibition by 2-DG supplemented feeding affects progenitor proliferation without having much effect on Hml^+^ Cortical Zone. This experiment implicates that progenitor differentiation is not dependent on glycolysis. We intend to put together a separate manuscript on Glycolysis and progenitor homeostasis on a later date.

Figure 2: blocking FAO by means of whd knockdown resulted in increased progenitors and decreased differentiating cells. Further in Figure 3, they show that loss of FAO also increases progenitor proliferation and thus progenitors are not entering quiescence. Collectively these results indicate that FAO is required for progenitors to enter quiescence, and thus their “differentiation defect” is secondary to their inability to enter quiescence. These results are quite clear, but when I was reading the description for Figure 2, I was misled to think that FAO blocks differentiation. Any ways to make this transition (Figure 2 to Figure 3) a bit less misleading?

Thanks lot for appreciating our work. Sorry for the confusion created. We have worked on the description of Figure 2 and modified it.

In Figure 4, they show that upregulation of FAO is sufficient to induce quiescence and subsequent differentiation in progenitors.Figure 5 demonstrate the inter-relationship between glycolysis and FAO. Defective FAO led to sustained glycolysis.Figure 6-7 slightly shift in the focus and now addresses FAO's function in histone acetylation. The authors now suggest that it is the key to differentiation.Figure 8 talks about JNK pathway in progenitor differentiation.Although the authors' claims are well supported in general, Figures 6-8 make me wonder whether FAO contributes to progenitor cell differentiation through metabolic aspects at all. Although each piece of discoveries in this paper is important, I don't know if the authors can claim that FAO regulates progenitor differentiation through BOTH metabolic regulation and histone acetylation. Especially because Figure7 shows that acetate supplementation is enough to rescue whd mutant phenotype, one has to wonder whether “metabolic” aspect ever contributes to self-renewal/differentiation choices. And if this is all about histone acetylation, what is the importance of glycolysis in self-renewing progenitors? Of course, it is an intriguing possibility that this is nature's elegant solution, natural shifts in metabolism also triggers epigenetic programming. I am not asking to do experiments to tell apart the possibilities, but the fact that their data do not necessarily show the requirement of FAO regulating metabolism per se in regulating differentiation, and the possibilities have to be carefully discussed.

We want to draw the attention of the reviewer to the fact that the metabolic signal ROS is known to trigger JNK in the progenitors to elicit their differentiation (Owusu-Ansah and Banerjee, 2009). We now show that JNK in the progenitors at this stage regulates the metabolic process: FAO through the transcription of *whd* (the rate-limiting enzyme of FAO).

The end product of FAO is Acetyl CoA, a metabolite implicated in histone acetylation/epigenetic modification. We show that loss of JNK or whd affects progenitor differentiation and histone acetylation. Restoring histone acetylation alone through acetate supplementation results in rescuing the differentiation defect in the mutants.

The acetate supplemented is converted to the end product of FAO: Acetyl CoA, the metabolite that is essential for histone acetylation. The involvement of Acetyl CoA in facilitating differentiation is further evidenced when on genetically downregulating AcCoAs (the major enzyme in acetyl CoA generation) from the progenitor leads to a halt in their differentiation.

Our study thus establishes that for hemocyte progenitor differentiation, the metabolic process FAO involves its metabolite Acetyl CoA for epigenetic modification.

We have discussed this aspect in the Discussion section of the revised manuscript. Thanks for your suggestion.

I understand it might be technically difficult, but can they tell that histones are less acetylated in self-renewing progenitors compared to quiescent progenitors?

Although we are not presenting the observations on the requirement of glycolysis in the proliferating progenitors of early instars, we can comment that there will not be much difference in the overall acetylation of histones. The levels of acetyl COA required for acetylation in proliferating progenitors of the second instar are not compromised owing to the high glycolytic index at this stage. Later on, in development, glycolysis dampens, and FAO picks up to provide the Acetyl COA moiety required for histone acetylation in the progenitors.

I think that Figure 8 is unnecessary. This does not add to the major message of the paper.

Thanks for your input. We have done additional experiments to establish the ROS-JNK-FAO circuit is essential for differentiation. JNK has been previously implicated in epigenetic modification via histone acetylation. Our study unravels the unexpected link of JNK with epigenetics through FAO. We have accordingly modified the model. Please refer to Figure 8—figure supplement 1 of the revised manuscript.

Reviewer #2:Tiwari et al. present data that they claim suggests a potential role for FAO in metabolic regulation of progenitors in *Drosophila* larval lymph gland. They interpret their data to say that self-renewing and quiescent progenitors in the LG differ in their metabolic state. The latter exhibit FAO which results in acetylCoA production and thereby increase in histone acetylation and altered gene expression. Pharmacological increase in acetylation has the same effect whereas reduced histone acetylation maintains progenitor self-renewal. They state that the major outcome of the study is coupling FAO with epigenetic modifications linking to regulation of differentiation by JNK.The metabolic state of blood cell progenitors has been studied extensively in vertebrates and to a limited extent in *Drosophila*. This study further characterizes progenitor sub-populations for their metabolic status.However, a major shortcoming of the manuscript is the misinterpretation of literature on which their hypothesis and analyses are based. As a result, the fundamental question being addressed about self-renewal and quiescence in LG progenitors is flawed. Further, there are systemic problems with the methods used to analyze, present and quantify data in this manuscript that make it impossible for me to judge the validity of their conclusions.

We used the terminologies “self-renewal” and “quiescence” to denote the “proliferative” and “non-proliferative” states of the second instar and late third instar lymph gland progenitors, respectively. Our interpretation regarding the quiescent state of the progenitors was based on the published literature available at that time point. We want to bring to your kind attention that in a recent work from the lab (Sharma et al., 2019), we found that the Dome^+^ Pxn^-^ progenitors in late third instar lymph gland are arrested in the G2 phase of the cell cycle. We have done an extensive analysis of the cell cycle in different progenitor sub-populations to establish the G2 arrest.

However, we do understand that self-renewal has not yet been demonstrated in the LG. We are very sorry for the confusion created regarding the cell cycle status of the early progenitors. In the revised version of our manuscript, we have used the terminology “proliferating” and “G2 arrested” to address the cell cycle status of the lymph gland progenitors.

Controls are rarely presented and cell sub-populations are not identified or marked. It is standard practice to include a control for all gene expression especially when reported for the first time and also to identify and distinguish between cell sub-populations when differences in their properties are claimed. Furthermore, the data in this paper are difficult to interpret and I am confused by how the authors came to their conclusions.There is a general lack of quantification that would be required to make the interpretations presented.

We fully understand your concern and have taken the utmost care of staging, used appropriate markers, and did rigorous quantitation of expression patterns and phenotypes wherever possible. Thanks for your guidance.

Hence the first part (Figure 1) that emphasizes self-renewal and quiescence distracts from the main findings and the data are unclear and analyses questionable. Since the role of JNK in LG differentiation and its connection to metabolism are already established, the results presented here are only incremental to our understanding of LG progenitor maintenance.

Here we disagree with the reviewer’s statement that our result is only an incremental to our understanding of LG progenitor differentiation. Although ROS dependent JNK activation was implicated in the differentiation of hemocyte progenitors, the mechanistic basis of this phenomenon was not known. We show that JNK can regulate Fat metabolism by the transcriptional activation of the rate-limiting enzyme of FAO, *whd*. Acetyl-CoA, the product of FAO, in turn, is essential for causing epigenetic modification. Our study, therefore, unravels the unexpected link of JNK with FAO. That FAO activation is mandatory for differentiation is clear from the observation that upon JNK/bsk loss, the halted differentiation can be rescued by upregulating FAO (through L-carnitine supplementation). Please refer to Figure 8 in the revised manuscript.

This is the first report that describes the involvement of the metabolic process FAO and its metabolite Acetyl CoA for epigenetic modification crucial for hemocyte progenitor differentiation. Our study further links the previously described ROS-JNK circuit to metabolism and therefore, should not be undermined.

Further, a lot of the experiments related to this section will have to be repeated with appropriate markers and controls and quantitated rigorously to be convincing. Hence I think the present manuscript is not suitable for eLife.A detailed review is provided below:Introduction:The authors seem to be unaware that of the different hematopoietic populations in vertebrates and how they compare with that of *Drosophila*. A major problem is the idea that there are self-renewing and quiescent progenitors in the larval LG. It is important to keep in mind that the LG is a powerful but limited model of vertebrate hematopoiesis. Hence a one-on-one correlation between cell types, states and metabolic status should not be made. The Abstract should be corrected to avoid the use of terms such as self-renewal and quiescence. The authors start the Introduction with the aim of testing the relation between metabolic intermediates and histone modifications but deviate to self-renewal and quiescence.

As per the suggestion in the preamble, we have split the paper into two. In the current submission, we have worked on FAO and its downstream effects on the hemocyte progenitor differentiation. We have connected ROS-JNK and FAO circuit regulating progenitor differentiation by epigenetic modification.

Introduction sectionStudies on various stem cell scenarios primarily in Hematopoietic Stem Cells(HSCs) have established that various states adopted by stem cells like, quiescence, proliferation, and differentiation are liable to different metabolic requirements.This suggests that a stem cell can occur in multiple states and is dynamically switching between them. However, this is not the case. Note that the references cited are for multiple sub-populations of vertebrate HSC.

Sorry for the confusion created. We have modified the statement in the revised manuscript.

There has been no formal demonstration of stem cells in the LG nor of self-renewal. However, the authors interpret all their data in the context of self-renewal and quiescence and hence this is incorrect and misleading. Further, the experimental evidence does not support the conclusions (see below).

Although self-renewal has not yet been experimentally demonstrated in LG, it has neither been ruled out. In fact, the stem cells have been identified in larval 1st instar lymph gland through clonal analysis (Minakhina et al., 2010) as well as using several parameters of stem cells like multipotency, niche dependency, lineage tracing, gain and loss of function genetic analyses along with genetic ablation studies (Dey et al., 2016).

Our aim of discussing HSC stems from the idea that the requirement of FAO metabolism is worked out in HSC, but its role in progenitor is yet to described. Using *Drosophila* larval blood progenitors, we proposed to work out the role of FAO.

Introduction paragraph twoThe blood progenitors found in *Drosophila* late larval lymph gland are akin to the vertebrate common myeloid progenitors (CMP)(Owusu-Ansah and Banerjee, 2009). They are quiescent, have high levels of ROS (Owusu-Ansah and Banerjee, 2009), lack differentiation markers, and can give rise to all *Drosophila* blood lineages.This again is a misinterpretation. Firstly, CMPs are not quiescent, only long-term repopulating HSCs are proven to be quiescent.

We have used an understanding prevailing in the field, but understanding your concern we have removed this comparison.

LG progenitor quiescence has not been conclusively demonstrated either. The authors interpret lack of proliferation as quiescence, these are two very different cellular states and associated with different gene expression profiles as well as cell phenotype.

Our interpretation regarding the quiescent state of the progenitors was based on the published literature available at that time point. We want to bring to your kind attention that a recent work from the lab (Sharma et al., 2019), we found that the Dome^+^ Pxn^-^ progenitors in late third instar lymph gland are arrested in the G2 phase of the cell cycle. We have done an extensive analysis of the cell cycle in different progenitor sub-populations to establish the G2 arrest.

However, we do understand that self-renewal has not yet been demonstrated in the LG. We are very sorry for the confusion created regarding the cell cycle status of the progenitors. In the revised version of our manuscript, we have used the terminology “proliferating” and “G2 arrested” to address the cell cycle status of the lymph gland progenitors.

Finally, unlike in vertebrate studies, the ability of a LG progenitor to give rise to "all *Drosophila* blood lineages" has not been demonstrated.

We chose to disagree with this point. Studies from Dr. Utpal Banerjee’s laboratory have established that the progenitors are multi-potential (Jung et al., 2005). Also, different studies have shown that transdifferentiation (Leitao et al., 2015) can happen in the blood lineages, causing a change in fate from one cell type to the other (Leitao et al., 2015). We have also tried it out with the potential HSCs as well as the progenitors. In all cases, multipotentcy has been endorsed (Dey et al., 2016).

Introduction paragraph three:This should be edited to indicate that not all transcription factors are conserved between the two systems.

Modified accordingly.

The authors equate proliferation with self-renewal- these are two different processes that share gene networks. EDU labelling indicates proliferation not self-renewal. Self-renewal is a very special property of stem cells that can also differentiate. Similarly, absence of EDU label indicates the cells are not proliferating or are cycling slowly, quiescence requires existence in the G0 phase, which has not been demonstrated in the LG

We used the terminologies “self-renewal” and “quiescence” to denote the “proliferative” and “non-proliferative” states of the second instar and late third instar lymph gland progenitors, respectively. Our interpretation regarding quiescent state of the progenitors was based on the published literature available at that time point. We want to bring to your kind attention that in a recent work from the lab (Sharma et al., 2019), we found that the Dome^+^ Pxn^-^ progenitors in late third instar lymph gland are arrested in the G2 phase of the cell cycle. We have done an extensive analysis of the cell cycle in different progenitor sub-populations to establish the G2 arrest.

However, we do understand that self-renewal has not yet been demonstrated in the LG. We are very sorry for the confusion created regarding the cell cycle status of the early progenitors. In the revised version of our manuscript, we have used the terminology “proliferating” and “G2 arrested” to address the cell cycle status of the lymph gland progenitors.

Introduction paragraph four:Dynamic states of self-renewal, quiescence and differentiation.Note that a single cell does not switch between these phenotypic states. Differentiation is normally a one-way street down the lineage through each state, and necessitates metabolic shifts.The data do not provide evidence that the progenitors primed to differentiate rely on FAO- this is merely correlative.

We beg to disagree with this comment. It was previously shown that ROS in the progenitors primes them for differentiation (Owusu-Ansah and Banerjee, 2009). Our results show that despite having high ROS upon FAO loss, the progenitors are unable to differentiate. We have employed molecular as well as genetic analysis encompassing both RNAi mediated as well as the classical loss of function analyses. Additionally, overexpression studies employing genetic constructs as well as pharmacological methods, firmly establishes the central role of FAO in the submitted manuscript.

General comments on results:Pharmacological treatments: This is another major point that the rest of the paper relies on, yet the authors again show no controls and inadequate quantification.

We had previously incorporated the control for each pharmacological treatment. In the revised manuscript we have quantitated both the controls and experimental. Please refer to figures: Figure 2—figure supplement 1, Figure 4, Figure 5, Figure 7 and Figure 8.

While the entire paper rests on analysis of a few hundred cells in the LG, the Western blot analysis is done with the whole larva and results are interpreted in the context of the LG.

We had done western blot analysis to ascertain the difference, if any, on the level of Histone acetylation in the whole larvae. Once the defect was seen, the same experiment was repeated for acetate-supplemented condition to ensure that, indeed, the supplementation has globally restored the acetylation level.

However, we have not used the above result alone to interpret the result in the context of LG. In addition to several genetic analyses to downregulate FAO, we have done extensive clonal analyses of the lymph gland progenitors. We generated mosaic clones using *hsFlp/Ay-GAL4* mediating RNAi knockdown of *DrosophilaCPT1* orthologue *whd* and subjected them to immunohistochemistry using antibody against H3K9 acetylation. The clonal patches positively marked with GFP (where *whd* is knocked down) showed a significant drop in H3K9 acetylation levels compared to surrounding hemocyte progenitors. Along with H3k9 acetylation, the level of histone H4 acetylation visualized by pan anti-H4 acetylation antibody revealed a drastic drop in total H4 acetylation level in *whd* knockdown clonal patches. Both the expression of H3K9 and pan H4 acetylation remains unaltered in mock/wild type clones.

Please refer to Figure 6K-M of the revised manuscript.

The authors claim that there are metabolic differences between progenitor sub-populations that correlate with potential to differentiate. However pharmacological treatments such as acetate feeding is done at the level of the whole animal. How much of acetate is sensed by the progenitors in question?

Pharmacological feeding was assessed and standardized in a dose-dependent manner. A lower dosage of 1 mM acetate supplemented fly food was not able to rescue the differentiation defects of *whd^1^*. However, a higher dose of 50 mM acetate supplemented fly food was able to rescue the progenitor differentiation as well as histone acetylation levels of the whole larvae.

To assay whether this restores the H3K9 acetylation level in the blood progenitors, we performed anti-immunostaining of H3K9 acetylation and quantitated the mean fluorescence intensity of H3K9 acetylation signal in Dome^+^ progenitors. Please refer to Figure 7H-L of the revised manuscript.

As evident from this figure that although acetate supplementation has no significant effect on control, it rescues the acetylation level and, therefore, the differentiation defect in *whd^1^* homozygous lymph gland.

Thanks for your suggestion for performing immuno-staining of H3K9 acetylation in situ in the LG.

How does a general increase in acetylation affect histone in various tissues including the LG? As systemic signals are key to maintaining blood progenitor homeostasis, the effects seen could be due to cell extrinsic factors.

Acetate supplementation rescued our differentiation defects both in *bsk* as well as *whd* mutant. Our western blot analysis reflected the restoration of histone acetylation in the case of the experiment while there was no change in the control. That this rescue is not systemic becomes further evident when acetate fed control larvae did not show any alteration in differentiation and were comparable to non-fed control. Please refer to Figures 7H-I.

The conclusion that this treatment proves the role of FAO and histone acetylation in progenitor quiescence is far-fetched.

We chose to differ with this statement. Through our extensive molecular and genetic analyses, we have established FAO mediated epigenetic regulation essential for differentiation and not quiescence.

Further there is no evidence of that glycolytic and FAO metabolizing sub-population are differentially affected.

In the current manuscript, we have addressed the differentiation aspect of hemocyte progenitors at third instar larval stages. At the late third instar stage, the LG is populated by Dome^+^ progenitors only. We have shown that JNK mediated FAO activation is essential for the differentiation of the dome^+^ progenitors.

It is similarly unclear in which progenitor sub-populations whd and JNK signaling are affected.

Sorry for the confusion caused. JNK and *whd* signaling are affected in the Dome^+^ progenitors of late third instar LG.

Quantification is required for differentiation checked by P1 and progenitor status should be checked by dome; also differentiation to crystal cells and lamellocytes should be examined.

Thanks for your suggestions. We have included all these analyses in the revised manuscript. Please refer to Figure 2. However, as far as lamellocyte is concerned, neither wild type nor *whd*^1^ mutant lymph glands have them. We have not included this result.

This is the first report of metabolic requirement for progenitor maintenance; a thorough analysis of various differentiation markers is essential.

Thanks for recognizing our effort. In the revised manuscript we have tried our best to strengthen our findings.

Figure wise commentsFigure 1 is fraught with misinterpretations. (Results paragraph one and two). Further there is no quantitation whatsoever and a lack of controls. Two different developmental stages are compared and cell identity is arbitrarily assigned. No markers are used for self-renewing or quiescent cells and it is not clear how the authors identify these. Edu positive cells are only a subset of the second instar LG (these could be differentiating) and seen only in the CZ of the third instar.

In the current version we have used relevant markers to characterize the phenotype.

Figure 1D, E states that self-renewing early stage progenitors exhibit elevated levels of glucose uptake compared to late stage quiescent progenitors. In this case a proliferation assay is required along with markers to demarcate the MZ/CZ boundaries to show that the uptake is less in the prohemocyte pool of the 3rd instar lymph gland.

We want to draw your attention to the fact that there is a technical constraint in performing any antibody assay or proliferation assay with 2-NBDG. Mild fixation is essential for 2 – NBDG assay. Upon normal fixation or more extended washing (required for Antibody staining) 2-NBDG signal quenches. The restricted peripheral signal of 2-NBDG in HmldsRed suggests that the maximum uptake is in the peripheral hemocytes instead of the inner ones (Figure 5—figure supplement 1).

It is likely that the 2NBDG labeled cells could be late progenitors in the second instar that persist in CZ. Without appropriate markers this cannot be distinguished. Further third instar progenitors may be slower proliferating or arrested in the cell cycle, this is not quiescence. A cell cycle analysis needs to be done through L2 and L3 to resolve the two states, so called self-renewal and differentiation.What the authors call quiescence is likely a non-proliferative state that precedes differentiation, which is well documented in many developmental contexts. Similarly as mentioned in Results paragraph two, glycolysis is seen in rapidly proliferating cells, such as cancer cells, this is not necessarily self renewal.GlutGFP expression is patchy (Figure 1F, G) and not clear whether it is in or around the cells, this looks more like background. A no primary antibody control and a Glut RNAi control are required. MZ should be marked by dome or DEcad and perform EdU labeling in order to show that High Glut 1-GFP is expressed in proliferating progenitors. Similarly aldolase expression (supplement 1A, B) is not convincing and is seen only in the periphery of the second instar.

We thank the reviewer for the input. However, in the current manuscript, the focus is on progenitor differentiation and the requirement of FAO in this process. We intend to communicate the significance of glycolysis and early hemocyte progenitor homeostasis in a different manuscript and include the results in it.

Figure 1K: Which area of the LG is shown? Nuclear staining should be included.

Nuclear staining has now been included for most of the figures in the revised version.

Figure 1L-O' it is impossible to comment on the status of mitochondrial network with this analysis. Which part of the LG is shown? High resolution images with live tracking and video microscopy are required to analyze mitochondrial morphology, length and dynamics. Without detailed information regarding these parameters in L2 and L3 progenitors and in differentiated CZ cells, one cannot make any comparisons as there is a large variation in mitochondrial size and morphology within any population. This needs co-analysis with progenitor and differentiation markers and thorough quantitation- CZ/ MZ mitochondrial pattern needs to be mapped first.

The focus of our current submission is FAO’s role in progenitor differentiation. We have concentrated on L3. Since the site of FAO is mitochondria, we wanted to have a feel for the mitochondrial status in the Dome^+^ progenitors. In the current manuscript, we have performed super-resolution microscopy to visualize the mitochondria better, the genotype used is *DomeGal4.UAS-RFP >UAS-mito-GFP*. Please refer to Figure 1—figure supplement 1 and Video 1.

Figure 1Q, R: the accumulation/increase of Lsd-2 in the 3rd instar lymph gland would be clear if 2nd instar images are presented for Nile Red.

Nile red data have now been endorsed by LipiTOX (a validated assay for neutral lipids). All these results have been quantified and presented in Figure 1—figure supplement 2A-C.

Figure 1 requires quantification and insets indicating region of interest; double labeling with progenitor marker and or CZ markers or EdU labeling is essential.

We have addressed all these issues in the current Figure 1.

Figure 2: Dapi staining or demarcation of LG boundary as well as MZ/CZ marker staining should be shown. Crystal cell and lamellocytes differentiation status should also be analyzed.

We have included the DAPI channel and the quantitation data for this experiment. Please refer to Figure 2.

2H,L Q Graph is misleading, it suggests there is a decrease in dome^+^ area. Y axis should indicate plamatocyte % or area. What is differentiation index?

We have modified the Y-axis to cortical zone area by total lymph gland area in Figure 2.

Figure 3E-G, the authors claim higher number of EdU incorporation is seen in the progenitors, however the images provided show higher EdU signal at the periphery; also CZ/MZ boundary is not demarcated, progenitor marker should be used and data quantitated.

We have performed the same experiment with markers for the progenitor zone (Dome^+^). Quantitative analysis has now been included for the same to resolve the confusion.

Figure 5F-H quantitation required for EdU labeling to show that feeding with 2-DG rescues the hyperproliferation phenotype.

We have provided the required quantitation for this experiment. Please refer to Figure 5I-L.

Figure 6A-C claims loss of Histone Acetyl Transferase (HAT), Gcn5 and chm phenocopies differentiation defect seen in the progenitors of FAO loss of function. What is the effect on proliferation in panels 6A-G. Does it corroborate with active proliferation seen in whd mutants.

We thank the reviewer for the suggestion. We have performed the EdU incorporation in Dome-GAL4-UAS-GFP> UAS-Gcn5-RNAi and Dome-GAL4-UAS-GFP> UAS-chm-RNAi. In both, the case unlike whd1 mutant, the hemocytes were not proliferating. We can infer that Gcn5 and chm facilitate hemocyte progenitor differentiation.

For Figure 7F materials method suggests western blots were done from larva of the respective genotypes, this cannot be correlated to the LG effects. It would be more appropriate to show H3K acetylation in situ in the lymph gland.

Thanks for your comments. We have now provided evidences of the upregulation of H3K9 acetylation level in the lymph gland by immunostaining in both Acetate as well as L-carnitine fed conditions.

Please refer to Figure 7H-L and the main text of the revised manuscript for acetate supplementation. For L-carnitine supplementation, please refer to Figure 7O-Q for H3K acetylation status in situ in the lymph gland.

Figure 8A- C, expression of dominant negative allele of basket (bsk, *Drosophila* ortholog of Jun-Kinase) results in a compromised differentiation analogous to whd knockdown. Can the phenotype be rescued by feeding L-carnitine or acetate?

We thank the reviewer for this suggestion.

We have done both acetate as well as L-carnitine supplementation *Dome> bskDN*.

Indeed in both cases, we see a significant rescue of differentiation. Please refer to (Figure 8M-Q) for acetate feeding and L-carnitine supplementation (Figure 8R-V).

These results are discussed in in the main text of the revised manuscript

The effect of Jun (bsk) on whd is shown at the transcriptional level. Does Jun (bsk) affect enzymes of the FAO pathway?

Thank you very much for this suggestion.

We have performed qRT-PCR analysis of transcripts of enzymes involved in FAO pathway in Dome-GAL4- UAS-GFP> UAS-Bsk^DN^. The transcription of *CPT1/whd, Mcad, Mtpa, Scully, Mtpb, and yip2* were assayed upon down-regulation of *bsk* from hemocyte progenitors. The transcript level of *CPT1/whd* indicated a ~41% drop while the expression of other b-oxidation enzymes showed no significant alteration. This observation established that JNK controls the transcription of *CPT1/whd,* thereby regulating FAO. Please refer to Figure 8I and the main text of the revised manuscript.

Also Jnk is known to activate Foxo, what is the status of Foxo reporter (Thor-LacZ) in whd mutants.

Due to the technical limitation of recombining thor lacZ with *whd^1^* (both are on the second chromosome), we have used another bona fide reporter for JNK: puckered (puc)lacZ. Our analysis reveals that in the *whd^1^*mutant lymph gland, there is a significant increase in puc lacZ expressing cells indicating a higher JNK activation in the progenitors. It therefore is evident that in *whd*^1^ LG high ROS and high JNK are unable to initiate differentiation in absence of FAO.

**Author response image 1. sa2fig1:** 

Foxo overexpression advances differentiation (Owusu-Ansah and Banerjee, 2009), does overexpression of Foxo affect/ rescue whd mutant phenotype by inducing differentiation.

Thanks for your suggestion. We have done these experiments. Upon overexpression of Foxo, we also see a similar increase in differentiation, as reported (Owusu-Ansah and Banerjee, 2009). However, one copy loss of *whd* in the background of Foxo overexpression rescues the ectopic differentiation. Please refer to Figure 8D-H and the main text of the revised manuscript.

[Editors’ note: what follows is the authors’ response to the second round of review.]

Essential revisions:Authors should talk to non-*Drosophila* colleagues to help them in thoroughly revising the following aspects of the manuscript:1) Abstract is not well written and should be improved, specially the last two sentences. It does not summarize the major aspects of the work and will not attract the attention of any reader that has not a special interest in the *Drosophila* LG.

Thanks for your suggestion. We have accordingly worked on the Abstract as well as the Introduction and made necessary changes to attract the attention of the readers.

2) Cartoon depicted in Figure 1A and explanation in the Introduction do not have anything to do with each other (CZ only explained in the text and others only in the figure) in the figure and does not clarify anything. It's just confusing and should be completely changed and re-done.

We have worked on Figure 1A and also on the explanation of the same. Please refer to Introduction paragraph three.

3) Last paragraph of the Introduction should summarize the major findings and they do not so. References to the JNK/ROS axis are lacking. Jnk itself is not introduced. Again, this part of the manuscript is intended to attract the non-expert community

Please refer to the revised version of the manuscript final paragraph of the Introduction. We have built up the last paragraph keeping the suggestions in mind.

Thanks for your input.

4) First part of the results: the why of using two developmental stages and different markers (pvf2, Dome, Hml, etc) is not clearly explained anywhere.

*Drosophila* larval lymph gland consists of heterogeneous progenitors. Using molecular markers, these progenitor subpopulations are referred to as pre-progenitors, progenitors, and intermediate progenitors, respectively (Figure 1). The pre-progenitors can also be visualized by Pvf2 expression in first, second, and early third instar larval lymph gland. However, by the late third instar, only progenitors and intermediate progenitors are present in the lymph gland. Therefore, it was necessary to use an early third instar stage to characterize the pre-progenitors. We have explained this in the revised manuscript in the first paragraph of the Results section.

Dome^+^, Dome^+^ Pxn^+^, Dome^-^, etc nomenclature is confusing (2N-O). Any way of improving it? Authors should be able to avoid technicalities.

We have used the published annotation to avoid confusion. However, instead of using these nomenclatures throughout, in the revised manuscript, we have once described it and thereafter described them without the technical details. “These progenitor subpopulations will be henceforth referred to as pre-progenitors, progenitors, and intermediate progenitors (IPs), respectively.”

5) Tens of mistakes (prolifeartes, progenitors experiences, etc) should be corrected.

We are extremely sorry for these unintentional mistakes. We have tried to fix all of them. Thanks for pointing them out.

6) Writing and conclusions in all sections should be improved.

Thanks for your input. We have worked extensively on the writing and conclusions in all the sections keeping the reviewers suggestions in mind.

Scientific issues:7) Lipid markers and CG3902 expression. A Z section is required to demonstrate that the differential expression/incorporation levels between progenitor and differentiated cells is real not a consequence of the focal plane or thickness of the tissue.

We want to draw your attention for the mentioned figures. Five optical sections of 1µm thickness from the middle of the Z stack were merged into a single section. We have mentioned this detail in the respective figure legends.

Therefore, the differential expression/incorporation levels observed between progenitor and differentiated cells evident are not an outcome of the focal plane or thickness of the tissue.

Please find the montage of the confocal stacks from which the Z projection was used for the panel for your information.

**Author response image 2. sa2fig2:** CG3902 YFP. Five optical sections of 1µm thickness from the middle of the Z stack (No. 9-13) were merged into a single section for this panel.

**Author response image 3. sa2fig3:** LipidTOX. Five optical sections of 1µm thickness from the middle of the Z stack (No. 7-11) were merged into a single section for this panel.

**Author response image 4. sa2fig4:** Nile Red. Five optical sections of 1µm thickness from the middle of the Z stack (No. 10-14) were merged into a single section for this panel.

8) Authors should use ubiquitous Gal4 drivers (NOT dome-Gal4) to demonstrate differences in the amount of mitochondria and cell cycle profiles (Fucci) between progenitor and differentiated cells.

In the first version of the manuscript, we did use the Ubi-Gal4 to drive UAS-mitoGFP. However, we were suggested in the first review to use Dome gal4 instead. Since we were characterizing the dome progenitors, we went ahead and did dome-Gal4>UAS-mito-HA-GFP or Dual FUCCI.

Also, to be noted for reasons unknown, Ubi-Gal4 is not uniformly expressed in the lymph gland.

**Author response image 5. sa2fig5:** 

Since we have already analyzed the progenitor pool, the best we could do was to look at the differentiated cells to have a complete picture. Taking the suggestion in mind, we used Pxn-Gal4 to drive UAS-mito-HA-GFP and UAS-FUCCI to assay the mitochondria and cell cycle status of the differentiating cells. Since the differentiated cells are not the focus of the current study, we have not included these data in the revised manuscript. However, we would like to share our findings with the reviewers.

**Author response image 6. sa2fig6:** 

This experiment reveals that there are plenty of reticular mitochondria in the differentiating hemocytes. Despite having plenty of mitochondria, our results demonstrate a higher glucose uptake in the differentiating hemocytes with basal level expressions of the components of FAO. These results indicate that glucose metabolism followed by oxidative phosphorylation might be prevalent in the differentiating cells. It will be interesting to pursue metabolic regulation of differentiating cells as a separate project.

**Author response image 7. sa2fig7:** 

This experiment demonstrates that in the late third instar lymph gland most of the differentiating cells are in S phase (red), very few in G1 (green) and G2-M(yellow).

9) Some negative results (changes in Hh and ROS levels w/o any impact on the process) should be moved to supplementary data.

We have moved these two data in the Supplementary Figure. Please refer to: Figure 3—figure supplement 1. High ROS is associated with the differentiation of the dome^+^ progenitor (Owusu-Ansah and Banerjee, 2009). However, we found that the progenitors of whd^1^ despite having high levels of ROS are unable to differentiate. At this point, we started appreciating the role of FAO in differentiation. Keeping this twist in mind, although we have moved this data in a supplementary figure, we chose to club it under separate heading to emphasize the importance of FAO in the differentiation process.

Please refer to subsection “Failure in differentiation of hemocyte progenitors upon loss of FAO is not due to decline in ROS levels”.

10) CycB degradation of FlyFucci is not degraded in G1, as authors claim in subsection “Loss of FAO caused an increase in progenitor proliferation, and high redox levels”.

Our sincere apology for this mistake, thanks a lot for bringing this to our notice. We have corrected it in the revised version.

11) Single channels to monitor differences in EdU incorporation levels should be included (3G-I, 5M-P). Figure 5E, F is not described properly in the text.

We have included the single channel for EdU incorporations in all the mentioned figures. Please refer to Figure 3A-B', Figure 4 I-J', Figure4 L-M', and Figure 5 I-L' of the revised manuscript. We have described Figure 5E, F in subsection “FAO loss in hemocyte progenitors leads to sustained glycolysis”.

12) Y axis in panel 3F: should be ROS and not Ci, right?

Sorry for this unintentional mistake. Thanks for pointing this out. We have corrected it in the revised version.

13) Materials and methods: Codes and/or references of all Fly stocks required. By the way, many of the fly lines used in the work are not included in Materials and methods. Stocks such as AyGal4 should be properly explained.

We have taken care of all of these in the revised version. Please also refer to the Materials and methods for details of stocks generated for this study. AyGal4 is now described in the Materials and method section in details.

14) There is a lot of room for improvement in the organization of panels within Figures. Labels of most histograms are too small and error bars are not explained in the legends (SD or SDM?).

Thanks for your suggestions. We have worked on them and are open to any further suggestions. We have increased font size of the labels of all the histograms and have now explained the error bar as SDM in the figure legends and Materials and methods section.

15) What is the trigger for differentiation? Is dietary lipid content a regulator of this as has been shown in vertebrates for regulation by FAO?Mere carnitine supplementation is not sufficient as carnitine is not an essential nutrient, is abundant in the body under regular diet and carnitine deficiency results from mutations in its transport machinery. As the claim is that whd links JNK to FAO, and is transcriptionally regulated by JNK, a genetic rescue experiment with whd is essential, especially as Foxo overexpression increases differentiation. This manuscript often relies on compound treatment strategies alone despite having a genetically tractable model. Yet the role and effect of endogenously generated systemic cues is completely disregarded.

We want to draw your attention to a genetic manipulation that we had done in the previously submitted manuscript (Figure 8D-H). Genetic removal of one copy of whd was sufficient to prevent the precocious differentiation observed in progenitor specific overexpression of FOXO. To endorse our claim further in the current manuscript, we done a converse experiment. We have overexpressed whd in the progenitors, which lacked JNK (dome-Gal4. UAS-GFP>UAS-bsk^DN^, whd-OE). Our results show that the progenitors that were unable to differentiate upon loss of JNK, resume their differentiation, with whd overexpression. Please see Figure 8—figure supplement 1L-P.

From both the genetic data we can conclude that FAO acts downstream of JNK to facilitate differentiation. Please refer to paragraph six of subsection “JNK signaling regulates FAO in the hemocyte progenitors”.

[Editors' note: further revisions were suggested prior to acceptance, as described below.]

Essential revisions1) In the Introduction, the authors contrast hematopoietic stem cells (HSCs) in mammals with *Drosophila* hematopoietic progenitors, assuming that studying *Drosophila* hematopoietic progenitors will provide insights into how mammalian non-HSC progenitors are regulated by FAO. However, given that *Drosophila* does not have HSCs, and whether they represent HSCs or progenitors remain unclear (or whether it's comparison to mammalians HSC vs. progenitor is more relevant). In mammalian HSCs, FAO is required for stem cell maintenance, whereas in *Drosophila* LG, FAO is required for differentiation. I see that this difference may have driven the authors to make the above contrast, but I don't think it's a well-grounded argument and they should modify the Discussion accordingly.

We have modified the Introduction and Discussion accordingly in the revised version.

2) Figure 1—figure supplement 1C shows reticular mitochondria as an indication of high FAO. Ideally, they should show less reticular mitochondria in other cell types in comparison.

We are happy to let you know that we had this data for the differential mitochondrial structure in differentiated hemocytes compared to progenitors (Streptavidin marking the mitochondria in Hml >GFP lymph gland). For reasons unknown to us HmlΔ-GAL4 is unable to drive UAS-mito-HA-GFP, which we had employed to assay the status of mitochondria in the progenitors. We, had therefore, used streptavidin labeling to visualize the mitochondrial status in the differentiated hemocytes. Interestingly, differentiated hemocytes (HmlΔ-GAL4> UAS-GFP) show less reticular mitochondria, in comparison to the progenitors (Hml>GFP negative). We have included this result in the revised version (please refer to Figure 1—figure supplement 1D-D').

3) Figure 4: to nail down the pathway, the epistasis analysis would be ideal, such as combination of genetic mutations (inhibit differentiation, such as whd mutant) and L-carnitine (drive differentiation). Especially, they conduct similar experiments in Figure 5, Figure 6, and Figure 8.

Although this could have been an ideal experiment, we did not carry out this analysis previously. Since whd null mutant and carnitine, both are part of the carnitine shuttle that imports carnitine fat moiety into mitochondria. CPT1/whd is the enzyme present in the outer mitochondrial membrane, which converts fatty acyl-CoA into fatty acyl-carnitine by utilizing the carnitine moiety. Therefore, we rationalized that upregulation of FAO by carnitine supplementation in whd null homozygous mutants would not aid in the epistatic analysis.

However, we have now used whd1 heterozygous, which incidentally also gives an impressive phenotype (more progenitor, less differentiation compared to control). L-carnitine feeding to the heterozygous restores differentiation to a great extent. We have included this in the revised version (please refer to Figure4—figure supplement 1A-E).

4) Figure 5I-L, 2-DG treatment rescues G2/M arrest defects in whd mutant, but does not seem to rescue differentiation defect. Any discussions?

We had mentioned this aspect in the Result section. Now in the revised version we have expanded this thought and included it in the Discussion, as “The glycolytic surge in FAO mutants is not capable to take them through the differentiation process. This indicates that for the process of differentiation the acetyl moiety derived from FAO plays a key role to facilitate hemocyte progenitor differentiation”.